# Closing the Curvature Gap: Full Transformer Hessians and Their Implications for Scaling Laws

## Abstract

The lack of theoretical results for Layer Normalization and feedforward Hessians has left a gap in the study of Transformer optimization landscapes. We address this by deriving explicit second-order expressions for these components, thereby completing the Hessian characterization of full Transformer blocks. Our results generalize prior self-attention analyses and yield estimations for the role of each sublayer in curvature propagation. We demonstrate how these Hessian structures inform both convergence dynamics and the empirical scaling laws governing large-model performance. Further, we propose a Taylor-expansionbased framework for analyzing loss differences to quantify convergence trajectories. By extending Hessian theory to the full Transformer architecture, this work establishes a new foundation for theoretical and empirical investigations of optimization in large-scale deep learning.

**Keywords:** Transformer Hessians, Layer Normalization, Scaling laws, Convergence dynamics, Loss landscape, Optimization geometry.

## 1 Introduction

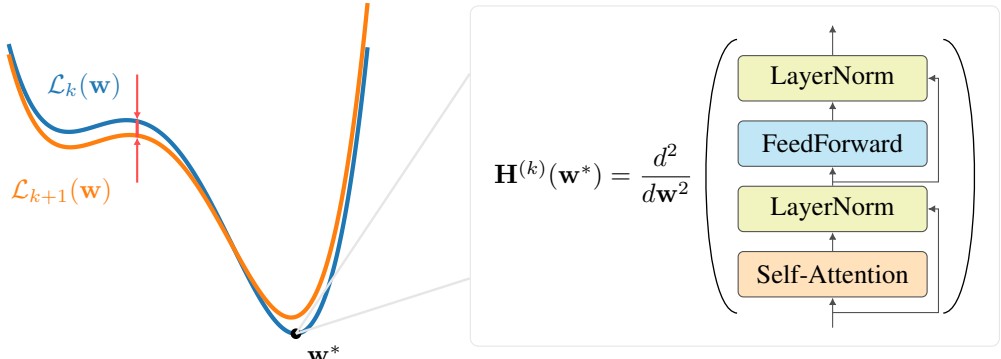

**(a)** Loss landscape convergence    **(b)** Hessian-based Transformer analysis

Figure 1: **Overview of our observations.** Part (a) shows the loss function landscape, which is a surface in the parameters space, and how it changes as the dataset size increases. Part (b) shows the schematic view of a proposed method — carry out an analysis of a Transformer's Hessian, which greatly impacts on a loss landscape convergence, leading to a sample size determination framework.

Transformers Vaswani et al. (2017) have revolutionized deep learning, achieving state-of-the-art performance across natural language processing Devlin et al. (2019); Brown et al. (2020), computer vision Dosovitskiy et al. (2021); Wu et al. (2020), Their empirical success is underpinned by predictable improvements in model quality with increased dataset size, as described by neural scaling laws Kaplan et al. (2020); Hoffmann et al. (2022); Bahri et al. (2024). However, many domains, such

as medical imaging Poulain et al. (2022) and scientific discovery Jumper et al. (2021), face severe data constraints where acquiring additional samples is costly or infeasible Chen et al. (2025). This tension necessitates a rigorous theoretical understanding of how dataset size shapes the optimization landscape and influences training dynamics.

Existing theoretical analyses of Transformer optimization landscapes are incomplete. While recent studies have derived Hessian expressions for self-attention mechanisms Ormaniec et al. (2024); Zhang et al. (2024), the full Transformer blockincluding LayerNorm and feed-forward networks (FFNs)lacks a comprehensive theoretical characterization Noci et al. (2022); Zhang et al. (2025a). These components critically influence optimization dynamics, such as gradient flow and convergence rates Noci et al. (2022); Yang et al. (2024), and generalization behavior Zhang et al. (2025b); Csordás et al. (2021). Without a complete curvature analysis, our understanding of Transformer training dynamics, convergence properties, and scaling behavior remains limited Fort and Jastrzebski (2019).

In this work, we provide the first complete theoretical analysis of the Hessian for full Transformer blocks, extending beyond prior self-attention analyses Ormaniec et al. (2024); Zhang et al. (2024) to include explicit second-order expressions for LayerNorm and FFNs. Our analysis derives rigorous bounds on how the loss landscape evolves with dataset size, offering a novel framework for understanding landscape stabilization in Transformers. These results have implications for optimization challenges (e.g., vanishing gradients Hochreiter (1998)), scaling laws (e.g., compute-optimal training Kaplan et al. (2020); Hoffmann et al. (2022)), and critical batch size estimation McCandlish et al. (2018); Zhang et al. (2025c).

**Contributions.** Our main contributions are:

- We derive the first full Hessian expressions for Transformer blocks, including explicit treatment of LayerNorm and FFNs, filling a critical gap in prior analyses.

- We establish theoretical bounds on the loss landscapes evolution with dataset size, providing a rigorous framework for understanding landscape stabilization.

- We validate our theoretical predictions through experiments on Vision Transformers, demonstrating practical relevance across data regimes.

Our work bridges theoretical deep learning and practical Transformer deployment, enabling new insights into optimization difficulties, efficient scaling strategies, and future theoretical investigations of large-scale deep learning.

**Outline.** The rest of the paper is organized as follows. In Section 2, we review related work, categorizing existing research into key topics and highlighting their main contributions. Section 3 introduces the notation and presents preliminary calculations essential for our analysis. In Section 4, we derive theoretical bounds for the norm of the Hessian matrix and the norm of the difference between loss functions. Section 5 provides an empirical study validating these theoretical results. Section 6 discuss and summarize our findings, offering insights and conclusions. Additional experiments are in Appendix A and proofs of theorems are included in Appendices B-D.

## 2 RELATED WORK

**Geometry of Neural Network Loss Landscapes** Foundational studies characterize neural loss geometry via Hessians, including class-aligned high-curvature directions Fort and Jastrzebski (2019), random-matrix perspectives on spectra and optimization Pennington et al. (2017), and connectivity and double-descent phenomena Garipov et al. (2018); Singh et al. (2022); Draxler et al. (2019); Nguyen et al. (2017), with flattening observed at large learning rates Wang et al. (2023). Our work complements this line by showing how curvature of Transformer blocks changes with dataset size, providing explicit second-order bounds that formalize landscape stabilization under data growth. This links classical geometric insights to a data-scaling axis that was previously qualitative.

**Hessian-Based Analysis and Generalization** Prior Hessian analyses for fully connected and convolutional networks reveal spectral structure and low effective rank with implications for convergence and smoothness Kiselev and Grabovoy (2024); Meshkov et al. (2024). We extend these ideas to

Transformers by deriving explicit LayerNorm/FFN second derivatives and blockwise spectral-norm bounds, thereby closing a missing piece in second-order geometry for this architecture.

**Loss Landscapes in Transformers** While Transformers Vaswani et al. (2017) have inspired curvature analyses focused on attention Ormaniec et al. (2024) and studies of sample complexity, generalization, and stagewise dynamics Zhang et al. (2025b); Li et al. (2023); Hoogland et al. (2025), a full-block second-order treatment has remained incomplete. We provide the missing LayerNorm/FFN Hessians and assemble a complete blockwise Hessian for a Transformer layer, aligning theory with empirical curvature structure. This enables a principled account of how Transformer curvature evolves with data and training.

**Dataset Size and Loss Landscape Convergence** Work on compute-optimal scaling and sample-related flatness highlights the importance of balancing data and model size Hoffmann et al. (2022); Wu et al. (2017), and visualization tools hint at stabilization thresholds without theory Xie et al. (2024). Building on Hessian frameworks from other architectures Kiselev and Grabovoy (2024); Meshkov et al. (2024) and attention derivatives Ormaniec et al. (2024), we derive a second-order bound that decays as $1/k$. This yields actionable diagnostics for curvature-aware training and data budgeting in Transformers.

## 3 PRELIMINARIES

We adopt row-wise vectorization $\text{vec}_r(\cdot)$ from Ormaniec et al. (2024); Noci et al. (2022). For a matrix-valued function $\mathbf{N} : \mathbb{R}^{p \times q} \to \mathbb{R}^{n \times d}$ differentiable w.r.t. weight matrices $\mathbf{W}_i \in \mathbb{R}^{p_i \times q_i}$ and $\mathbf{W}_j \in \mathbb{R}^{p_j \times q_j}$, the Jacobian is $\frac{\partial \mathbf{N}}{\partial \mathbf{W}_i} := \frac{\partial \text{vec}_r(\mathbf{N})}{\partial \text{vec}_r(\mathbf{W}_i)^\top} \in \mathbb{R}^{nd \times p_i q_i}$, and the Hessian block is

$\frac{\partial^2 \mathbf{N}}{\partial \mathbf{W}_i \partial \mathbf{W}_j} := \frac{\partial \text{vec}_r(\frac{\partial \mathbf{N}}{\partial \mathbf{W}_i})}{\partial \text{vec}_r(\mathbf{W}_j)^\top} \in \mathbb{R}^{(nd \cdot p_i q_i) \times p_j q_j}$. Key properties (e.g., for products, Kronecker, inverses, Hadamard powers) are detailed in Appendix B.

Let $f_{\mathbf{w}}(\cdot)$ denote a neural network (here, a Self-Attention layer or full Transformer block) with parameters $\mathbf{w} \in \Omega$. Given a twice-differentiable loss $l(\cdot, \cdot)$, the per-sample loss is $l_i(\mathbf{w}) := l(f_{\mathbf{w}}(\mathbf{x}_i), \mathbf{y}_i)$. The empirical loss over $L = k$ samples is $\mathcal{L}_k(\mathbf{w}) = \frac{1}{k} \sum_{i=1}^{k} l_i(\mathbf{w})$, with Hessian $\mathbf{H}^{(k)}(\mathbf{w}) = \frac{1}{k} \sum_{i=1}^{k} \nabla_{\mathbf{w}}^2 l_i(\mathbf{w})$.

**Assumption 1.** *At local minimum* $\mathbf{w}^*$, $\nabla \mathcal{L}_{k-1}(\mathbf{w}^*) = \nabla \mathcal{L}_k(\mathbf{w}^*) = 0$.

Our study on the feasibility of this assumption is in Appendix A.2.

Consider input embeddings $\mathbf{X} \in \mathbb{R}^{L \times d_V}$. A single-head Self-Attention layer outputs

$$\mathbf{F}(\mathbf{X}) = \mathbf{A}(\mathbf{X})\mathbf{X}\mathbf{W}_V, \tag{1}$$

where $\mathbf{A}(\mathbf{X}) = \text{softmax}\left(\frac{\mathbf{X}\mathbf{W}_Q \mathbf{W}_K^\top \mathbf{X}^\top}{\sqrt{d_K}}\right)$, and $\mathbf{W}_Q, \mathbf{W}_K \in \mathbb{R}^{d_V \times d_K}$, $\mathbf{W}_V \in \mathbb{R}^{d_V \times d_V}$.

Full Transformer block is:

$$\text{LayerNorm}\Big(\text{LayerNorm}(\mathbf{X} + \mathbf{F}(\mathbf{X})) + \text{FFN}(\text{LayerNorm}(\mathbf{X} + \mathbf{F}(\mathbf{X})))\Big) \tag{2}$$

where $\text{FFN}(\cdot)$ is a fully connected block with a non-linear activation within it. LayerNorm for an input matrix $\mathbf{U} \in \mathbb{R}^{m \times n}$ is $\text{LayerNorm}(\mathbf{U})_{i,j} = \gamma_j \frac{\mathbf{U}_{i,j} - \mu_i}{\sqrt{\sigma_i^2}} + \beta_j$, where $\mu_i = \frac{1}{m} \sum_{j=1}^{m} \mathbf{U}_{i,j}$, $\sigma_i^2 = \frac{1}{m} \sum_{j=1}^{m} (\mathbf{U}_{i,j} - \mu_i)^2$. More details on a transformer block are in Section 4.2.

**Assumption 2.** *For input matrices to LayerNorm (e.g.,* $\mathbf{X} + \mathbf{F}(\mathbf{X})$, $\mathbf{Y} + \text{FFN}(\mathbf{Y})$*), the per-row variances satisfy* $\min_i \sigma_i^2 > 0$.

It's a technical assumption for the proof part simplification and numerical stability. The same effect can be achieved by adding some positive constant to the denominator, but it makes calculations harder. In our case this assumption is required for $\mathbf{X} + \mathbf{F}(\mathbf{X})$ and $\mathbf{Y} + \text{FFN}(\mathbf{Y})$, defined in Transformer block 5.

We use mean-squared error loss: $l(\cdot, \mathbf{Target}) = \frac{1}{L d_V} \|\cdot - \mathbf{Target}\|_F^2$. Hessians decompose via Gauss-Newton: for composite $\mathcal{L}_k \circ f_{\mathbf{w}}$,

$$\frac{\partial^2 (\mathcal{L}_k \circ f_{\mathbf{w}})}{\partial \mathbf{W}_i \partial \mathbf{W}_j} = \frac{\partial f_{\mathbf{w}}}{\partial \mathbf{W}_i}(\cdot)^\top \frac{\partial^2 \mathcal{L}_k}{\partial f_{\mathbf{w}}^2}(f_{\mathbf{w}}(\cdot)) \frac{\partial f_{\mathbf{w}}}{\partial \mathbf{W}_j}(\cdot) + \left(\frac{\partial \mathcal{L}_k}{\partial f_{\mathbf{w}}}(f_{\mathbf{w}}(\cdot)) \otimes \mathbf{I}_{p_i q_i}\right) \frac{\partial^2 f_{\mathbf{w}}}{\partial \mathbf{W}_i \partial \mathbf{W}_j}(\cdot) \tag{3}$$

## 4 METHOD

In this section, we derive generalized Hessian expressions for the self-attention layer and extend them to a full transformer block, leveraging these to analyze the convergence of the loss function surface as the dataset size increases. Our approach builds on the theoretical framework of Ormaniec et al. (2024), adapting and generalizing their results.

### 4.1 HESSIAN OF THE SELF-ATTENTION LAYER

We begin by analyzing the Hessian of a single self-attention layer with parameters $\mathbf{w} = \{\mathbf{W}_Q, \mathbf{W}_K, \mathbf{W}_V\}$ as defined in Equation 1. The empirical loss is defined as: $\mathcal{L}_k(\mathbf{w}) = \frac{1}{k} \sum_{i=1}^{k} l(\mathbf{F}(\mathbf{X}_i), \mathbf{Target}_i)$, where $l(\mathbf{F}(\mathbf{X}_i), \mathbf{Target}_i)$ is a Loss function defined above.

The Hessian of $\mathcal{L}_k$ with respect to the parameters $\mathbf{w}$ is:

$$\mathbf{H}^{(k)}(\mathbf{w}) = \nabla_{\mathbf{w}}^2 \mathcal{L}_k(\mathbf{w}) = \frac{1}{k} \sum_{i=1}^{k} \nabla_{\mathbf{w}}^2 l_i(\mathbf{w}) = \frac{1}{k} \sum_{i=1}^{k} \mathbf{H}_i(\mathbf{w})$$

where $\mathbf{H}_k(\mathbf{w})$ is a hessian of the Self-Attention block for $\mathbf{w}$ being a pair of matrices from $\{\mathbf{W}_Q, \mathbf{W}_K, \mathbf{W}_V\}$. It can decomposed using the Gauss-Newton approximation 3:

$$\mathbf{H}_k(\mathbf{W}_i, \mathbf{W_j}) = \frac{\partial^2 l}{\partial \mathbf{W}_i \partial \mathbf{W}_j} = \mathbf{H}_o(\mathbf{W}_i, \mathbf{W}_j) + \mathbf{H}_f(\mathbf{W}_i, \mathbf{W}_j),$$

with $\mathbf{H}_o$ as the outer-product Hessian and $\mathbf{H}_f$ as the functional Hessian. The results for this decomposition can be calculated according to Theorems 3.1-3.2 from Ormaniec et al. (2024).

**Hessian's norm estimation**

Next, we introduce a theorem for estimation the spectral norm (Definition 1) of the Hessian for a single Self-Attention block.

**Theorem 1.** *Let $\| \cdot \|_2$ be a spectral matrix norm, then for a single Self-Attention layer we have*

$$\|\mathbf{H}_i(\mathbf{w}^*)\|_2 \leq M$$

*where*

$$M = 3 \max \left( \frac{2L}{d_V} \|\mathbf{X}\|_2^2, \right.$$

$$\frac{8}{L^3 d_V d_K} \|\mathbf{W}_K\|_2^2 \|\mathbf{W}_V\|_2^2 \|\mathbf{X}\|_2^6 + \frac{12}{d_V d_K} \sqrt{\min(L, d_V)} (L\|\mathbf{X}\|_2 \|\mathbf{W}_V\|_2 + \|\mathbf{Target}\|_2) \|\mathbf{W}_V\|_2 \|\mathbf{W}_K\|_2^2 \|\mathbf{X}\|_2^5,$$

$$\frac{4}{L d_V \sqrt{d_K}} \|\mathbf{W}_V\|_2 \|\mathbf{W}_K\|_2 \|\mathbf{X}\|_2^4 + \frac{4\sqrt{\min(L, d_V)}}{L^2 \sqrt{d_K}} (L\|\mathbf{X}\|_2 \|\mathbf{W}_V\|_2 + \|\mathbf{Target}\|_2) \|\mathbf{W}_K\|_2 \|\mathbf{X}\|_2^3,$$

$$\frac{8}{L^3 d_V d_K} \|\mathbf{W}_K\|_2 \|\mathbf{W}_Q\|_2 \|\mathbf{W}_V\|_2^2 \|\mathbf{X}\|_2^6 +$$

$$\left. + \frac{4\sqrt{\min(L, d_V)} (L\|\mathbf{X}\|_2 \|\mathbf{W}_V\|_2 + \|\mathbf{Target}\|_2)}{L d_V \sqrt{d_K}} \|\mathbf{W}_V\|_2 \left( 3L\|\mathbf{W}_K\|_2 \|\mathbf{W}_Q\|_2 \|\mathbf{X}\|_2^5 + \frac{d_V}{L} \|\mathbf{X}\|_2^3 \right) \right)$$

*The proof is provided in Appendix C.1.*

### 4.2 HESSIAN OF THE TRANSFORMER BLOCK

A transformer block extends the self-attention layer with a feed-forward network (FFN), residual connections, and layer normalization. The output is:

$$\mathbf{Y} = \text{LayerNorm}(\mathbf{X} + \mathbf{F}(\mathbf{X})) \tag{4}$$

$$\mathbf{Z} = \text{LayerNorm}(\mathbf{Y} + \text{FFN}(\mathbf{Y})), \tag{5}$$

where $\mathrm{FFN}(\mathbf{Y}) = \sigma(\mathbf{Y}\mathbf{W}_1 + \mathbf{b}_1)\mathbf{W}_2 + \mathbf{b}_2$, with $\mathbf{W}_1 \in \mathbb{R}^{d_V \times d_{\mathrm{ff}}}$, $\mathbf{W}_2 \in \mathbb{R}^{d_{\mathrm{ff}} \times d_V}$, $b_1 \in \mathbb{R}^{d_{\mathrm{ff}}}$, $b_2 \in \mathbb{R}^{d_V}$, and $\sigma$ as the activation (e.g., ReLU). The $\mathrm{LayerNorm}(\mathbf{X})$ operation is defined as follows. For an input matrix $\mathbf{X} \in \mathbb{R}^{L \times d_V}$, we compute:

1. Feature-wise mean and variance:

$$\mu_i = \frac{1}{d_V}\sum_{j=1}^{d_V}\mathbf{X}_{i,j}, \quad \sigma_i^2 = \frac{1}{d_V}\sum_{j=1}^{d_V}(\mathbf{X}_{i,j} - \mu_i)^2,$$

2. Normalized output with learnable parameters $\gamma, \beta \in \mathbb{R}^m$:

$$\mathrm{LayerNorm}(\mathbf{X})_{i,j} = \gamma_j \cdot \frac{\mathbf{X}_{i,j} - \mu_i}{\sqrt{\sigma_i^2}} + \beta_j.$$

The parameters are $\mathbf{w} = \{\mathbf{W}_Q, \mathbf{W}_K, \mathbf{W}_V, \mathbf{W}_1, \mathbf{W}_2, \mathbf{b}_1, \mathbf{b}_2, \gamma, \beta\}$, where $\gamma$ and $\beta$ are the scale and shift parameters of LayerNorm. For simplicity in Hessian analysis, one may assume $\gamma$ and $\beta$ are fixed (e.g., $\gamma = \mathbf{1}$, $\beta = \mathbf{0}$), though they are typically learnable.

**Theorem 2** (Jacobian of LayerNorm). *Let* $\mathbf{X} \in \mathbb{R}^{L \times d_V}$. *Define*

$$\mathbf{M}(\mathbf{X}) = \mathbf{X} - \frac{1}{d_V}\mathbf{X}\mathbf{1}_{d_V}\mathbf{1}_{d_V}^\top, \quad \sigma(\mathbf{X}) = \frac{1}{\sqrt{d_V}}\big(\mathbf{M}(\mathbf{X})^{\circ 2}\mathbf{1}_{d_V}\big)^{\circ 1/2}, \quad \mathbf{P}(\mathbf{X}) = \mathrm{diag}^{-1}(\sigma(\mathbf{X})).$$

*Then the Jacobian of*

$$LayerNorm(\mathbf{X}) = \mathbf{P}(\mathbf{X})\mathbf{M}(\mathbf{X})$$

*with respect to* $\mathbf{X}$ *is*

$$\frac{\partial\, LayerNorm(\mathbf{X})}{\partial \mathbf{X}} = (\mathbf{P}(\mathbf{X}) \otimes \mathbf{I}_{d_V})\left(\mathbf{I}_{Ld_V} - \frac{1}{d_V}(\mathbf{I}_L \otimes \mathbf{1}_{d_V \times d_V})\right) + (\mathbf{I}_L \otimes \mathbf{M}(\mathbf{X})^\top)\frac{\partial \mathbf{P}(\mathbf{X})}{\partial \mathbf{X}}.$$

**Theorem 3** (Hessian of LayerNorm). *Let* $LayerNorm(\mathbf{X}) = \mathbf{P}(\mathbf{X})\mathbf{M}(\mathbf{X})$ *with Jacobian* $\frac{\partial LayerNorm}{\partial \mathbf{X}} = (\mathbf{P} \otimes \mathbf{I}_{d_V})\mathbf{G} + (\mathbf{I}_L \otimes \mathbf{M}^\top)\mathbf{H}$, *where* $\mathbf{G} = \left(\mathbf{I}_{Ld_V} - \frac{1}{d_V}(\mathbf{I}_L \otimes \mathbf{1}_{d_V \times d_V})\right)$ *is constant and* $\mathbf{H} = \frac{\partial \mathbf{P}}{\partial \mathbf{X}}$ *as in Theorem 2. The Hessian is*

$$\frac{\partial^2 LayerNorm}{\partial \mathbf{X}^2} = ((\mathbf{P}(\mathbf{X}) \otimes \mathbf{I}_{d_V}) \otimes \mathbf{I}_{Ld_V})\frac{\partial^2 \mathbf{M}}{\partial \mathbf{X}^2} + \big(\mathbf{I}_{Ld_V} \otimes \mathbf{G}^\top\big)\frac{\partial(\mathbf{P}(\mathbf{X}) \otimes \mathbf{I}_{d_V})}{\partial \mathbf{X}} +$$
$$+ \big((\mathbf{I}_L \otimes \mathbf{M}^\top) \otimes \mathbf{I}_{Ld_V}\big)\frac{\partial^2 \mathbf{P}}{\partial \mathbf{X}^2} + \big(\mathbf{I}_{Ld_V} \otimes \mathbf{H}^\top\big)\frac{\partial(\mathbf{I}_L \otimes \mathbf{M}^\top)}{\partial \mathbf{X}},$$

*where where* $\frac{\partial^2 \mathbf{M}}{\partial \mathbf{X}^2} = 0$, *and other terms as derived in the proof.*

Proofs and detailed versions for Theorems 2-3 are provided in Appendices C.2 - C.3.

Before providing calculations for the whole Transformer Block we need to introduce an activation function matrix derivative.

**Lemma 1** (ReLU derivative and Hessian). *Let* $\mathbf{X} \in \mathbb{R}^{m \times n}$, *almost everywhere the following holds:*

$$\frac{\partial \mathrm{ReLU}(\mathbf{X})}{\partial \mathbf{X}} = \mathrm{diag}\big(\mathrm{vec}_r(\mathbf{1}_{\{\mathbf{X}>0\}})\big), \quad \frac{\partial^2 \mathrm{ReLU}(\mathbf{X})}{\partial \mathbf{X}^2} = \mathbf{0}.$$

The proof is in the Appendix D.

Thus, we calculate the derivatives and the Hessian of the proposed Transformer block representation 5 with respect to a square norm Loss, where we put $\mathbf{b}_{1,2} = 0$ in FFN block for simplicity of subsequent calculations and use ReLU as an activation layer.

**Theorem 4** (Transformer block derivative). *For Transformer block from 5 with* $\mathbf{S} = ReLU(\mathbf{Y}\mathbf{W}_1)\mathbf{W}_2 + \mathbf{Y}$ *and* $\mathbf{Z} = \mathrm{LayerNorm}(\mathbf{S})$:

$$\frac{\partial \mathbf{Z}}{\partial \mathbf{W}_i} = \mathbf{J}_Z \cdot \begin{cases} \mathbf{B}_i, & i \in \{1,2\} \\ \mathbf{J}_{SY}\mathbf{G}_i, & i \in \{K,Q,V\} \end{cases}$$

*where* $\mathbf{J}_Z = \frac{\partial \mathbf{Z}}{\partial \mathbf{S}}$, $\mathbf{B}_i = \frac{\partial \mathbf{S}}{\partial \mathbf{W}_i}$, $\mathbf{J}_{SY} = \frac{\partial \mathbf{S}}{\partial \mathbf{Y}}$, $\mathbf{G}_i = \frac{\partial \mathbf{Y}}{\partial \mathbf{W}_i}$.

More detailed of the theorem and it's proof can be found in Appendix C.4.

**Theorem 5** (Hessian of the Transformer block 5). *The Hessian blocks of the Transformer output* $\mathbf{Z}$ *w.r.t. parameters* $(\mathbf{W}_i, \mathbf{W}_j)$ *are*

$$\mathbf{H}_{\mathrm{tr}}^{(i,j)} := \frac{\partial^2 \mathbf{Z}}{\partial \mathbf{W}_i \partial \mathbf{W}_j} = (\mathbf{J}_Z \otimes \mathbf{I}_{n_i}) \, \boldsymbol{\xi}_{ij} + \left( \mathbf{I}_{Ld_V} \otimes \mathbf{B}_i^\top \right) \mathbf{H}_Z \mathbf{B}_j \tag{6}$$

*with* $\boldsymbol{\xi}_{ij} := \frac{\partial}{\partial \mathbf{W}_j} \left( \frac{\partial \mathbf{S}}{\partial \mathbf{W}_i} \right), \mathbf{J}_Z := \frac{\partial \, \mathrm{LayerNorm}(\mathbf{S})}{\partial \mathbf{S}}, \mathbf{H}_Z := \frac{\partial^2 \, \mathrm{LayerNorm}(\mathbf{S})}{\partial \mathbf{S}^2}$ *and* $\mathbf{B}_i := \frac{\partial \mathbf{S}}{\partial \mathbf{W}_i}$, *where* $\mathbf{S} := ReLU(\mathbf{YW}_1)\mathbf{W}_2 + \mathbf{Y}$

More detailed version of the theorem and the proof can be found in Appendix C.5.

We note that the theorem above is responsible for the $\frac{\partial^2 f_{\mathbf{w}}}{\partial \mathbf{W}_i \partial \mathbf{W}_j}$ part from the Hessian of the Loss function decomposition 3. Therefore, the whole Transformer Hessian can be represented as:

$$\frac{\partial^2 (\mathcal{L} \circ \mathbf{Z})}{\partial \mathbf{W}_i \partial \mathbf{W}_j} = \frac{\partial \mathbf{Z}}{\partial \mathbf{W}_i}^\top \frac{\partial^2 \mathcal{L}}{\partial \mathbf{Z}^2} \frac{\partial \mathbf{Z}}{\partial \mathbf{W}_j} + \left( \frac{\partial \mathcal{L}}{\partial \mathbf{Z}} (\mathbf{Z}(\cdot)) \otimes \mathbf{I}_{p_i q_i} \right) \mathbf{H}_{\mathrm{tr}}^{(i,j)}, \tag{7}$$

where $\mathcal{L}(\cdot) = \| \cdot - \mathbf{Target} \|_2^2$, it's second derivative is $\frac{2}{Ld_V}$, and $\frac{\partial \mathcal{L}}{\partial \mathbf{Z}}(\mathbf{Z}(\cdot))$ can be calculated similarly to $\mathbf{R}_m$ from Theorem 3.2 Ormaniec et al. (2024), thus, $\mathbf{R}_m^{\mathrm{tr}} = \mathrm{vec}_r(\mathbf{Z} - \mathbf{Target})^\top \otimes \mathbf{I}_m$, while $\frac{\partial \mathbf{Z}}{\partial \mathbf{W}_i}, \frac{\partial \mathbf{Z}}{\partial \mathbf{W}_j}$ are from Theorem 4 and $\mathbf{H}_{\mathrm{tr}}^{(i,j)}$ is from Theorem 5.

Therefore the transformer-block square-norm can be estimated according to the theorem

**Theorem 6** (Spectral-norm estimate of the Transformer Hessian). *Let* $\mathbf{H}_{\mathrm{tr}}^{(i,j)}$ *denote the* $(i,j)$-*th block of the Transformer Hessian from equation 12, where* $i,j \in \{1, 2, K, Q, V\}$ *and* $n_i = \dim(\mathbf{W}_i)$. *Then, for each pair* $(i,j)$,

$$\left\| \mathbf{H}_{\mathrm{tr}}^{(i,j)} \right\|_2 \leq \| \mathbf{J}_Z \|_2 \, \| \boldsymbol{\xi}_{ij} \|_2 + \| \mathbf{B}_i \|_2 \, \| \mathbf{H}_Z \|_2 \, \| \mathbf{B}_j \|_2, \tag{8}$$

*where* $\boldsymbol{\xi}_{ij} = \frac{\partial}{\partial \mathbf{W}_j} \left( \frac{\partial \mathbf{S}}{\partial \mathbf{W}_i} \right)$ *and* $\mathbf{B}_i = \frac{\partial \mathbf{S}}{\partial \mathbf{W}_i}$.

*Explicit expressions for each bound are stated in the proof.*

*Furthermore, estimation for the whole transformer Hessian can be calculated as:*

*Let* $\mathbf{H}_{\mathrm{tr}}$ *be the full Hessian arranged as a* $m_b \times n_b$ *block-matrix with blocks* $\mathbf{H}_{\mathrm{tr}}^{(i,j)}$, *where* $m_b = n_b = 5$ *(indexed by* $\{1, 2, K, Q, V\}$*). Then*

$$\| \mathbf{H}_{\mathrm{tr}} \|_2 \leq \sqrt{m_b n_b} \, \max_{i,j} \left( \frac{2}{Ld_V} \| \frac{\partial \mathbf{Z}}{\partial \mathbf{W}_i} \|_2 \| \frac{\partial \mathbf{Z}}{\partial \mathbf{W}_j} \|_2 + \| \mathbf{R}_m^{tr} \|_2 \| \mathbf{H}_{tr}^{(i,j)} \|_2 \right). \tag{9}$$

*Since* $m_b = n_b = 5$, *we get* $\| \mathbf{H}_{\mathrm{tr}} \|_2 \leq 5 \, \max_{i,j}(\cdots)$. *We denote this estimation as* $M_{tr}$.

The proof is provided in Appendix C.6.

### 4.3 Convergence of the Loss Function Surface

Similarly to Kiselev and Grabovoy (2024) let us use second-order Taylor approximation for the mentioned above loss functions at $\mathbf{w}^*$. We suppose that decomposition to the second order will be sufficient to study local behavior. The first-order term vanishes because the gradients $\nabla \mathcal{L}_k(\mathbf{w}^*)$ and $\nabla \mathcal{L}_{k+1}(\mathbf{w}^*)$ are zero according to Assumption 1:

$$\mathcal{L}_k(\mathbf{w}) \approx \mathcal{L}_k(\mathbf{w}^*) + \frac{1}{2} (\mathbf{w} - \mathbf{w}^*)^\top \mathbf{H}^{(k)}(\mathbf{w}^*)(\mathbf{w} - \mathbf{w}^*), \tag{10}$$

where we denoted the Hessian of $\mathcal{L}_k(\mathbf{w})$ with respect to parameters $\mathbf{w}$ at $\mathbf{w}^*$ as $\mathbf{H}^{(k)}(\mathbf{w}^*)$.

Next, we consider difference of losses $|\mathcal{L}_{k+1}(\mathbf{w}) - \mathcal{L}_k(\mathbf{w})|$ while increasing the sequence length.

**Theorem 7** (Convergence of Self-Attention and Transformer Blocks). *For a single self-attention block and a single transformer block 5 under the conditions that the loss function is bounded $0 \leqslant l(\mathbf{f}_{\mathbf{w}^*}(\mathbf{x}_i), \mathbf{y}_i) \leqslant L$, and the individual Hessians are bounded, the following holds:*

$$|\mathcal{L}_{k+1}(\mathbf{w}) - \mathcal{L}_k(\mathbf{w})| \leqslant \frac{2L}{k+1} + \frac{M\|\mathbf{w} - \mathbf{w}^*\|_2^2}{(k+1)},$$

*where for the self-attention block $M$ can be directly calculated from Theorem 1 and for the transformer block $M = M_{tr}$ is calculated according to Theorem 6.*

It's worth noting that $M$ in the theorem above is not a constant in terms of increasing the sequence length $k$, as soon as $M$ as in a function of $\|\mathbf{X}\|_2$ which changes during described process. For more details see Appendix C.1 and C.6.

The proof is provided in Appendix C.7.

## 5 EXPERIMENTS

To verify our theoretical estimates we conduct a comprehensive empirical study. We follow the same Transformer architecture we used in the main part of the paper, which is essentially post-norm (LayerNorm is after Self-Attention/FeedForward).

In particular, we consider an image classification task, implementing the Vision Transformer (ViT) architecture similar to Dosovitskiy et al. (2020), see Figure 2. Input image is patchified with linear projection and then goes to Transformer Encoder, which contains $L$ Transformer Blocks, while its outputs is averaged to obtain classification logits.

**Hessian entries visualization.** In this part we use a single Transformer block, which we train on a MNIST Deng (2012) dataset (see 1). Firstly, we put just one batch from a train dataloader to the initialized model and calculate the exact Hessian using `curvlinops` Python package for an efficient Hessian linear operator calculation. Visualizing it in a log-scale, in Figure 3 we emphasize the heterogenity in the magnitues of the entries.

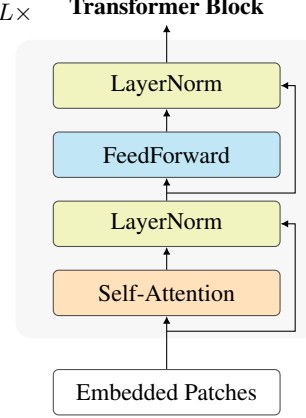

Figure 2: Transformer architecture we use in our experiments

| dataset | patch size | hidden dim | ff dim | num blocks |
|---------|-----------|-----------|--------|-----------|
| MNIST | 4 | 16 | 64 | 1 |
| CIFAR-100 | 4 | 128 | 512 | 8 |

Table 1: Vision Transformer (ViT) architectures hyperparameters we use in our experiments

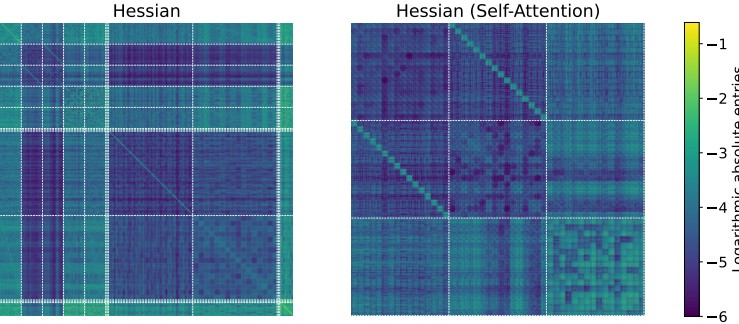

Figure 3: Hessian entries visualization for **an initialized model** with one Transformer Block. We see the entire magnitudes' heterogeneity, while the Values corresponding blocks have larger values.

We train the model for a number of epochs, obtaining pretty high accuracy on a validation dataset (>50%), and then visualize the Hessian's entries again, see Figure 4. One can see that each of the Hessian's blocks becomes more magnituded, however the Values-Values block exhibits the highest one.

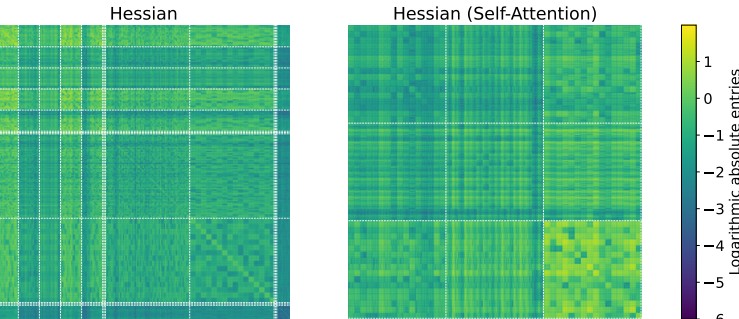

Figure 4: Hessian entries visualization for **a model trained for a number of epochs** with one Transformer Block. We see the entire magnitudes' heterogeneity, while the Values-Values corresponding block has the largest values.

This experiment shows exactly how the entire Transformer's Hessian is organized, which allows us to investigate each block part of it separately. In Appendix A.1 we continue this experiment by providing Parameters blocks changing over training epochs figures.

Further, we calculate the matrices' norms and their Hessians' norms, and show them in Figure 5

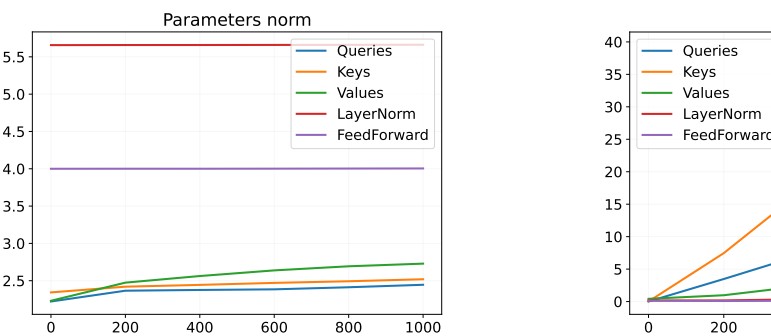

Figure 5: Parameters' blocks norms and their Hessians' norms, calculated exactly on one batch containing 128 examples from the MNIST training dataset.

Results show that the highest magnitude corresponds to the Keys and Values, while the other blocks exhibit much smaller absolute entries.

**Loss landscape convergence.** To further deep inside the dependence between loss function and its Hessian, we conduct and experiment corresponding to Theorem 7. Here we employ the other model configuration on a CIFAR-100 Krizhevsky (2009) dataset. Compared to similar one for a MNIST dataset, this model have $8\times$ more Transformer blocks and also $8\times$ wider hidden layers. During traning, it is also trained for a number of epochs to achieve >50% Accuracy on a validation dataset. The results are in Figure 6. The experiment setup is as follows:

1. Train the model until convergence and save the parameters $\mathbf{w}*$ (model checkpoint);

2. Start from the empty dataset, add data batch-by-batch and calculate mean loss value over the seen batches;

3. Calculate the absolute difference according to $|\mathcal{L}_{k+1}(\mathbf{w}) - \mathcal{L}_k(\mathbf{w})|$.

Our code is available at `https://anonymous.4open.science/r/transformer_hessians/`

## 6 DISCUSSION AND CONCLUSION

This work fills a key gap in the second-order analysis of Transformers by deriving explicit Jacobians and Hessians for LayerNorm and FFN in the $\text{vec}_r$ numerator-layout, and integrating them into a full block-level curvature decomposition. Theorems 2-3 and 4-5 yield end-to-end expressions that are compatible with Kronecker structure and commutation identities, while Theorems 1 and 6 provide spectral-norm bounds that connect curvature to input statistics, LayerNorm scales, and architectural hyperparameters. A direct consequence is a block-heterogeneous Hessian: Value- and Key-related terms dominate through softmax derivatives and input-dependent operators, FFN curvature is controlled by the piecewise linearity of ReLU, and LayerNorm contributes via per-row variance. The empirical results (e.g., Figures 3 and 4) match these predictions, with Values - Values blocks exhibiting the largest magnitudes after training.

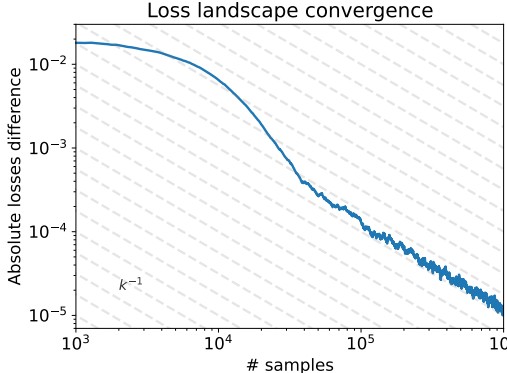

Figure 6: Absolute loss difference vs. the number of training samples in the dataset, plotted in log-log scale. The blue line represents the EMA of a desired dependency, while the gray one corresponds to the linear trend.

The second-order Taylor expansion in Theorem 7 gives a compact convergence inequality, $|\mathcal{L}_{k+1}(\mathbf{w}) - \mathcal{L}_k(\mathbf{w})| \leq 2L/(k + 1) + M\|\mathbf{w} - \mathbf{w}^*\|_2^2/(k + 1)$, where $M$ is provided by our Hessian bounds. This establishes a $1/(k + 1)$ decay of the local discrepancy between successive empirical objectives when curvature is controlled, and explains the observed stabilization of the loss landscape with increasing data. The loglog trend in Figure 6 follows this prediction, supporting the claim that increasing data size stabilizes the local geometry of the Transformer objective. Finally, the block-wise structure motivates curvature-aware training through per-block adaptation of learning rates, weight decay, or preconditioning, and provides a mechanistic rationale for switching from data scaling to model scaling near curvature stationarity, consistent with compute-optimal policies Kaplan et al. (2020); Hoffmann et al. (2022).

The analysis is local and assumes a shared minimizer for consecutive dataset sizes (Assumption 1). The present theoretical derivation focuses on a single-head, post-normalization transformer block under the mean-squared error loss. While extensions to multi-head attention, masking, and positional encodings are technically feasible within the established calculus, they are omitted for brevity. It should be emphasized that the underlying framework naturally generalizes to the cross-entropy loss, a generalization that has been explicitly validated in our experimental section. A primary direction for future work involves extending this analysis to deep, multi-layer transformer architectures.

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

# A  APPENDIX / SUPPLEMENTAL MATERIAL

## A.1  PARAMETERS BLOCKS CHANGING OVER TRAINING EPOCHS.

Here we continue the previous experiments, expanding the plots into separate parameters blocks entries changing. Again, we employ the MNIST's dataset version of our model (Figure 1). We log the matrices entries, norms, and Hessians during the first 1000 training steps. As we can see on Figures 7, 8, 9, 10, 11.

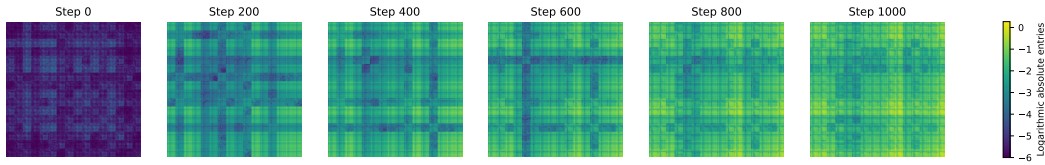

Figure 7: Queries entries visualization.

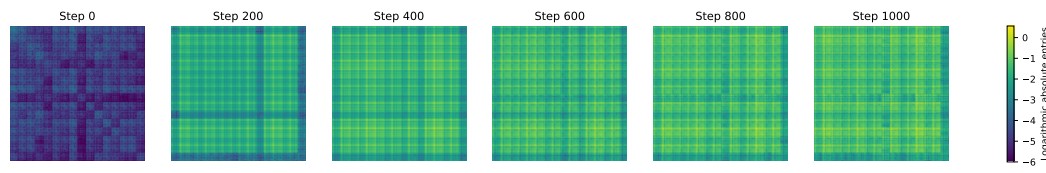

Figure 8: Keys entries visualization.

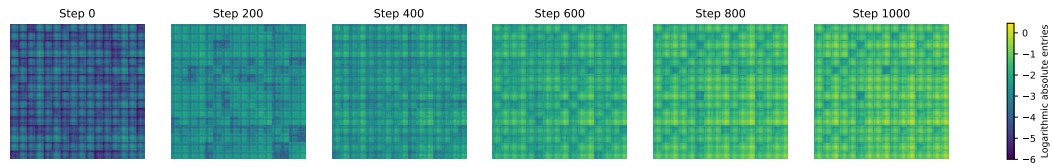

Figure 9: Values entries visualization.

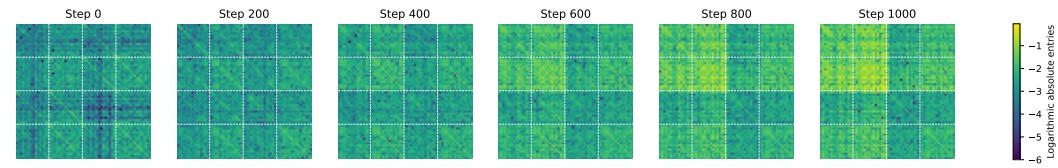

Figure 10: LayerNorm entries visualization.

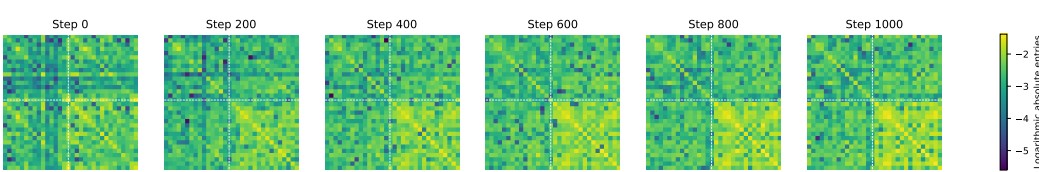

Figure 11: FeedForward entries visualization.

## A.2 ASSUMPTIONS VALIDATION

In this section we provide experimental validation of the assumptions stated in the text. Since Assumption 2 is technical, we focus on empirically validating Assumption 1.

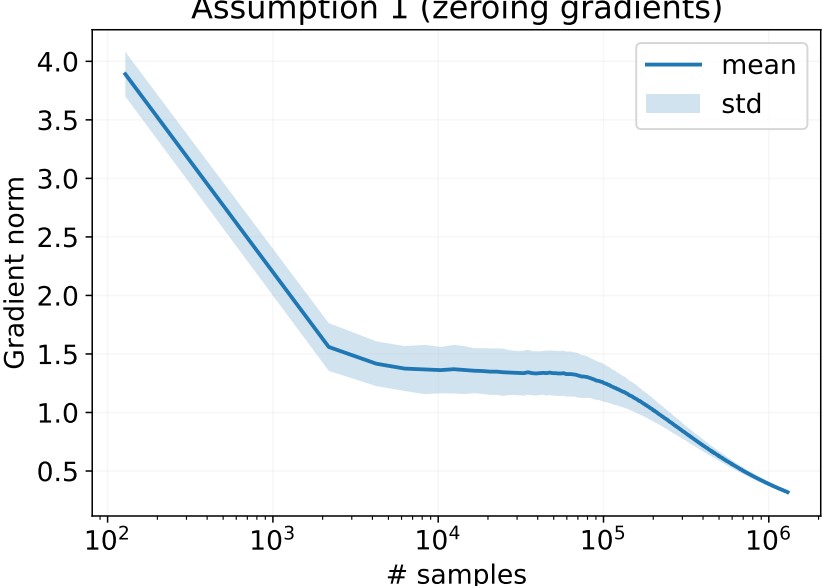

Figure 12: Validation of Assumption 1

Figure 12 presents the corresponding results, indicating that while Assumption 1 can be relaxed, its validity increases with longer sequence lengths (i.e., a larger number of samples).

## B APPENDIX / MATRIX CALCULUS PRELIMINARIES

### B.1 BASIC MATRIX OPERATIONS PROPERTIES

First, we define the notations and rules that we actively use in the text.

**Definition 1** (Matrix Norms). *For a matrix $\mathbf{A} \in \mathbb{R}^{m \times n}$:*

$$\|\mathbf{A}\|_2 = \sigma_1 \qquad \text{(Spectral norm, largest singular value)}$$

$$\|\mathbf{A}\|_F = \sqrt{\sum_{i=1}^{m}\sum_{j=1}^{n}|a_{ij}|^2} = \sqrt{\sum_{i=1}^{r}\sigma_i^2} \quad \text{(Frobenius norm)}$$

$$\|\mathbf{A}\|_1 = \max_{1 \le j \le n}\sum_{i=1}^{m}|a_{ij}| \qquad \text{(Maximum absolute column sum)}$$

$$\|\mathbf{A}\|_\infty = \max_{1 \le i \le m}\sum_{j=1}^{n}|a_{ij}| \qquad \text{(Maximum absolute row sum)}$$

$$\|\mathbf{A}\|_{\max} = \max_{i,j}|a_{ij}| \qquad \text{(Element-wise maximum, not a submultiplicative norm)}$$

**Definition 2** (Vectorization and Element-wise Operations). *Let $\mathbf{A}$ be a matrix and $\mathbf{v}$ be a vector.*

- $\mathrm{vec}_r(\mathbf{A})$ *denotes the row-wise vectorization of matrix $\mathbf{A}$.*

- $\mathbf{A}^{\circ\alpha}$ *denotes the element-wise $\alpha$-power of matrix $\mathbf{A}$, i.e., $(\mathbf{A}^{\circ\alpha})_{ij} = (\mathbf{A}_{ij})^\alpha$.*

- *diag*($\mathbf{v}$) *creates a diagonal matrix with vector* $\mathbf{v}$ *on its main diagonal.*

**Property 1** (Relation between vec and $\text{vec}_r$). *Let* $\mathbf{A} \in \mathbb{R}^{m \times n}$. *The row-wise vectorization operator* $\text{vec}_r$ *and the standard column-wise vectorization operator* vec *are related by the transpose:*

$$\text{vec}_r(\mathbf{A}) = \text{vec}(\mathbf{A}^\top)$$

**Definition 3** (Commutation Matrix). *The commutation matrix* $\mathbf{K}_{m,n} \in \mathbb{R}^{mn \times mn}$ *is the unique matrix such that for any matrix* $\mathbf{A} \in \mathbb{R}^{m \times n}$ *the following holds*

$$\mathbf{K}_{m,n} \text{vec}(\mathbf{A}) = \text{vec}(\mathbf{A}^\top)$$

*Using Property 1, we immediately have the relationship:*

$$\text{vec}_r(\mathbf{A}) = \mathbf{K}_{m,n} \text{vec}(\mathbf{A}) \quad and \quad \text{vec}(\mathbf{A}) = \mathbf{K}_{n,m} \text{vec}_r(\mathbf{A})$$

*since* $\mathbf{K}_{n,m} \mathbf{K}_{m,n} = \mathbf{I}_{mn}$.

From Magnus and Neudecker (1988) we utilize the property

**Property 2** (Row-wise vectorization of matrix product). *Let* $\mathbf{X}, \mathbf{A}, \mathbf{B}$ *be matrices with appropriate dimensions, then*

$$\text{vec}_r(\mathbf{A}\mathbf{X}\mathbf{B}) = (\mathbf{A} \otimes \mathbf{B}^\top)\text{vec}_r(\mathbf{X})$$

**Property 3** (Row-wise vectorization of Hadamard product). *Let* $\mathbf{A}, \mathbf{B} \in \mathbb{R}^{m \times n}$. *Then*

$$\text{vec}_r(\mathbf{A} \circ \mathbf{B}) = diag(\text{vec}_r(\mathbf{A}))\text{vec}_r(\mathbf{B})$$

*where* $\circ$ *denotes the Hadamard (element-wise) product. This result follows directly from Magnus and Neudecker (1988), where the similar result was obtained for column-wise vectorization.*

**Proposition 1** (Identification Theorem for Row-wise Vectorization). *Let* $\mathbf{F} : \mathbb{R}^{m \times n} \to \mathbb{R}^{p,q}$ *be a differentiable matrix-valued function of a matrix* $\mathbf{X} \in \mathbb{R}^{m \times n}$. *If the differential of* $\mathbf{F}$ *can be written as*

$$d\text{vec}_r(\mathbf{F}(\mathbf{X})) = \mathbf{J} \cdot d\text{vec}_r(\mathbf{X})$$

*for some matrix* $\mathbf{J} \in \mathbb{R}^{pq \times mn}$ *that does not depend on* $d\mathbf{X}$. *Then* $\mathbf{J}$ *is the Jacobian matrix of the transformation from* $\mathbf{X}$ *to* $\mathbf{F}(\mathbf{X})$ *with respect to row-wise vectorization. We denote this as:*

$$\frac{\partial \mathbf{F}(\mathbf{X})}{\partial \mathbf{X}} := \frac{\partial \text{vec}_r(\mathbf{F}(\mathbf{X}))}{\partial (\text{vec}_r(\mathbf{X}))^\top} = \mathbf{J}$$

*This is the* $\text{vec}_r$ *analogue of the fundamental Identification Theorem from Magnus and Neudecker (1988) for column-wise vectorization.*

**Property 4** (Element-wise division). *Let* $\mathbf{A} \in \mathbb{R}^{m \times n}$ *be a matrix and* $\mathbf{b} \in \mathbb{R}^{m \times 1}$ *be a vector. Then for matrix* $\mathbf{C} \in \in \mathbb{R}^{m \times n}$, *where* $c_{i,j} = \frac{a_{i,j}}{b_i}$ *is fulfilled that*

$$\mathbf{C} = diag^{-1}(\mathbf{b})\mathbf{A}$$

**Proposition 2** (Spectral norm of $\mathbf{1}_{L \times L}$ matrix). *Let* $\mathbf{A} = \mathbf{1}_{L \times L}$ *(a matrix full of 1). Then its spectral norm is*

$$\|\mathbf{A}\|_2 = L$$

*Proof.* Using basic Linear Algebra properties, we obtain $\text{tr}(\mathbf{A}) = L$ and $\text{rank}(\mathbf{A}) = 1 = \dim(\text{Im}(\mathbf{X}))$. Therefore, using $\dim(\text{Im}(\mathbf{X})) + \dim(\text{Ker}(\mathbf{X})) = L$, we get $\dim(\text{Ker}(\mathbf{X})) = L - 1$. Thus, for $i \in \{2, \dots L\}$ we get $\lambda_i = 0$ and for $\lambda_1 = L$. Then, the only non-null singular value of the matrix $\mathbf{A}$ is $\sqrt{L^2} = L$. Thus, we obtain that $\|\mathbf{A}\|_2 = L$, according to Definition 1. $\square$

## B.2 MATRIX-VALUED FUNCTIONS DERIVATIVE PROPERTIES

Next, we introduce the properties for calculating the matrix-valued function derivative.

**Property 5** (Matrix-Product derivative). *Let* $\mathbf{X}, \mathbf{A}, \mathbf{B}$ *be matrices with appropriate dimensions, then*

$$\frac{\partial \mathbf{A}\mathbf{X}\mathbf{B}}{\partial \mathbf{X}} = \mathbf{A} \otimes \mathbf{B}^\top$$

*where* $\mathbf{A}$ *and* $\mathbf{B}$ *have no dependence on* $\mathbf{X}$.

Detailed proof of this statement can be found in Singh et al. (2021).

**Property 6** (Kronecker-Product derivative). *Let* $\mathbf{X} \in \mathbb{R}^{n \times q}$ *and* $\mathbf{Y} \in \mathbb{R}^{p \times r}$. *Then*

$$\frac{\partial(\mathbf{X} \otimes \mathbf{Y})}{\partial \mathbf{X}} = (\mathbf{I}_n \otimes \mathbf{K}_{p,q} \otimes \mathbf{I}_r)(\mathbf{I}_{nq} \otimes \mathrm{vec}_r \mathbf{Y}),$$

*and analogously*

$$\frac{\partial(\mathbf{X} \otimes \mathbf{Y})}{\partial \mathbf{Y}} = (\mathbf{I}_n \otimes \mathbf{K}_{p,q} \otimes \mathbf{I}_r)(\mathrm{vec}_r \mathbf{X} \otimes \mathbf{I}_{pr}).$$

The detailed proof is in Ormaniec et al. (2024).

From the properties above, we derive calculations for special cases which we use in this paper.

**Proposition 3** (Matrix-valued functions multiplication derivative). *Let* $\mathbf{A}(\mathbf{X}) \in \mathbb{R}^{p \times r}$ *and* $\mathbf{B}(\mathbf{X}) \in \mathbb{R}^{r \times q}$ *be matrix-valued functions of the matrix* $\mathbf{X}$, *then*

$$\frac{\partial \mathbf{A}(\mathbf{X})\mathbf{B}(\mathbf{X})}{\partial \mathbf{X}} = (\mathbf{A} \otimes \mathbf{I}_q)\frac{\partial \mathbf{B}}{\partial \mathbf{X}} + (\mathbf{I}_p \otimes \mathbf{B}^\top)\frac{\partial \mathbf{A}}{\partial \mathbf{X}}$$

*Proof.* First, we apply a classic chain-rule for calculation a derivative of a complicated function and then combine it with Property 5

$$\frac{\partial \mathbf{A}(\mathbf{X})\mathbf{B}(\mathbf{X})}{\partial \mathbf{X}} = \frac{\partial \mathbf{A}\mathbf{B}}{\partial \mathbf{B}}\frac{\partial \mathbf{B}}{\partial \mathbf{X}} + \frac{\partial \mathbf{A}\mathbf{B}}{\partial \mathbf{A}}\frac{\partial \mathbf{A}}{\partial \mathbf{X}} = \frac{\partial \mathbf{A}\mathbf{B}\mathbf{I}_q}{\partial \mathbf{B}}\frac{\partial \mathbf{B}}{\partial \mathbf{X}} + \frac{\partial \mathbf{I}_p\mathbf{A}\mathbf{B}}{\partial \mathbf{A}}\frac{\partial \mathbf{A}}{\partial \mathbf{X}} =$$
$$= (\mathbf{A} \otimes \mathbf{I}_q)\frac{\partial \mathbf{B}}{\partial \mathbf{X}} + (\mathbf{I}_p \otimes \mathbf{B}^\top)\frac{\partial \mathbf{A}}{\partial \mathbf{X}}$$

$\square$

**Proposition 4** (Matrix-valued functions Kronecker product derivative). *Let* $\mathbf{A}(\mathbf{X}) \in \mathbb{R}^{n \times q}$ *and* $\mathbf{B}(\mathbf{X}) \in \mathbb{R}^{p \times r}$ *be matrix-valued functions of the matrix* $\mathbf{X}$, *then*

$$\frac{\partial \mathbf{A}(\mathbf{X}) \otimes \mathbf{B}(\mathbf{X})}{\partial \mathbf{X}} = (\mathbf{I}_n \otimes \mathbf{K}_{p,q} \otimes \mathbf{I}_r)\left((\mathrm{vec}_r \mathbf{A} \otimes \mathbf{I}_{pr})\frac{\partial \mathbf{B}}{\partial \mathbf{X}} + (\mathbf{I}_{nq} \otimes \mathrm{vec}_r \mathbf{B})\frac{\partial \mathbf{A}}{\partial \mathbf{X}}\right)$$

*Proof.* First, we apply a classic chain rule for calculating the derivative of a complicated function and then combine it with Property 6

$$\frac{\partial \mathbf{A}(\mathbf{X}) \otimes \mathbf{B}(\mathbf{X})}{\partial \mathbf{X}} = \frac{\partial \mathbf{A} \otimes \mathbf{B}}{\partial \mathbf{B}}\frac{\partial \mathbf{B}}{\partial \mathbf{X}} + \frac{\partial \mathbf{A} \otimes \mathbf{B}}{\partial \mathbf{A}}\frac{\partial \mathbf{A}}{\partial \mathbf{X}} =$$
$$= (\mathbf{I}_n \otimes \mathbf{K}_{p,q} \otimes \mathbf{I}_r)(\mathrm{vec}_r \mathbf{A} \otimes \mathbf{I}_{pr})\frac{\partial \mathbf{B}}{\partial \mathbf{X}} + (\mathbf{I}_n \otimes \mathbf{K}_{p,q} \otimes \mathbf{I}_r)(\mathbf{I}_{nq} \otimes \mathrm{vec}_r \mathbf{B})\frac{\partial \mathbf{A}}{\partial \mathbf{X}} =$$
$$= (\mathbf{I}_n \otimes \mathbf{K}_{p,q} \otimes \mathbf{I}_r)\left((\mathrm{vec}_r \mathbf{A} \otimes \mathbf{I}_{pr})\frac{\partial \mathbf{B}}{\partial \mathbf{X}} + (\mathbf{I}_{nq} \otimes \mathrm{vec}_r \mathbf{B})\frac{\partial \mathbf{A}}{\partial \mathbf{X}}\right)$$

$\square$

Next, we develop the operations that we introduced above and derive calculations using $\mathrm{vec}_r$ notation as we do in this paper.

**Proposition 5** (Derivative of the invert matrix). *For an invertible square matrix $\mathbf{D} \in \mathbb{R}^{n \times n}$, the derivative of its inverse is*

$$\frac{\partial \mathbf{D}^{-1}}{\partial \mathbf{D}} = -\mathbf{D}^{-1} \otimes \mathbf{D}^{-\top}.$$

*Proof.* This is a standard result in matrix calculus. The differential identity

$$d(\mathbf{D}^{-1}) = -\mathbf{D}^{-1} (d\mathbf{D}) \mathbf{D}^{-1}$$

appears in Petersen and Pedersen (2012) and in Magnus and Neudecker (1988). Applying the $\mathrm{vec}_r$ operator and using the property 2 yields

$$\mathrm{vec}_r(-\mathbf{D}^{-1} (d\mathbf{D}) \mathbf{D}^{-1}) = (-\mathbf{D}^{-1} \otimes \mathbf{D}^{-\top})\mathrm{vec}_r(d\mathbf{D})$$

By the definition and the identification theorem from Property 1 we obtain

$$\mathrm{vec}_r(d\mathbf{D}^{-1}) = \frac{\partial \mathrm{vec}_r \mathbf{D}^{-1}}{\partial \mathrm{vec}_r \mathbf{D}} \mathrm{vec}_r(d\mathbf{D})$$

Comparing two results we get $\frac{\partial \mathrm{vec}_r \mathbf{D}^{-1}}{\partial \mathrm{vec}_r \mathbf{D}} = (-\mathbf{D}^{-1} \otimes \mathbf{D}^{-\top})$

$\square$

**Proposition 6** (Derivative of $\mathrm{diag}(\cdot)$). *For $\mathbf{v} \in \mathbb{R}^{L \times 1}$, the derivative of the diagonalization map is*

$$\frac{\partial \mathrm{diag}(\mathbf{v})}{\partial \mathbf{v}} = \begin{pmatrix} \mathbf{e}_1 \otimes \mathbf{e}_1 & \ldots & \mathbf{e}_L \otimes \mathbf{e}_L \end{pmatrix},$$

*where $\mathbf{e}_i$ are the standard basis vectors in $\mathbb{R}^L$.*

*Proof.* By Definition 2, $\mathrm{diag}(\mathbf{v})$ places entry $v_i$ at position $(i, i)$ of the resulting diagonal matrix.

The derivative of $\mathrm{diag}(\mathbf{v})$ w.r.t. $v_i$ is the elementary matrix $\mathbf{E}_{ii} = \mathbf{e}_i \mathbf{e}_i^\top$ that has one in position $(i, i)$ and zeros elsewhere.

Applying the row-wise vectorization operator, we obtain

$$\mathrm{vec}_r(\mathbf{E}_{i,i}) = \mathbf{e}_i \otimes \mathbf{e}_i$$

by the standard Kroneckervec identity 2.

Stacking across $i = 1, \ldots, L$, the Jacobian becomes

$$\frac{\partial \mathrm{diag}(\mathbf{v})}{\partial \mathbf{v}} = \begin{pmatrix} \mathbf{e}_1 \otimes \mathbf{e}_1 & \ldots & \mathbf{e}_L \otimes \mathbf{e}_L \end{pmatrix},$$

$\square$

**Proposition 7** (Derivative of the Hadamard square). *For a matrix $\mathbf{A} \in \mathbb{R}^{m \times n}$, the derivative of the elementwise square is*

$$\frac{\partial \mathbf{A}^{\circ 2}}{\partial \mathbf{A}} = 2 \cdot \mathrm{diag}\big(\mathrm{vec}_r(\mathbf{A})\big).$$

*Proof.* By Definition 2, $(\mathbf{A}^{\circ 2})ij = (\mathbf{A}ij)^2$. Differentiating elementwise gives $d(\mathbf{A}^{\circ 2}) = 2\mathbf{A} \circ d\mathbf{A}$. Applying the $\mathrm{vec}_r$ operator and using Property 3, we obtain

$$\mathrm{vec}_r(d(\mathbf{A}^{\circ 2})) = 2 diag(\mathrm{vec}_r(\mathbf{A}))\mathrm{vec}_r(d\mathbf{A})$$

By the identification theorem from Property 1, this implies

$$\frac{\partial \mathbf{A}^{\circ 2}}{\partial \mathbf{A}} = \frac{\partial \mathrm{vec}_r(\mathbf{A}^{\circ 2})}{\partial \mathrm{vec}_r(\mathbf{A})} = 2 \cdot \mathrm{diag}\big(\mathrm{vec}_r(\mathbf{A})\big)$$

which establishes the result. $\square$

**Proposition 8** (Derivative of the Hadamard root). *For $\mathbf{A} \in \mathbb{R}^{m \times n}$ with positive entries, the derivative of the elementwise square root is*

$$\frac{\partial \mathbf{A}^{\circ \frac{1}{2}}}{\partial \mathbf{A}} = \tfrac{1}{2} \operatorname{diag}^{-1}\big(\operatorname{vec}_r^{\circ \frac{1}{2}}(\mathbf{A})\big).$$

*Proof.* Similarly to the proof of Proposition 7, we obtain $d(\mathbf{A}^{\circ 1/2}) = \tfrac{1}{2}\mathbf{A}^{\circ -1/2} \circ d\mathbf{A}$ Thus, writing in vectorized form gives

$$\frac{\partial \mathbf{A}^{\circ \frac{1}{2}}}{\partial \mathbf{A}} = \frac{\partial \operatorname{vec}_r(\mathbf{A}^{\circ \frac{1}{2}})}{\partial \operatorname{vec}_r(\mathbf{A})} = \tfrac{1}{2} \operatorname{diag}^{-1}\big(\operatorname{vec}_r^{\circ \frac{1}{2}}(\mathbf{A})\big).$$

$\square$

**Proposition 9** (Transposed Matrix derivative). *Let $\mathbf{A} \in \mathbb{R}^{m \times n}$, then the following holds:*

$$\frac{\partial \mathbf{A}^{\top}}{\partial \mathbf{A}} = \mathbf{K}_{n,m}$$

*Proof.* Combining a similar property from Magnus and Neudecker (1988) for column-wise vectorization with the column-row connection rule 1 and 3 we obtain the theorem statement. $\square$

### B.3 MATRIX NORM PROPERTIES

Similarly to Petersen and Pedersen (2012) we introduce a matrix norms table comparison.

**Property 7** (Matrix norm inequalities). *Let $\mathbf{A} \in \mathbb{R}^{m \times n}$. Then the following inequalities hold between different matrix norms:*

| $X$ \ $Y$ | $\|\mathbf{A}\|_{\max}$ | $\|\mathbf{A}\|_1$ | $\|\mathbf{A}\|_{\infty}$ | $\|\mathbf{A}\|_2$ | $\|\mathbf{A}\|_F$ |
|---|---|---|---|---|---|
| $\|\mathbf{A}\|_{\max}$ | | *1* | *1* | *1* | *1* |
| $\|\mathbf{A}\|_1$ | $m$ | | $m$ | $\sqrt{m}$ | $\sqrt{m}$ |
| $\|\mathbf{A}\|_{\infty}$ | $n$ | $n$ | | $\sqrt{n}$ | $\sqrt{n}$ |
| $\|\mathbf{A}\|_2$ | $\sqrt{mn}$ | $\sqrt{n}$ | $\sqrt{m}$ | | *1* |
| $\|\mathbf{A}\|_F$ | $\sqrt{mn}$ | $\sqrt{n}$ | $\sqrt{m}$ | $\sqrt{d}$ | |

*where $d = \operatorname{rank}(\mathbf{A})$. The table should be read as: for any two norms $\|\cdot\|_X$ and $\|\cdot\|_Y$,*

$$\|\mathbf{A}\|_X \leq c \cdot \|\mathbf{A}\|_Y$$

*where $c$ is the constant found at the intersection of row $X$ and column $Y$.*

**Property 8** (Matrix sum norm). *Let $\mathbf{A}$ and $\mathbf{B}$ be matrices from $\mathbb{R}^{m \times n}$, then*

$$\|\mathbf{A} + \mathbf{B}\|_2 \leq \|\mathbf{A}\|_2 + \|\mathbf{B}\|_2 \tag{11}$$

**Property 9** (Kronecker product norm). *Let $\mathbf{A} \in \mathbb{R}^{m \times n}$ and $\mathbf{B} \in \mathbb{R}^{p \times q}$, then the following holds*

$$\|\mathbf{A} \otimes \mathbf{B}\|_2 = \|\mathbf{A}\|_2 \|\mathbf{B}\|_2$$

**Property 10** (Matrix product norm). *Let $\mathbf{A} \in \mathbb{R}^{m \times n}$ and $\mathbf{B} \in \mathbb{R}^{n \times q}$, then the following holds*

$$\|\mathbf{A}\mathbf{B}\|_2 \leq \|\mathbf{A}\|_2 \|\mathbf{B}\|_2$$

The properties above can be found in Magnus and Neudecker (1988).

**Property 11** (Block-matrix norm inequality). *Let* $\mathbf{A} \in \mathbb{R}^{m \times n}$ *be a block-matrix, each block of which is a matrix* $\mathbf{B}_{i,j}$, *thus the following holds*

$$\|\mathbf{A}\|_2 \le \sqrt{mn} \max_{i,j} \|\mathbf{B}_{i,j}\|_2$$

*Note, if matrix* $\mathbf{A}$ *is block-diagonal, then the strict equality holds* $\|\mathbf{A}\|_2 = \max_i \|\mathbf{B}_{i,i}\|_2$.

**Property 12** (Transposed matrix norm). *Let* $\mathbf{A} \in \mathbb{R}^{m \times n}$, *then*

$$\|\mathbf{A}\|_2 = \|\mathbf{A}^\top\|_2$$

## C   APPENDIX / PROOFS OF THE THEOREMS

### C.1   PROOF OF THEOREM 1

*Proof.* From Lemma A.3 Noci et al. (2022) and using Properties 10 and 9

$$\|\frac{\partial \mathbf{A}}{\partial \mathbf{T}}\|_2 = \frac{1}{L}\|\mathbf{I}_L\|_2\|\mathbf{I}_L - \frac{1}{L}\mathbf{1}_{L \times L}\|_2 \le \frac{1}{L}$$

Here we used that $\frac{1}{L}\mathbf{1}_{L \times L}$ is a projection matrix, therefore $\mathbf{I}_L - \frac{1}{L}\mathbf{1}_{L \times L}$ is a projection matrix and it's norm is $\|\mathbf{I}_L - \frac{1}{L}\mathbf{1}_{L \times L}\|_2 \le 1$.

Next we estimate the $\mathbf{Z}_1$ norm, utilizing the same Properties 10 and 9

$$\|\mathbf{Z}_1\|_2 \le \|\mathbf{I}_L \otimes \mathbf{X}^\top\|_2 \|\frac{\partial \mathbf{A}}{\partial \mathbf{T}}\|_2 \|\mathbf{X} \otimes \mathbf{X}\|_2 \le \|\mathbf{X}\|_2 \frac{1}{L}\|\mathbf{X}\|_2^2 = \frac{1}{L}\|\mathbf{X}\|_2^3$$

where we used Property 12 for $\|\mathbf{X}\|_2 = \|\mathbf{X}^\top\|_2$.

Now we calculate estimations for the outer-product Hessian part.

But before that we estimate $\|\mathbf{A}\|_2$. This block itself is a row-wise softmax matrix. Thus, each element $\mathbf{A}_{i,j} \le 1$. Next we use Property 7 and obtain $\|\mathbf{A}\|_{\max} \le \|\mathbf{A}\|_2 \le \sqrt{LL}\|\mathbf{A}\|_{\max} = L\|\mathbf{A}\|_{max} \le L$. Therefore, the $\|\mathbf{M}_1\|_2 = \|\mathbf{A}\mathbf{X}\|_2 \le L\|\mathbf{X}\|_2$.

Thus, the $\|\mathbf{H}_o(\mathbf{W}_i, \mathbf{W}_j)\|_2$ is estimated below:

$$\|\mathbf{H}_o(\mathbf{W}_V, \mathbf{W}_V)\|_2 \le \frac{2}{Ld_V}\|\mathbf{M}_1\|_2^2 1 \le \frac{2}{Ld_V}\|\mathbf{A}\|_2^2\|\mathbf{X}\|_2^2 \le \frac{2}{Ld_V}L^2\|\mathbf{X}\|_2^2 = \frac{2L}{d_V}\|\mathbf{X}\|_2^2$$

$$\begin{aligned}
\|\mathbf{H}_o(\mathbf{W}_Q, \mathbf{W}_Q)\|_2 &\le \|\frac{2}{Ld_Vd_K}(\mathbf{I}_{d_V} \otimes \mathbf{W}_K^\top)\mathbf{Z}_1^\top(\mathbf{I}_L \otimes \mathbf{W}_V\mathbf{W}_V^\top)\,\mathbf{Z}_1(\mathbf{I}_{d_V} \otimes \mathbf{W}_K)\|_2 \\
&\le \frac{2}{Ld_Vd_K}\|\mathbf{W}_K\|_2^2\|\mathbf{Z}_1\|_2^2\|\mathbf{W}_V\|_2^2 \le \frac{2}{Ld_Vd_K}\|\mathbf{W}_K\|_2^2\|\mathbf{W}_V\|_2^2\frac{1}{L^2}\|\mathbf{X}\|_2^6 = \\
&= \frac{2}{L^3d_Vd_K}\|\mathbf{W}_K\|_2^2\|\mathbf{W}_V\|_2^2\mathbf{X}\|_2^6
\end{aligned}$$

$$\begin{aligned}
\|\mathbf{H}_o(\mathbf{W}_V, \mathbf{W}_Q)\|_2 &\le \frac{2}{Ld_V\sqrt{d_K}}\|\mathbf{M}_1^\top \otimes \mathbf{W}_V^\top\|_2\|\mathbf{Z}_1\|_2\|\mathbf{I}_{d_V} \otimes \mathbf{W}_K\|_2 \\
&\le \frac{2}{Ld_V\sqrt{d_K}}L\|\mathbf{X}\|_2\|\mathbf{W}_V\|_2\frac{1}{L}\|\mathbf{X}\|_2^3\|\mathbf{W}_K\|_2 \\
&= \frac{2}{Ld_V\sqrt{d_K}}\|\mathbf{W}_V\|_2\|\mathbf{W}_K\|_2\|\mathbf{X}\|_2^4
\end{aligned}$$

$$\|\mathbf{H}_o(\mathbf{W}_Q, \mathbf{W}_K)\|_2 \le \frac{2}{Ld_V d_K}\|(\mathbf{I}_{d_V} \otimes \mathbf{W}_K^\top)\mathbf{Z}_1^\top(\mathbf{I}_L \otimes \mathbf{W}_V \mathbf{W}_V^\top)\mathbf{Z}_1(\mathbf{W}_Q \otimes \mathbf{I}_{d_V})\mathbf{K}d_K, d_V\|_2$$

$$\le \frac{2}{L^3 d_V d_K}\|\mathbf{W}_K\|_2\|\mathbf{W}_Q\|_2\|\mathbf{W}_V\|_2^2\|\mathbf{X}\|_2^6$$

where we use Properties 10, 9 and $\|\mathbf{K}_{d_V d_K}\|_2 = 1$, because $\mathbf{K}_{m,n}$ is a commutation matrix from Definition 3.

Next we derive functional-part estimation. First we provide analysis for $\mathbf{R}_m = \mathrm{vec}_r(\mathbf{F}(\mathbf{X}) - \mathbf{Target})^T \otimes \mathbf{I}_m$ from Theorem 3.2 from Ormaniec et al. (2024). Since $\mathrm{vec}_r(\cdot)$ is a vectorization procedure $\|\mathrm{vec}_r(\mathbf{F}(\mathbf{X}) - \mathbf{Target})\|_2 = \|\mathbf{F}(\mathbf{X}) - \mathbf{Target}\|_F \le \sqrt{\mathrm{rank}(\mathbf{F}(\mathbf{X}) - \mathbf{Target})}\|\mathbf{F}(\mathbf{X}) - \mathbf{Target}\|_2$ according to Property 7. Therefore, we obtain

$$\|\mathbf{R}_m\| \le \sqrt{\mathrm{rank}(\mathbf{F}(\mathbf{X}) - \mathbf{Target})}\|\mathbf{F}(\mathbf{X}) - \mathbf{Target}\|_2 \le \sqrt{\mathrm{rank}(\mathbf{F}(\mathbf{X}) - \mathbf{Target})}(\|\mathbf{A}\|_2\|\mathbf{X}\|_2\|\mathbf{W}_V\|_2 + \|\mathbf{Target}\|_2)$$

$$\le \sqrt{\mathrm{rank}(\mathbf{F}(\mathbf{X}) - \mathbf{Target})}(L\|\mathbf{X}\|_2\|\mathbf{W}_V\|_2 + \|\mathbf{Target}\|_2)$$

where we used Properties 10, 8

Next we estimate the shuffling matrix norm, utilizing standard properties

$$\|\mathbf{S}\|_2 = \|(\mathbf{I}_{d_V} \otimes \mathbf{K}_{d_V, d_V})(\mathrm{vec}_r(\mathbf{I}_{d_V}) \otimes \mathbf{I}_{d_V})\|_2 \le \|\mathrm{vec}_r(\mathbf{I}_{d_V})\|_2 = \|\mathbf{I}_{d_V}\|_F = \sqrt{d_V}$$

Next challenging part is computing bounds for $\|\frac{\partial^2 \mathbf{A}}{\partial \mathbf{T}^2}\|_2$. In Lemma C1 from Ormaniec et al. (2024) the a block form of this expression is provided:

$$\frac{\partial^2 \mathbf{A}_{i,j}}{\partial \mathbf{T}_{i,:}\partial \mathbf{T}_{i,:}} = \mathbf{A}_{i,j}\left(2\mathbf{A}_{i,:}\mathbf{A}_{i,:}^\top + \mathbf{E}_{j,j}^{L,L} - \mathrm{diag}(\mathbf{A}_{i,:}) - \mathbf{e}_j\mathbf{A}_{i,:}^\top - \mathbf{A}_{i,:}\mathbf{e}_j^\top\right) \in \mathbb{R}^{L \times L},$$

where $\mathbf{E}_{j,j}^{L,L} = \mathbf{e}_j\mathbf{e}_j^\top \in \mathbb{R}^{L \times L}$ therefore it contains only one non-zero element that equals 1 in $(j,j)$ position. Additionally, it's explicitly said that the second derivative of the row-wise softmax has a block-diagonal structure. Thus, we use block matrix Property 11: $\left\|\frac{\partial^2 \mathbf{A}}{\partial \mathbf{T}^2}\right\|_2 = \max_{i,j}\left\|\frac{\partial^2 \mathbf{A}_{i,j}}{\partial \mathbf{T}_{i,:}\partial \mathbf{T}_{i,:}}\right\|_2$.

Thus, we conduct $\|\frac{\partial^2 \mathbf{A}_{i,j}}{\partial \mathbf{T}_{i,:}\partial \mathbf{T}_{i,:}}\|_2$ estimation. As we stated before $\mathbf{A}_{i,j} \le 1$. Now $\|\mathbf{A}_{i,:}\mathbf{A}_{i,:}^\top\|_2$: as soon as $\mathbf{A}_{i,:}$ is a row in a softmax matrix, values in it sum up to 1. Thus, we can use the vector-matrix inequalities to obtain: $\|\mathbf{A}_{i,:}\mathbf{A}_{i,:}^\top\|_2 \le \|\mathbf{A}_{i,:}\|_2^2 \le \|\mathbf{A}_{i,:}\|_1^2 = 1$. After that we conduct $\|\mathbf{E}_{j,j}^{m,n}\|_2 = \|\mathbf{e}_j\mathbf{e}_j^\top\|_2 \le 1$. Then we estimate $\|\mathrm{diag}(\mathbf{A}_{i,:})\|_2$. For diagonal matrices we can easily obtain that $\|\mathrm{diag}(\mathbf{A}_{i,:})\|_2 = \max_j \mathbf{A}_{i,j} \le 1$. Next we estimate $\mathbf{e}_j\mathbf{A}_{i,:}^\top$ and $\mathbf{A}_{i,:}\mathbf{e}_j^\top$ norms: the matrices $\mathbf{e}_j\mathbf{A}_{i,:}^\top$ and $\mathbf{A}_{i,:}\mathbf{e}_j^\top$ are rank-1 matrices with only one non-zero row and one non-zero column respectively, containing elements of $\mathbf{A}_{i,:}$. Their spectral norms can be estimated $\|\mathbf{A}_{i,:}\|_2 \le 1$.

Therefore, we provide an estimation:

$$\|\frac{\partial^2 \mathbf{A}}{\partial \mathbf{T}^2}\|_2 \le 6$$

In this way we can easily obtain the $\|\mathbf{Z}_2\|_2$ estimation

$$\|\mathbf{Z}_2\|_2 = \|\left(\mathbf{I}_L \otimes \mathbf{X}^\top \otimes \mathbf{X}^\top \otimes \mathbf{X}^\top\right)\left(\partial^2 \mathbf{A}/\partial \mathbf{T}^2\right)(\mathbf{X} \otimes \mathbf{X})\|_2 \le \|\mathbf{X}\|_2^5\|\frac{\partial^2 \mathbf{A}}{\partial \mathbf{T}^2}\|_2 \le 6\|\mathbf{X}\|_2^5$$

After that, we proceed to the estimation of the functional Hessian norms.

$$\|\mathbf{H}_f(\mathbf{W}_V, \mathbf{W}_V)\|_2 = 0$$

$$\|\mathbf{H}_f(\mathbf{W}_Q, \mathbf{W}_Q)\|_2 = \frac{2}{Ld_V d_K}\|\mathbf{R}_{d_V d_K}\left(\mathbf{I}_L \otimes \mathbf{W}_V^\top \otimes \mathbf{I}_{d_V} \otimes \mathbf{W}_K^\top\right)\mathbf{Z}_2\left(\mathbf{I}_{d_V} \otimes \mathbf{W}_K\right)\|_2,$$

$$\le \frac{2}{Ld_V d_K}\|\mathbf{R}_{d_V d_K}\|_2\|\mathbf{W}_V\|_2\|\mathbf{W}_K\|_2\|\mathbf{Z}_2\|_2\|\mathbf{W}_K\|_2$$

$$\le \frac{2}{Ld_V d_K}6\sqrt{\mathrm{rank}(\mathbf{F}(\mathbf{X}) - \mathbf{Target})}(L\|\mathbf{X}\|_2\|\mathbf{W}_V\|_2 + \|\mathbf{Target}\|_2)\|\mathbf{W}_V\|_2\|\mathbf{W}_K\|_2^2\|\mathbf{X}\|_2^5 =$$

$$= \frac{12}{d_V d_K}\sqrt{\mathrm{rank}(\mathbf{F}(\mathbf{X}) - \mathbf{Target})}(L\|\mathbf{X}\|_2\|\mathbf{W}_V\|_2 + \|\mathbf{Target}\|_2)\|\mathbf{W}_V\|_2\|\mathbf{W}_K\|_2^2\|\mathbf{X}\|_2^5$$

$$\|\mathbf{H}_{\mathrm{f}}(\mathbf{W}_V, \mathbf{W}_Q)\|_2 = \frac{2}{Ld_V\sqrt{d_K}}\|\mathbf{R}_{d_V^2}\left(\mathbf{I}_L \otimes \mathbf{S}\right)\mathbf{Z}_1\left(\mathbf{I}_{d_V} \otimes \mathbf{W}_K\right)\|_2 \leq$$

$$\leq \frac{2}{Ld_V\sqrt{d_K}}\|\mathbf{R}_{d_V^2}\|_2\|\mathbf{S}\|_2\|\mathbf{Z}_1\|_2\|\mathbf{W}_K\|_2 \leq$$

$$\leq \frac{2}{Ld_V\sqrt{d_K}}\sqrt{\operatorname{rank}(\mathbf{F}(\mathbf{X}) - \mathbf{Target})}(L\|\mathbf{X}\|_2\|\mathbf{W}_V\|_2 + \|\mathbf{Target}\|_2)\sqrt{d_V}\frac{1}{L}\|\mathbf{X}\|_2^3\|\mathbf{W}_K\|_2 =$$

$$= \frac{2\sqrt{\operatorname{rank}(\mathbf{F}(\mathbf{X}) - \mathbf{Target})}}{L^2\sqrt{d_V d_K}}(L\|\mathbf{X}\|_2\|\mathbf{W}_V\|_2 + \|\mathbf{Target}\|_2)\|\mathbf{W}_K\|_2\|\mathbf{X}\|_2^3$$

$$\|\mathbf{H}_{\mathrm{f}}(\mathbf{W}_Q, \mathbf{W}_K)\| \leq \frac{2}{Ld_V d_K}\|\mathbf{R}_{d_V d_K}\left(\mathbf{I}_L \otimes \mathbf{W}_V^\top \otimes \mathbf{I}_{d_V} \otimes \mathbf{W}_K^\top\right)\mathbf{Z}_2\left(\mathbf{W}_Q \otimes \mathbf{I}_{d_V}\right)\mathbf{K}_{d_K, d_V}\|_2 +$$

$$+ \frac{2}{Ld_V\sqrt{d_K}}\|\mathbf{R}_{d_V}\left(\mathbf{I}_L \otimes \mathbf{W}_V^\top \otimes \mathbf{I}_{d_V}\right)\left(\mathbf{Z}_1 \otimes \mathbf{I}_{d_V}\right)\mathbf{S} \otimes \mathbf{I}_{d_K}\|_2 \leq$$

$$\leq \frac{2}{Ld_V d_K}\sqrt{\operatorname{rank}(\mathbf{F}(\mathbf{X}) - \mathbf{Target})}(L\|\mathbf{X}\|_2\|\mathbf{W}_V\|_2 + \|\mathbf{Target}\|_2)\|\mathbf{W}_V\|_2\|\mathbf{W}_K\|_2\|\mathbf{W}_Q\|_2 6\|\mathbf{X}\|_2^5 +$$

$$+ \frac{2}{Ld_V\sqrt{d_K}}\sqrt{\operatorname{rank}(\mathbf{F}(\mathbf{X}) - \mathbf{Target})}(L\|\mathbf{X}\|_2\|\mathbf{W}_V\|_2 + \|\mathbf{Target}\|_2)\|\mathbf{W}_V\|_2\frac{1}{L}\|\mathbf{X}\|_2^3\sqrt{d_V} =$$

$$= \frac{2\sqrt{\operatorname{rank}(\mathbf{F}(\mathbf{X}) - \mathbf{Target})}(L\|\mathbf{X}\|_2\|\mathbf{W}_V\|_2 + \|\mathbf{Target}\|_2)}{Ld_V\sqrt{d_V d_K}}\|\mathbf{W}_V\|_2 \cdot$$

$$\cdot \left(3L\|\mathbf{W}_K\|_2\|\mathbf{W}_Q\|_2\|\mathbf{X}\|_2^5 + \frac{d_V}{L}\|\mathbf{X}\|_2^3\right),$$

Therefore we can obtain the final hessian estimation according to Property 7, where we used number of block equal to 3 from $\{K, Q, V\}$:

$$\|\mathbf{H}(\mathbf{W}_i, \mathbf{W}_j)\|_2 \leq 3\max_{i,j\in\{Q,K,V\}}\left(\|\mathbf{H}_o(\mathbf{W}_i, \mathbf{W}_j)\|_2 + \|\mathbf{H}_f(\mathbf{W}_i, \mathbf{W}_j)\|_2\right)$$

And now after substituting results :

$$\|\mathbf{H}(\mathbf{W}_i, \mathbf{W}_j)\|_2 \leq$$

$$\leq 3\max\left(\frac{2L}{d_V}\|\mathbf{X}\|_2^2,\right.$$

$$\frac{2}{L^3 d_V d_K}\|\mathbf{W}_K\|_2^2\|\mathbf{W}_V\|_2^2\|\mathbf{X}\|_2^6 + \frac{12}{d_V d_K}\sqrt{\operatorname{rank}(\mathbf{F}(\mathbf{X}) - \mathbf{Target})}(L\|\mathbf{X}\|_2\|\mathbf{W}_V\|_2 + \|\mathbf{Target}\|_2)\|\mathbf{W}_V\|_2\|\mathbf{W}_K\|_2^2\|\mathbf{X}\|_2^5,$$

$$\frac{2}{Ld_V\sqrt{d_K}}\|\mathbf{W}_V\|_2\|\mathbf{W}_K\|_2\|\mathbf{X}\|_2^4 + \frac{2\sqrt{\operatorname{rank}(\mathbf{F}(\mathbf{X}) - \mathbf{Target})}}{L^2\sqrt{d_V d_K}}(L\|\mathbf{X}\|_2\|\mathbf{W}_V\|_2 + \|\mathbf{Target}\|_2)\|\mathbf{W}_K\|_2\|\mathbf{X}\|_2^3,$$

$$\frac{2}{L^3 d_V d_K}\|\mathbf{W}_K\|_2\|\mathbf{W}_Q\|_2\|\mathbf{W}_V\|_2^2\|\mathbf{X}\|_2^6 +$$

$$\left. + \frac{2\sqrt{\operatorname{rank}(\mathbf{F}(\mathbf{X}) - \mathbf{Target})}(L\|\mathbf{X}\|_2\|\mathbf{W}_V\|_2 + \|\mathbf{Target}\|_2)}{Ld_V\sqrt{d_V d_K}}\|\mathbf{W}_V\|_2\left(3L\|\mathbf{W}_K\|_2\|\mathbf{W}_Q\|_2\|\mathbf{X}\|_2^5 + \frac{d_V}{L}\|\mathbf{X}\|_2^3\right)\right)$$

The obtained expression we denote as $M$. The obtained inequalities can be simplified by $\operatorname{rank}(\mathbf{F}(\mathbf{X}) - \mathbf{Target}) \leq \min(L, d_V)$. That ends the proof. $\qquad\square$

## C.2 Proof of Theorem 2

**Theorem 8** (Detailed version of Theorem 2)**.** *Let* $\mathbf{X} \in \mathbb{R}^{L \times d_V}$. *Define*

$$\mathbf{M}(\mathbf{X}) = \mathbf{X} - \frac{1}{d_V}\mathbf{X}\mathbf{1}_{d_V}\mathbf{1}_{d_V}^\top, \quad \sigma(\mathbf{X}) = \frac{1}{\sqrt{d_V}}\left(\mathbf{M}(\mathbf{X})^{\circ 2}\mathbf{1}_{d_V}\right)^{\circ 1/2}, \quad \mathbf{P}(\mathbf{X}) = \operatorname{diag}^{-1}(\sigma(\mathbf{X})).$$

*Then the Jacobian of*

$$LayerNorm(\mathbf{X}) = \mathbf{P}(\mathbf{X})\mathbf{M}(\mathbf{X})$$

*with respect to* $\mathbf{X}$ *is*

$$\frac{\partial\, LayerNorm(\mathbf{X})}{\partial \mathbf{X}} = (\mathbf{P}(\mathbf{X}) \otimes \mathbf{I}_{d_V})\left(\mathbf{I}_{Ld_V} - \frac{1}{d_V}(\mathbf{I}_L \otimes \mathbf{1}_{d_V \times d_V})\right) + (\mathbf{I}_L \otimes \mathbf{M}(\mathbf{X})^\top)\frac{\partial \mathbf{P}(\mathbf{X})}{\partial \mathbf{X}}.$$

*Moreover,*

$$\frac{\partial \mathbf{P}}{\partial \mathbf{X}} = \frac{1}{\sqrt{d_V}}\left(-\mathbf{D}^{-1}{\otimes}\mathbf{D}^{-\top}\right)\left(\mathbf{e}_1{\otimes}\mathbf{e}_1, \dots, \mathbf{e}_L{\otimes}\mathbf{e}_L\right)\left(\mathrm{diag}^{-1}(\mathrm{vec}_r^{1/2}(\mathbf{M}^{\circ 2}\mathbf{1}_{d_V}))(\mathbf{I}_L{\otimes}\mathbf{1}_{d_V}^\top)\mathrm{diag}(\mathrm{vec}_r(\mathbf{M}))\frac{\partial \mathbf{M}}{\partial \mathbf{X}}\right),$$

*with* $\mathbf{D} = \mathrm{diag}(\sigma(\mathbf{X}))$.

*Proof.* We represent LayerNorm layer as

$$LayerNorm(\mathbf{X}) = \mathbf{P}(\mathbf{X})\mathbf{M}(\mathbf{X})$$

where $\mathbf{P}(\mathbf{X}) = \mathbf{D}^{-1}$, where $\mathbf{D} = diag(\sigma(\mathbf{X}))$ and $\mathbf{M}(\mathbf{X}) = (\mathbf{X} - \mu(\mathbf{X})\mathbf{1}_{d_V}^\top)$ according to Property 4.

Using the matrix-product derivative rule from Property 3 we obtain:

$$\frac{\partial LayerNorm(\mathbf{X})}{\partial \mathbf{X}} = (\mathbf{P}(\mathbf{X}) \otimes \mathbf{I}_{d_V})\frac{\partial \mathbf{M}}{\partial \mathbf{X}} + (\mathbf{I}_L \otimes \mathbf{M}^\top)\frac{\partial \mathbf{P}}{\partial \mathbf{X}}$$

Let's start with $\frac{\partial \mathbf{M}}{\partial \mathbf{X}}$. Using simple matrix calculus properties we can obtain $\mathbf{M}(\mathbf{X}) = (\mathbf{X} - \mu(\mathbf{X})\mathbf{1}_{d_V}^\top) = (\mathbf{X} - \frac{1}{d_V}\mathbf{X}\mathbf{1}_{d_V}\mathbf{1}_{d_V}^\top) = (\mathbf{X} - \frac{1}{d_V}\mathbf{X}\mathbf{1}_{d_V \times d_V})$. Thus, the derivative is

$$\frac{\partial \mathbf{M}}{\partial \mathbf{X}} = \frac{\partial(\mathbf{X} - \frac{1}{d_V}\mathbf{X}\mathbf{1}_{d_V \times d_V})}{\partial \mathbf{X}} = (\mathbf{I}_L \otimes \mathbf{I}_{d_V}) - \frac{1}{d_V}(\mathbf{I}_L \otimes \mathbf{1}_{d_V \times d_V})$$

Next, we calculate the $\frac{\partial \mathbf{P}}{\partial \mathbf{X}}$. First, we start with the transformation of $\sigma(\mathbf{X})$ expression. We can rewrite it in the matrix terms $\sigma(\mathbf{X}) = (\frac{1}{d_V}(\mathbf{X} - \mu(X)\mathbf{1}_{d_V}^\top)^{\circ 2}\mathbf{1}_{d_V})^{\circ\frac{1}{2}} = \frac{1}{\sqrt{d_V}}\left(\mathbf{M}(\mathbf{X})^{\circ 2}\mathbf{1}_{d_V}\right)^{\circ\frac{1}{2}}$. Here, $\circ\alpha$ operation is element-wise $\alpha$-powering from Definition 2.

Therefore, we can apply chain rule and get

$$\frac{\partial \mathbf{P}}{\partial \mathbf{X}} = \frac{\partial \mathbf{D}^{-1}}{\partial \mathbf{D}}\frac{\partial diag(\sigma(\mathbf{X}))}{\partial \sigma(\mathbf{X})}\frac{\partial \sigma(\mathbf{X})}{\partial \mathbf{X}}$$

Therefore, by utilizing Properties 7, 8 and 5 we can find

$$\frac{\partial \sigma(\mathbf{X})}{\partial \mathbf{X}} = \frac{1}{\sqrt{d_V}}\frac{\partial \tau^{\circ\frac{1}{2}}}{\partial \tau}\frac{\partial \tau}{\partial \mathbf{Q}}\frac{\partial \mathbf{Q}}{\partial \mathbf{X}},$$

Here $\tau = \mathbf{Q}\cdot\mathbf{1}_L$ and $\mathbf{Q} = \mathbf{M}^{\circ 2}$. Thus, we can continue calculations and obtain

$$\frac{\partial \sigma(\mathbf{X})}{\partial \mathbf{X}} = \frac{1}{\sqrt{d_V}}\frac{\partial \tau^{\circ\frac{1}{2}}}{\partial \tau}\frac{\partial \mathbf{Q}\cdot\mathbf{1}_{d_V}}{\partial \mathbf{Q}}\frac{\partial \mathbf{M}^{\circ 2}}{\partial \mathbf{M}}\frac{\partial \mathbf{M}}{\partial \mathbf{X}} =$$

$$= \frac{1}{\sqrt{d_V}}\frac{1}{2}diag^{-1}(\mathrm{vec}_r^{\circ\frac{1}{2}}(\tau))(\mathbf{I}_L \otimes \mathbf{1}_{d_V}^T)2\cdot diag(\mathrm{vec}_r(\mathbf{M}))\frac{\partial \mathbf{M}}{\partial \mathbf{X}} =$$

$$= \frac{1}{\sqrt{d_V}}diag^{-1}(\mathrm{vec}_r^{\circ\frac{1}{2}}(\mathbf{M}^{\circ 2}\cdot\mathbf{1}_{d_V}))\cdot(\mathbf{I}_L \otimes \mathbf{1}_{d_V}^T)\cdot diag(\mathrm{vec}_r(\mathbf{M}))\frac{\partial \mathbf{M}}{\partial \mathbf{X}}$$

Therefore, by applying 5 and 6 for the first and second multiplier, we obtain

$$\frac{\partial \mathbf{P}}{\partial \mathbf{X}} = \frac{1}{\sqrt{d_V}} \left( -\mathbf{D}^{-1} \otimes \mathbf{D}^{-\top} \right) \begin{pmatrix} \mathbf{e}_1 \otimes \mathbf{e}_1 & \ldots & \mathbf{e}_L \otimes \mathbf{e}_L \end{pmatrix} \cdot$$

$$\left( diag^{-1}(\mathrm{vec}_r^{\circ \frac{1}{2}}(\mathbf{M}^{\circ 2} \cdot \mathbf{1}_{d_V})) \cdot (\mathbf{I}_L \otimes \mathbf{1}_{d_V}^T) \cdot diag(\mathrm{vec}_r(\mathbf{M})) \frac{\partial \mathbf{M}}{\partial \mathbf{X}} \right)$$

Therefore, we found the first derivative of the LayerNorm function:

$$\frac{\partial \mathrm{LayerNorm}(\mathbf{X})}{\partial \mathbf{X}} = (\mathbf{P}(\mathbf{X}) \otimes \mathbf{I}_{d_V}) \frac{\partial \mathbf{M}}{\partial \mathbf{X}} + (\mathbf{I}_L \otimes \mathbf{M}^\top) \frac{\partial \mathbf{P}}{\partial \mathbf{X}} =$$

$$= (\mathbf{P}(\mathbf{X}) \otimes \mathbf{I}_{d_V}) \frac{\partial \mathbf{M}}{\partial \mathbf{X}} +$$

$$+ (\mathbf{I}_L \otimes \mathbf{M}^\top) \frac{1}{\sqrt{d_V}} \left( -\mathbf{D}^{-1} \otimes \mathbf{D}^{-\top} \right) \begin{pmatrix} \mathbf{e}_1 \otimes \mathbf{e}_1 & \ldots & \mathbf{e}_L \otimes \mathbf{e}_L \end{pmatrix} \cdot$$

$$\cdot \left( diag^{-1}(\mathrm{vec}_r^{\circ \frac{1}{2}}(\mathbf{M}^{\circ 2} \cdot \mathbf{1}_{d_V})) \cdot (\mathbf{I}_L \otimes \mathbf{1}_{d_V}^T) \cdot diag(\mathrm{vec}_r(\mathbf{M}) \frac{\partial \mathbf{M}}{\partial \mathbf{X}} \right)$$

where $\mathbf{M}(\mathbf{X}) = (\mathbf{X} - \frac{1}{d_V}\mathbf{X}\mathbf{1}_{d_V \times d_V})$, $\mathbf{P}(\mathbf{X}) = diag^{-1}(\sigma(\mathbf{X}))$ and $\frac{\partial \mathbf{M}}{\partial \mathbf{X}} = (\mathbf{I}_L \otimes \mathbf{I}_{d_V}) - \frac{1}{d_V}(\mathbf{I}_L \otimes \mathbf{1}_{d_V \times d_V})$

That ends the proof.

$\square$

### C.3 PROOF OF THEOREM 3

*Proof.* Now, we calculate the second derivative $\frac{\partial^2 \mathrm{LayerNorm}}{\partial \mathbf{X}^2}$. Using the matrix product derivative property 5, we obtain:

$$\frac{\partial^2 \mathrm{LayerNorm}}{\partial \mathbf{X}^2} = ((\mathbf{P}(\mathbf{X}) \otimes \mathbf{I}_{d_V}) \otimes \mathbf{I}_{L d_V}) \frac{\partial^2 \mathbf{M}}{\partial \mathbf{X}^2} + \left( \mathbf{I}_{L d_V} \otimes (\frac{\partial \mathbf{M}}{\partial \mathbf{X}})^\top \right) \frac{\partial (\mathbf{P}(\mathbf{X}) \otimes \mathbf{I}_{d_V})}{\partial \mathbf{X}} +$$

$$+ ((\mathbf{I}_L \otimes \mathbf{M}^\top) \otimes \mathbf{I}_{L d_V}) \frac{\partial^2 \mathbf{P}}{\partial \mathbf{X}^2} + \left( \mathbf{I}_{L d_V} \otimes (\frac{\partial \mathbf{P}}{\partial \mathbf{X}})^\top \right) \frac{\partial (\mathbf{I}_L \otimes \mathbf{M}^\top)}{\partial \mathbf{X}}$$

Here, we have $\mathbf{P} \in \mathbb{R}^{L \times L}, \mathbf{M} \in \mathbb{R}^{L \times d_V}, \frac{\partial \mathbf{M}}{\partial \mathbf{X}} \in \mathbb{R}^{L d_V \times L d_V}, \frac{\partial \mathbf{P}}{\partial \mathbf{X}} \in \mathbb{R}^{L^2 \times L d_V}$

Next, we can easily obtain, using Properties 6, 9:

$$\frac{\partial^2 \mathbf{M}}{\partial \mathbf{X}^2} = 0$$

$$\frac{\partial (\mathbf{P}(\mathbf{X}) \otimes \mathbf{I}_{d_V})}{\partial \mathbf{X}} = \frac{\partial (\mathbf{P} \otimes \mathbf{I}_L)}{\partial \mathbf{P}} \frac{\partial \mathbf{P}}{\partial \mathbf{X}} = (\mathbf{I}_L \otimes \mathbf{K}_{L,L} \otimes \mathbf{I}_L) (\mathbf{I}_{L^2} \otimes \mathrm{vec}_r(\mathbf{I}_L)) \frac{\partial \mathbf{P}}{\partial \mathbf{X}}$$

$$\frac{\partial (\mathbf{I}_L \otimes \mathbf{M}^\top)}{\partial \mathbf{X}} = \frac{\partial (\mathbf{I}_L \otimes \mathbf{M}^\top)}{\partial \mathbf{M}^\top} \frac{\partial \mathbf{M}^\top}{\partial \mathbf{M}} \frac{\partial \mathbf{M}}{\partial \mathbf{X}} = (\mathbf{I}_L \otimes \mathbf{K}_{d_V,L} \otimes \mathbf{I}_L) (\mathrm{vec}_r(\mathbf{I}_L) \otimes \mathbf{I}_{L d_V}) \mathbf{K}_{d_V,L} \frac{\partial \mathbf{M}}{\partial \mathbf{X}}$$

Now, we analyze the second-order derivative of the $\mathbf{P}$ matrix. To derive correct calculations we need to write the dimensions of each multiplier in the calculated first derivative out. Matrix $\mathbf{D}$ is a $diag(\sigma(\mathbf{X}))$, the size of vector $\sigma(\mathbf{X})$ is $L \times 1$, therefore, $\mathbf{D} \in \mathbb{R}^{L \times L}$ and the part $\left( -\mathbf{D}^{-1} \otimes \mathbf{D}^{-\top} \right) \in \mathbb{R}^{L^2 \times L^2}$. Next, we note that the size of each basis vector $\mathbf{e}_i$ is $L \times 1$, thus we obtain $\mathbf{e}_i \otimes \mathbf{e}_i \in \mathbb{R}^{L^2 \times 1}$ and $\begin{pmatrix} \mathbf{e}_1 \otimes \mathbf{e}_1 & \ldots & \mathbf{e}_L \otimes \mathbf{e}_L \end{pmatrix} \in \mathbb{R}^{L^2 \times L}$. As we discussed earlier, $\mathbf{M}(\mathbf{X}) \in \mathbb{R}^{L \times d_V}$, then $M \cdot \mathbf{1}_{d_V} \in \mathbb{R}^{L \times 1}$, and we can derive the size of $diag^{-1}(\mathrm{vec}_r^{\circ \frac{1}{2}}(\mathbf{M}^{\circ 2} \cdot \mathbf{1}_{d_V}))$, which is $L \times L$. Next multipliers are $(\mathbf{I}_L \otimes \mathbf{1}_{d_V}^T) \in \mathbb{R}^{L \times L d_V}$ and $diag(\mathrm{vec}_r(M)) \in \mathbb{R}^{L d_V \times L d_V}$. The last one is $\frac{\partial \mathbf{M}}{\partial \mathbf{X}}$, which we have already calculated, it's size is $L d_V \times L d_V$. Therefore, the whole derivative $\frac{\partial \mathbf{P}}{\partial \mathbf{X}}$ is from $\mathbb{R}^{L^2 \times L d_V}$.

We start with $\frac{\partial \mathbf{P}}{\partial \mathbf{X}} = \frac{1}{\sqrt{d_V}} \mathbf{A}_1(\mathbf{X}) \cdot \mathbf{B}_1(\mathbf{X})$, where $\mathbf{A}_1 = \left(-\mathbf{D}^{-1} \otimes \mathbf{D}^{-\top}\right)$ and $\mathbf{B}_1$ is the other multiplier.

Therefore, using Property 3 we obtain

$$\frac{\partial^2 \mathbf{P}}{\partial \mathbf{X}^2} = \frac{1}{\sqrt{d_V}} \frac{\partial \mathbf{A}_1(\mathbf{X}) \cdot \mathbf{B}_1(\mathbf{X})}{\partial \mathbf{X}} = \frac{1}{\sqrt{d_V}} \left(\mathbf{A}_1 \otimes \mathbf{I}_{Ld_V}\right) \frac{\partial \mathbf{B}_1}{\partial \mathbf{X}} + \left(\mathbf{I}_{L^2} \otimes \mathbf{B}_1^\top\right) \frac{\partial \mathbf{A}_1}{\partial \mathbf{X}}$$

Now we focus on calculating $\frac{\partial \mathbf{A}_1}{\partial \mathbf{X}}$ on the current step. Utilising the rule 4 we can simply get:

$$\frac{\partial \mathbf{A}_1}{\partial \mathbf{X}} = \frac{\partial \left(-\mathbf{D}^{-1} \otimes \mathbf{D}^{-\top}\right)}{\partial \mathbf{X}} = (\mathbf{I}_L \otimes \mathbf{K}_{L,L} \otimes \mathbf{I}_L)\Big(\left(\mathbf{I}_{L^2} \otimes \mathrm{vec}_r(\mathbf{D}^{-\top})\right) \cdot \frac{\partial - \mathbf{D}^{-1}}{\partial \mathbf{X}} +$$
$$+ \left(\mathrm{vec}_r(-\mathbf{D}^{-1}) \otimes \mathbf{I}_{L^2}\right) \cdot \frac{\partial \mathbf{D}^{-\top}}{\partial \mathbf{X}}\Big)$$

By using the transposed matrix and the invert matrix derivative properties 9, 5, we obtain: $\frac{\partial - \mathbf{D}^{-1}}{\partial \mathbf{X}} = \frac{\partial - \mathbf{D}^{-1}}{\partial \mathbf{D}} \frac{\partial \mathbf{D}}{\partial \mathbf{X}} = \left(\mathbf{D}^{-1} \otimes \mathbf{D}^{-\top}\right) \frac{\partial \mathbf{D}}{\partial \mathbf{X}}$ and $\frac{\partial \mathbf{D}^{-\top}}{\partial \mathbf{X}} = \frac{\partial \mathbf{D}^{-\top}}{\partial \mathbf{D}^{-1}} \frac{\partial \mathbf{D}^{-1}}{\partial \mathbf{D}} \frac{\partial \mathbf{D}}{\partial \mathbf{X}} = \mathbf{K}_{L,L} \left(-\mathbf{D}^{-1} \otimes \mathbf{D}^{-\top}\right) \frac{\partial \mathbf{D}}{\partial \mathbf{X}}$, where we the $\frac{\partial \mathbf{D}}{\partial \mathbf{X}}$ as we calculated earlier, while computing the first LayerNorm's derivative is $\frac{\partial \mathbf{D}}{\partial \mathbf{X}} = \left(\mathbf{e}_1 \otimes \mathbf{e}_1 \quad \dots \quad \mathbf{e}_L \otimes \mathbf{e}_L\right) \left(diag^{-1}(\mathrm{vec}_r^{\circ \frac{1}{2}}(\mathbf{M}^{\circ 2} \cdot \mathbf{1}_{d_V})) \cdot (\mathbf{I}_L \otimes \mathbf{1}_{d_V}^T) \cdot diag(\mathrm{vec}_r(\mathbf{M})) \frac{\partial \mathbf{M}}{\partial \mathbf{X}}\right)$

And now we proceed to the calculations of the remaining part derivative.

We first assign new $\mathbf{A}_2$ and $\mathbf{B}_2$ for clear calculations. We have $\mathbf{B}_1 = \left(\mathbf{e}_1 \otimes \mathbf{e}_1 \quad \dots \quad \mathbf{e}_L \otimes \mathbf{e}_L\right) \left(diag^{-1}(\mathrm{vec}_r^{\circ \frac{1}{2}}(\mathbf{M}^{\circ 2} \cdot \mathbf{1}_{d_V})) \cdot (\mathbf{I}_L \otimes \mathbf{1}_{d_V}^T) \cdot diag(\mathrm{vec}_r(\mathbf{M})) \frac{\partial \mathbf{M}}{\partial \mathbf{X}}\right)$ and we assign new $\mathbf{A}_2$ and new $\mathbf{B}_2$ as $\mathbf{A}_2 = diag^{-1}(\mathrm{vec}_r^{\circ \frac{1}{2}}(\mathbf{M}^{\circ 2} \cdot \mathbf{1}_{d_V}))$, $\mathbf{B}_2 = (\mathbf{I}_L \otimes \mathbf{1}_{d_V}^T) \cdot diag(\mathrm{vec}_r(\mathbf{M})) \frac{\partial \mathbf{M}}{\partial \mathbf{X}}$ and we denote $\mathbf{E} = \left(\mathbf{e}_1 \otimes \mathbf{e}_1 \quad \dots \quad \mathbf{e}_L \otimes \mathbf{e}_L\right)$. Thus, $\mathbf{B}_1 = \mathbf{E}\mathbf{A}_2\mathbf{B}_2$

While $\mathbf{E}$ is a constant matrix we can apply the simplified matrix product derivative rule 3 and obtain

$$\frac{\partial \mathbf{B}_1}{\partial \mathbf{X}} = \frac{\partial \mathbf{E}\mathbf{A}_2\mathbf{B}_2}{\partial (\mathbf{A}_2\mathbf{B}_2)} \frac{\partial \mathbf{A}_2\mathbf{B}_2}{\partial \mathbf{X}} = (\mathbf{E} \otimes \mathbf{I}_{Ld_V}) \frac{\partial \mathbf{A}_2\mathbf{B}_2}{\partial \mathbf{X}}$$
$$= (\mathbf{E} \otimes \mathbf{I}_{Ld_V}) \left((\mathbf{A}_2 \otimes \mathbf{I}_{Ld_V}) \frac{\partial \mathbf{B}_2}{\partial \mathbf{X}} + (\mathbf{I}_L \otimes \mathbf{B}_2^\top) \frac{\partial \mathbf{A}_2}{\partial \mathbf{X}}\right)$$

Now, we introduce the last $\mathbf{A}_3$ and $\mathbf{B}_3$ assignment. We represent $\mathbf{B}_2$ as $\mathbf{B}_2 = \mathbf{J}\mathbf{A}_3\mathbf{B}_3$, where $\mathbf{J} = (\mathbf{I}_L \otimes \mathbf{1}_{d_V}^T)$, $\mathbf{A}_3 = diag(\mathrm{vec}_r(\mathbf{M}))$ and $\mathbf{B}_3 = \frac{\partial \mathbf{M}}{\partial \mathbf{X}}$.

Similarly to the previous step we firstly apply simplified matrix product derivative rule 3 and get

$$\frac{\partial \mathbf{B}_2}{\partial \mathbf{X}} = \frac{\partial \mathbf{J}\mathbf{A}_3\mathbf{B}_3}{\partial (\mathbf{A}_3\mathbf{B}_3)} \frac{\partial \mathbf{A}_3\mathbf{B}_3}{\partial \mathbf{X}} = (\mathbf{J} \otimes \mathbf{I}_{Ld_V}) \frac{\partial \mathbf{A}_3\mathbf{B}_3}{\partial \mathbf{X}}$$
$$= (\mathbf{J} \otimes \mathbf{I}_{Ld_V}) \left((\mathbf{A}_3 \otimes \mathbf{I}_{Ld_V}) \frac{\partial \mathbf{B}_3}{\partial \mathbf{X}} + (\mathbf{I}_{Ld_V} \otimes \mathbf{B}_3^\top) \frac{\partial \mathbf{A}_3}{\partial \mathbf{X}}\right)$$

Where both Jacobian matrices can be found easily $\frac{\partial \mathbf{A}_3}{\partial \mathbf{X}} = \frac{\partial diag(\mathrm{vec}_r(\mathbf{M}))}{\partial \mathbf{X}} = \frac{\partial diag(\mathbf{v})}{\partial (\mathbf{v})} \frac{\partial \mathrm{vec}_r(\mathbf{M})}{\partial \mathbf{M}} \frac{\partial \mathbf{M}}{\partial \mathbf{X}}$

Where we have already calculated $\frac{\partial diag(\mathbf{v})}{\partial (\mathbf{v})} = \left(\mathbf{e}_1 \otimes \mathbf{e}_1 \quad \dots \quad \mathbf{e}_L \otimes \mathbf{e}_L\right)$ according to the property 6, here $\mathbf{e}_i \in \mathbb{R}^{Ld_V \times 1}$, additionally $\frac{\partial \mathrm{vec}_r(\mathbf{M})}{\partial \mathbf{M}}$ is simply $\mathbf{I}_{Ld_V}$. As for $\frac{\partial \mathbf{B}_3}{\partial \mathbf{X}}$ for current $\mathbf{B}$ it is $\frac{\partial \mathbf{B}_3}{\partial \mathbf{X}} = \frac{\partial^2 \mathbf{M}}{\partial \mathbf{X}^2} = 0$

The last step in our analysis is putting every part of our calculations together. In our notation we can simplify the expression

$$\frac{\partial^2 \mathbf{P}}{\partial \mathbf{X}^2} = \frac{1}{\sqrt{d_V}} \left(\mathbf{A}_1 \otimes \mathbf{I}_{Ld_V}\right) \frac{\partial \mathbf{B}_1}{\partial \mathbf{X}} + \left(\mathbf{I}_{L^2} \otimes \mathbf{B}_1^\top\right) \frac{\partial \mathbf{A}_1}{\partial \mathbf{X}}$$

where $\frac{\partial \mathbf{B}_1}{\partial \mathbf{X}}$, $\frac{\partial \mathbf{A}_1}{\partial \mathbf{X}} \mathbf{B}_1$ and it's definitions $\mathbf{A}_1$, $\mathbf{B}_1$ are given above.

The last step in the proof is simply combining all together and substituting all calculated derivatives into the LayerNorm's Hessian.

That ends the proof. $\qquad\square$

### C.4 PROOF OF THEOREM 4

**Theorem 9** (More detailed version of Theorem 4). *The Transformer block is defined in 5*

*The derivative $\frac{\partial \mathbf{Z}}{\partial \mathbf{W}_i}$ is as follows.*

*For $i \in \{1, 2\}$:*

$$\frac{\partial \mathbf{Z}}{\partial \mathbf{W}_i} = \frac{\partial LayerNorm(FFN(\mathbf{Y}) + \mathbf{Y})}{\partial (FFN(\mathbf{Y}) + \mathbf{Y})} \frac{\partial (FFN(\mathbf{Y}) + \mathbf{Y})}{\partial \mathbf{W}_i},$$

*where*

$$\frac{\partial (FFN(\mathbf{Y}) + \mathbf{Y})}{\partial \mathbf{W}_i} = \begin{cases} \left(\mathbf{I}_L \otimes \mathbf{W}_2^\top\right) \text{diag}\big(\text{vec}_r(\mathbf{1}_{\{\mathbf{X}>0\}})\big) \left(\mathbf{Y} \otimes \mathbf{I}_{d_{ff}}\right), & \text{for } i = 1 \\ \sigma(\mathbf{Y}\mathbf{W}_1) \otimes \mathbf{I}_{d_V}, & \text{for } i = 2 \end{cases},$$

*and $\frac{\partial LayerNorm(FFN(\mathbf{Y})+\mathbf{Y})}{\partial (FFN(\mathbf{Y})+\mathbf{Y})}$ can be calculated following Theorem 2 and is explicitly given in the proof*

*For $i \in \{K, Q, V\}$:*

$$\frac{\partial \mathbf{Z}}{\partial \mathbf{W}_i} = \frac{\partial LayerNorm(FFN(\mathbf{Y}) + \mathbf{Y})}{\partial (FFN(\mathbf{Y}) + \mathbf{Y})} \frac{\partial (FFN(\mathbf{Y}) + \mathbf{Y})}{\partial \mathbf{Y}} \frac{\partial \mathbf{Y}}{\partial \mathbf{W}_i},$$

*where*

$$\frac{\partial (FFN(\mathbf{Y}) + \mathbf{Y})}{\partial \mathbf{Y}} = \left(\mathbf{I}_L \otimes \mathbf{W}_2^\top\right) \text{diag}\big(\text{vec}_r(\mathbf{1}_{\{\mathbf{X}>0\}})\big) \left(\mathbf{I}_L \otimes \mathbf{W}_1^\top\right) + \left(\mathbf{I}_L \otimes \mathbf{I}_{d_V}\right),$$

*and $\frac{\partial \mathbf{Y}}{\partial \mathbf{W}_i} = \frac{\partial LayerNorm(\mathbf{F}(\mathbf{X})+\mathbf{X})}{\partial (\mathbf{F}(\mathbf{X})+\mathbf{X})} \frac{\partial \mathbf{F}(\mathbf{X})}{\partial \mathbf{W}_i}$, with $\frac{\partial \mathbf{F}(\mathbf{X})}{\partial \mathbf{W}_i}$ is calculated according to Lemma A.2 from Noci et al. (2022) and $\frac{\partial LayerNorm(\mathbf{F}(\mathbf{X})+\mathbf{X})}{\partial (\mathbf{F}(\mathbf{X})+\mathbf{X})}$ is calculated according to Theorem 2.*

*Proof.* It's worth noting that in our notation $\mathbf{X} \in R^{L \times d_V}, \mathbf{Y} \in R^{L \times d_V}, \mathbf{W}_1 \in R^{d_V \times d_{ff}}, \text{ReLU}(\mathbf{Y}\mathbf{W_1}) \in R^{L \times d_{ff}}, \mathbf{W}_2 \in R^{d_{ff} \times d_V}$.

We consider the Transformer block as it's defined in 5, explicitly:

$$\mathbf{Y} = \text{LayerNorm}(\mathbf{F}(\mathbf{X}) + \mathbf{X}),$$
$$\mathbf{Z} = \text{LayerNorm}(FFN(\mathbf{Y}) + \mathbf{Y}),$$

We derive calculations for the first derivative of the whole transformer block $\frac{\partial \mathbf{Z}}{\partial \mathbf{W}_i}$.

For $i \in \{1, 2\}$:

$$\frac{\partial \mathbf{Z}}{\partial \mathbf{W}_i} = \frac{\partial \text{LayerNorm}(FFN(\mathbf{Y}) + \mathbf{Y})}{\partial (FFN(\mathbf{Y}) + \mathbf{Y})} \frac{\partial (FFN(\mathbf{Y}) + \mathbf{Y})}{\partial \mathbf{W}_i}$$

where

$$\frac{\partial (FFN(\mathbf{Y}) + \mathbf{Y})}{\partial \mathbf{W}_i} = \frac{\partial (FFN(\mathbf{Y}))}{\partial \mathbf{W}_i} = \frac{\partial \mathbf{I}_L \sigma(\mathbf{Y}\mathbf{W}_1) \mathbf{W}_2 \mathbf{I}_{d_V}}{\partial \mathbf{W}_i}$$

Therefore, using Property 5:

$$\text{for } i = 2 : \frac{\partial \mathbf{I}_L \sigma(\mathbf{Y}\mathbf{W}_1)\mathbf{W}_2\mathbf{I}_{d_V}}{\partial \mathbf{W}_i} = \sigma(\mathbf{Y}\mathbf{W}_1) \otimes \mathbf{I}_{d_V}$$

$$\text{for } i = 1 : \frac{\partial \mathbf{I}_L \sigma(\mathbf{Y}\mathbf{W}_1)\mathbf{W}_2\mathbf{I}_{d_V}}{\partial \mathbf{W}_i} = \frac{\partial \sigma(\mathbf{Y}\mathbf{W}_1)\mathbf{W}_2}{\partial \sigma(\mathbf{Y}\mathbf{W}_1)} \frac{\partial \sigma(\mathbf{Y}\mathbf{W}_1)}{\partial \mathbf{Y}\mathbf{W}_1} \frac{\partial \mathbf{Y}\mathbf{W}_1}{\partial \mathbf{W}_1}$$

$$= \left(\mathbf{I}_L \otimes \mathbf{W}_2^\top\right) \frac{\partial \sigma(\mathbf{Y}\mathbf{W}_1)}{\partial \mathbf{Y}\mathbf{W}_1} \left(\mathbf{I}_L \otimes \mathbf{W}_1^\top\right)$$

According to Lemma 1, we obtain

$$\text{for } i = 1 : \frac{\partial \mathbf{I}_L \sigma(\mathbf{Y}\mathbf{W}_1)\mathbf{W}_2\mathbf{I}_{d_V}}{\partial \mathbf{W}_i} = \left(\mathbf{I}_L \otimes \mathbf{W}_2^\top\right) \text{diag}\left(\text{vec}_r(\mathbf{1}_{\{\mathbf{X}>0\}})\right) \left(\mathbf{Y} \otimes \mathbf{I}_{d_{ff}}\right)$$

Thus for $i \in \{1, 2\}$ the following holds:

$$\frac{\partial(\text{FFN}(\mathbf{Y}) + \mathbf{Y})}{\partial \mathbf{W}_i} = \begin{cases} \left(\mathbf{I}_L \otimes \mathbf{W}_2^\top\right) \text{diag}\left(\text{vec}_r(\mathbf{1}_{\{\mathbf{X}>0\}})\right) \left(\mathbf{Y} \otimes \mathbf{I}_{d_{ff}}\right), \text{for } i = 1 \\ \sigma(\mathbf{Y}\mathbf{W}_1) \otimes \mathbf{I}_{d_V}, \text{for } i = 2 \end{cases}$$

and the whole Transformer block derivative can be calculated as:

$$\frac{\partial \mathbf{Z}}{\partial \mathbf{W}_i} = \begin{cases} \frac{\partial \text{LayerNorm}(\text{FFN}(\mathbf{Y})+\mathbf{Y})}{\partial(\text{FFN}(\mathbf{Y})+\mathbf{Y})} \left(\mathbf{I}_L \otimes \mathbf{W}_2^\top\right) \text{diag}\left(\text{vec}_r(\mathbf{1}_{\{\mathbf{X}>0\}})\right) \left(\mathbf{Y} \otimes \mathbf{I}_{d_{ff}}\right), \text{for } i = 1 \\ \frac{\partial \text{LayerNorm}(\text{FFN}(\mathbf{Y})+\mathbf{Y})}{\partial(\text{FFN}(\mathbf{Y})+\mathbf{Y})} \sigma(\mathbf{Y}\mathbf{W}_1) \otimes \mathbf{I}_{d_V}, \text{for } i = 2 \end{cases}$$

where according to Theorem 2

$$\frac{\partial \text{LayerNorm}(\text{FFN}(\mathbf{Y}) + \mathbf{Y})}{\partial(\text{FFN}(\mathbf{Y}) + \mathbf{Y})} = \left(\mathbf{P}(\text{FFN}(\mathbf{Y}) + \mathbf{Y}) \otimes \mathbf{I}_{d_V}\right) \frac{\partial \mathbf{M}}{\partial(\text{FFN}(\mathbf{Y}) + \mathbf{Y})} +$$

$$+ \left(\mathbf{I}_L \otimes \mathbf{M}^\top\right) \frac{1}{\sqrt{d_V}} \left(-\mathbf{D}^{-1} \otimes \mathbf{D}^{-\top}\right) \left(\mathbf{e}_1 \otimes \mathbf{e}_1 \quad \dots \quad \mathbf{e}_L \otimes \mathbf{e}_L\right) \cdot$$

$$\cdot \left(diag^{-1}(\text{vec}_r^{\circ\frac{1}{2}}(\mathbf{M}^{\circ 2} \cdot \mathbf{1}_{d_V})) \cdot (\mathbf{I}_L \otimes \mathbf{1}_{d_V}^T) \cdot diag(\text{vec}_r(\mathbf{M})) \frac{\partial \mathbf{M}}{\partial(\text{FFN}(\mathbf{Y}) + \mathbf{Y})}\right)$$

where $\mathbf{M}(\text{FFN}(\mathbf{Y}) + \mathbf{Y}) = ((\text{FFN}(\mathbf{Y}) + \mathbf{Y}) - \frac{1}{d_V}(\text{FFN}(\mathbf{Y}) + \mathbf{Y})\mathbf{1}_{d_V \times d_V})$, $\mathbf{P}((\text{FFN}(\mathbf{Y}) + \mathbf{Y})) = diag^{-1}(\sigma(\text{FFN}(\mathbf{Y}) + \mathbf{Y}))$ and $\frac{\partial \mathbf{M}}{\partial(\text{FFN}(\mathbf{Y})+\mathbf{Y})} = (\mathbf{I}_L \otimes \mathbf{I}_{d_V}) - \frac{1}{d_V}(\mathbf{I}_L \otimes \mathbf{1}_{d_V \times d_V})$, and here $\sigma$ is simply calculated according to the LayerNorm definition.

Next, we derive calculations for $i \in \{K, Q, V\}$

$$\frac{\partial \mathbf{Z}}{\partial \mathbf{W}_i} = \frac{\partial \text{LayerNorm}(\text{FFN}(\mathbf{Y}) + \mathbf{Y})}{\partial(\text{FFN}(\mathbf{Y}) + \mathbf{Y})} \frac{\partial(\text{FFN}(\mathbf{Y}) + \mathbf{Y})}{\partial \mathbf{Y}} \frac{\partial \mathbf{Y}}{\partial \mathbf{W}_i}$$

Utilizing Property 5 and Lemma 1, we obtain:

$$\frac{\partial(\text{FFN}(\mathbf{Y}) + \mathbf{Y})}{\partial \mathbf{Y}} = \frac{\partial \text{FFN}(\mathbf{Y})}{\partial \mathbf{Y}} + \frac{\partial \mathbf{Y}}{\partial \mathbf{Y}} = \frac{\partial \text{FFN}(\mathbf{Y})}{\partial \mathbf{Y}} + (\mathbf{I}_L \otimes \mathbf{I}_{d_V}) = \frac{\partial \sigma(\mathbf{Y}\mathbf{W}_1)\mathbf{W}_2}{\partial \mathbf{Y}} + (\mathbf{I}_L \otimes \mathbf{I}_{d_V}) =$$

$$= \left(\mathbf{I}_L \otimes \mathbf{W}_2^\top\right) \frac{\partial \sigma(\mathbf{Y}\mathbf{W}_1)}{\partial \mathbf{Y}\mathbf{W}_1} \frac{\partial \mathbf{Y}\mathbf{W}_1}{\partial \mathbf{Y}} + (\mathbf{I}_L \otimes \mathbf{I}_{d_V}) =$$

$$= \left(\mathbf{I}_L \otimes \mathbf{W}_2^\top\right) \text{diag}\left(\text{vec}_r(\mathbf{1}_{\{\mathbf{X}>0\}})\right) \left(\mathbf{I}_L \otimes \mathbf{W}_1^\top\right) + (\mathbf{I}_L \otimes \mathbf{I}_{d_V})$$

and for calculating $\frac{\partial \mathbf{Y}}{\partial \mathbf{W}_i}$ we use Lemma A.2 from Noci et al. (2022):

$$\frac{\partial \mathbf{F}}{\partial \mathbf{W}_V} = \text{softmax}\left(\frac{\mathbf{X}\mathbf{W}_Q\mathbf{W}_K^\top\mathbf{X}^\top}{\sqrt{d_K}}\right)\mathbf{X} \otimes \mathbf{I}_{d_V}$$

$$\frac{\partial \mathbf{F}}{\partial \mathbf{W}_Q} = \left(\mathbf{I}_L \otimes \mathbf{W}_V^\top\mathbf{X}^\top\right)\frac{\partial \mathbf{A}}{\partial \mathbf{M}}\left(\frac{\mathbf{X} \otimes \mathbf{X}\mathbf{W}_K}{\sqrt{d_K}}\right),$$

*where:*

$$\frac{\partial \mathbf{A}}{\partial \mathbf{M}} = \text{blockdiag}\left(\frac{\partial \mathbf{A}_i}{\partial \mathbf{M}_i^\top}\right)$$

*and* $\frac{\partial \mathbf{A}_i}{\partial \mathbf{M}_i^\top} = \text{diag}(\mathbf{A}_i) - \mathbf{A}_i\mathbf{A}_i^\top$, where $\mathbf{A}_i$ is the i-th row of $\mathbf{A}$ in a column vector format. Finally, under the uniform-attention assumption it simplifies to:

$$\frac{\partial \mathbf{A}}{\partial \mathbf{M}} = \frac{1}{n}\mathbf{I}_L \otimes \left(\mathbf{I}_L - \frac{1}{L}\mathbf{1}_{L\times L}\right)$$

Additionally, we can easily expand the result on $\mathbf{W}_K$, where we apply the property 9, therefore:

$$\frac{\partial \mathbf{F}}{\partial \mathbf{W}_K} = \left(\mathbf{I}_L \otimes \mathbf{W}_V^\top\mathbf{X}^\top\right)\frac{\partial \mathbf{A}}{\partial \mathbf{M}}\left(\frac{(\mathbf{X}\mathbf{W}_Q \otimes \mathbf{X})\mathbf{K}_{d_V d_K}}{\sqrt{d_k}}\right),$$

Thus $\frac{\partial \mathbf{Y}}{\partial \mathbf{W}_i}$ can be calculated as follows:

$$\frac{\partial \mathbf{Y}}{\partial \mathbf{W}_i} = \frac{\partial \text{LayerNorm}(\mathbf{F}(\mathbf{X}) + \mathbf{X})}{\partial \mathbf{W}_i} = \frac{\partial \text{LayerNorm}(\mathbf{F}(\mathbf{X}) + \mathbf{X})}{\partial(\mathbf{F}(\mathbf{X}) + \mathbf{X})}\frac{\partial \mathbf{F}(\mathbf{X})}{\partial \mathbf{W}_i}$$

where $\frac{\partial \mathbf{F}(\mathbf{X})}{\partial \mathbf{W}_i}$ is calculated according to Lemma A.2 from Noci et al. (2022), which we mentioned earlier above and $\frac{\partial \text{LayerNorm}(\mathbf{F}(\mathbf{X})+\mathbf{X})}{\partial(\mathbf{F}(\mathbf{X})+\mathbf{X})}$ is calculated according to Theorem 2.

Substituting the expressions ends the proof. $\qquad\square$

## C.5   PROOF OF THEOREM 5

**Theorem 10** (Detailed version of Theorem 5). *Let* $\mathbf{X} \in \mathbb{R}^{L\times d_V}$, $\mathbf{Y} \in \mathbb{R}^{L\times d_V}$, $\mathbf{W}_1 \in \mathbb{R}^{d_V\times d_{ff}}$, $\mathbf{W}_2 \in \mathbb{R}^{d_{ff}\times d_V}$, $\mathbf{W}_Q, \mathbf{W}_K \in \mathbb{R}^{d_V\times d_K}$, $\mathbf{W}_V \in \mathbb{R}^{d_V\times d_V}$. *Define*

$$\mathbf{S}(\mathbf{Y}, \mathbf{W}_1, \mathbf{W}_2) = \sigma(\mathbf{Y}\mathbf{W}_1)\mathbf{W}_2 + \mathbf{Y} \in \mathbb{R}^{L\times d_V}, \qquad \mathbf{Z} = \text{LayerNorm}(\mathbf{S}) \in \mathbb{R}^{L\times d_V},$$

*and abbreviate (according to Theorems 2–3):*

$$\mathbf{J}_Z := \frac{\partial \text{LayerNorm}(\mathbf{S})}{\partial \mathbf{S}} \in \mathbb{R}^{Ld_V\times Ld_V}, \quad \mathbf{H}_Z := \frac{\partial^2 \text{LayerNorm}(\mathbf{S})}{\partial \mathbf{S}^2} \in \mathbb{R}^{(Ld_V)^2\times Ld_V}$$

*Let further*

$$\mathbf{D}_\sigma := \text{diag}\left(\text{vec}_r(\mathbf{1}_{\{\mathbf{Y}\mathbf{W}_1>0\}})\right) \in \mathbb{R}^{Ld_{ff}\times Ld_{ff}}$$

*from Lemma 1.*

*Define the residual-Jacobian*

$$\mathbf{J}_{SY} := \frac{\partial \mathbf{S}}{\partial \mathbf{Y}} = (\mathbf{I}_L \otimes \mathbf{W}_2^\top)\mathbf{D}_\sigma(\mathbf{I}_L \otimes \mathbf{W}_1^\top) + (\mathbf{I}_L \otimes \mathbf{I}_{d_V}) \in \mathbb{R}^{Ld_V\times Ld_V},$$

*and for the first residual* $\mathbf{Y} = \text{LayerNorm}(\mathbf{F}(\mathbf{X}) + \mathbf{X})$, *set*

$$\mathbf{J}_Y := \frac{\partial \text{LayerNorm}(\mathbf{F}(\mathbf{X}) + \mathbf{X})}{\partial(\mathbf{F}(\mathbf{X}) + \mathbf{X})} \in \mathbb{R}^{Ld_V\times Ld_V}, \quad \mathbf{H}_Y := \frac{\partial^2 \text{LayerNorm}(\mathbf{F}(\mathbf{X}) + \mathbf{X})}{\partial(\mathbf{F}(\mathbf{X}) + \mathbf{X})^2} \in \mathbb{R}^{(Ld_V)^2\times Ld_V}$$

*calculated by Theorems 2–3.*

*Denote parameter sizes*

$$n_1 = d_V d_{ff}, \quad n_2 = d_{ff} d_V, \quad n_Q = n_K = d_V d_K, \quad n_V = d_V^2.$$

*Let the attention-side Jacobians (from Theorem 4, can be calculated according to Noci et al. (2022)) be*

$$\mathbf{G}_V := \frac{\partial \mathbf{F}}{\partial \mathbf{W}_V} \in \mathbb{R}^{Ld_V \times n_V}, \quad \mathbf{G}_Q := \frac{\partial \mathbf{F}}{\partial \mathbf{W}_Q} \in \mathbb{R}^{Ld_V \times n_Q}, \quad \mathbf{G}_K := \frac{\partial \mathbf{F}}{\partial \mathbf{W}_K} \in \mathbb{R}^{Ld_V \times n_K}.$$

*For $i \in \{1, 2\}$ and $k \in \{K, Q, V\}$, define first-layer Jacobians*

$$\mathbf{B}_1 := \frac{\partial \mathbf{S}}{\partial \mathbf{W}_1} = (\mathbf{I}_L \otimes \mathbf{W}_2^\top) \mathbf{D}_\sigma (\mathbf{Y} \otimes \mathbf{I}_{d_{ff}}) \in \mathbb{R}^{Ld_V \times n_1},$$

$$\mathbf{B}_2 := \frac{\partial \mathbf{S}}{\partial \mathbf{W}_2} = \sigma(\mathbf{Y}\mathbf{W}_1) \otimes \mathbf{I}_{d_V} \in \mathbb{R}^{Ld_V \times n_2},$$

$$\mathbf{B}_k := \frac{\partial \mathbf{S}}{\partial \mathbf{W}_k} = \mathbf{J}_{SY} \mathbf{J}_Y \mathbf{G}_k \in \mathbb{R}^{Ld_V \times n_k}.$$

*Then the Hessian blocks of the Transformer output $\mathbf{Z}$ w.r.t. parameters $(\mathbf{W}_i, \mathbf{W}_j)$ are*

$$\boxed{\mathbf{H}_{\mathrm{tr}}^{(i,j)} := \frac{\partial^2 \mathbf{Z}}{\partial \mathbf{W}_i \partial \mathbf{W}_j} = (\mathbf{J}_Z \otimes \mathbf{I}_{n_i}) \boldsymbol{\xi}_{ij} + (\mathbf{I}_{Ld_V} \otimes \mathbf{B}_i^\top) \mathbf{H}_Z \mathbf{B}_j} \tag{12}$$

*with*

$$\boldsymbol{\xi}_{ij} := \frac{\partial}{\partial \mathbf{W}_j} \left( \frac{\partial \mathbf{S}}{\partial \mathbf{W}_i} \right) \in \mathbb{R}^{(Ld_V \cdot n_i) \times n_j}.$$

*The second Jacobians $\boldsymbol{\xi}_{ij}$ for all pairs $(i, j)$ are given almost everywhere by:*

*1) Pure-FFN pairs:*

$$\boldsymbol{\xi}_{11} = \mathbf{0}_{(Ld_V \cdot n_1) \times n_1}, \qquad \boldsymbol{\xi}_{22} = \mathbf{0}_{(Ld_V \cdot n_2) \times n_2},$$

$$\boldsymbol{\xi}_{12} = (\mathbf{I}_L \otimes \mathbf{K}_{d_V, d_{ff}} \otimes \mathbf{I}_{d_V}) (\mathbf{I}_{Ld_{ff}} \otimes \mathrm{vec}_r(\mathbf{I}_{d_V})) (\mathbf{D}_\sigma (\mathbf{Y} \otimes \mathbf{I}_{d_{ff}})),$$

$$\boldsymbol{\xi}_{21} = (\mathbf{I}_L \otimes \mathbf{W}_2^\top) \mathbf{D}_\sigma (\mathbf{I}_L \otimes \mathbf{K}_{d_{ff}, d_V} \otimes \mathbf{I}_{d_{ff}}) (\mathbf{I}_{Ld_V} \otimes \mathrm{vec}_r(\mathbf{I}_{d_{ff}})).$$

*Both $\boldsymbol{\xi}_{12}$ and $\boldsymbol{\xi}_{21}$ are $(Ld_V \cdot n_1) \times n_2$ and $(Ld_V \cdot n_2) \times n_1$ respectively. They agree almost everywhere when pre- and post-composed in equation 12 (see symmetry discussion).*

*2) FFN–attention pairs ($k \in \{K, Q, V\}$):*

$$\boldsymbol{\xi}_{1k} = ((\mathbf{I}_L \otimes \mathbf{W}_2^\top)\mathbf{D}_\sigma \otimes \mathbf{I}_{n_k}) (\mathbf{I}_L \otimes \mathbf{K}_{d_{ff}, d_V} \otimes \mathbf{I}_{d_{ff}}) (\mathbf{I}_{Ld_V} \otimes \mathrm{vec}_r(\mathbf{I}_{d_{ff}})) (\mathbf{J}_Y \mathbf{G}_k),$$

$$\boldsymbol{\xi}_{2k} = (\mathbf{I}_L \otimes \mathbf{K}_{d_V, d_{ff}} \otimes \mathbf{I}_{d_V}) (\mathbf{I}_{Ld_{ff}} \otimes \mathrm{vec}_r(\mathbf{I}_{d_V})) (\mathbf{D}_\sigma(\mathbf{I}_L \otimes \mathbf{W}_1^\top) \mathbf{J}_Y \mathbf{G}_k).$$

*Dimensions: $\boldsymbol{\xi}_{1k} \in \mathbb{R}^{(Ld_V \cdot n_1) \times n_k}$ and $\boldsymbol{\xi}_{2k} \in \mathbb{R}^{(Ld_V \cdot n_2) \times n_k}$.*

*3) Pure-attention pairs ($k, \ell \in \{K, Q, V\}$):*

$$\boldsymbol{\xi}_{k\ell} = (\mathbf{J}_{SY} \otimes \mathbf{I}_{n_k}) [(\mathbf{I}_{Ld_V} \otimes \mathbf{G}_k^\top) (\mathbf{H}_Y \mathbf{G}_\ell) + (\mathbf{J}_Y \otimes \mathbf{I}_{n_k}) \boldsymbol{\Phi}_{k\ell}],$$

*where $\boldsymbol{\Phi}_{k\ell} := \frac{\partial \mathbf{G}_k}{\partial \mathbf{W}_\ell} \in \mathbb{R}^{(Ld_V \cdot n_k) \times n_\ell}$ are second derivatives of the attention map $\mathbf{F}$ w.r.t. its weights. The exact values are calculated in Lemma 2 basing on the results from Ormaniec et al. (2024). All matrices are dimensionally consistent: $\boldsymbol{\xi}_{k\ell} \in \mathbb{R}^{(Ld_V \cdot n_k) \times n_\ell}$.*

*Finally, the Hessian block equation 12 has size $\mathbf{H}_{\mathrm{tr}}^{(i,j)} \in \mathbb{R}^{(Ld_V \cdot n_i) \times n_j}$.*

*Moreover, all mixed blocks are symmetric almost everywhere:*

$$\mathbf{H}_{\mathrm{tr}}^{(i,j)} = \mathbf{H}_{\mathrm{tr}}^{(j,i)} \quad a.e.,$$

*because (i) the only nonlinearities with potentially nonzero second differential are LayerNorm (handled by $\mathbf{H}_Z, \mathbf{H}_Y$ which are symmetric by construction in Theorem 3) and ReLU (whose Hessian is zero a.e., Lemma 1), and (ii) all remaining mappings are multilinear in the parameters; thus, by repeated applications of Proposition 3 and Proposition 6, the mixed-partials commute almost everywhere.*

*Proof.* We differentiate the Jacobian from Theorem 4 using Proposition 3 (matrix-product derivative), Proposition 6 (Kronecker-product derivative), Proposition 9, the Identification Theorem 1, and Lemma 1.

Step 1. For any $i \in \{1, 2, K, Q, V\}$ we have

$$\frac{\partial \mathbf{Z}}{\partial \mathbf{W}_i} = \mathbf{J}_Z \mathbf{B}_i, \qquad \mathbf{J}_Z \in \mathbb{R}^{Ld_V \times Ld_V},$$

where $\mathbf{B}_i := \frac{\partial \mathbf{S}}{\partial \mathbf{W}_i}$ is given casewise by

$$\mathbf{B}_1 = (\mathbf{I}_L \otimes \mathbf{W}_2^\top) \mathbf{D}_\sigma (\mathbf{Y} \otimes \mathbf{I}_{d_{ff}}) \in \mathbb{R}^{Ld_V \times n_1}, \quad \mathbf{B}_2 = \sigma(\mathbf{YW}_1) \otimes \mathbf{I}_{d_V} \in \mathbb{R}^{Ld_V \times n_2},$$

$$\mathbf{B}_k = \mathbf{J}_{SY} \mathbf{J}_Y \mathbf{G}_k \in \mathbb{R}^{Ld_V \times n_k}, \qquad k \in \{K, Q, V\},$$

with $\mathbf{J}_{SY} = \frac{\partial \mathbf{S}}{\partial \mathbf{Y}} = (\mathbf{I}_L \otimes \mathbf{W}_2^\top)\mathbf{D}_\sigma(\mathbf{I}_L \otimes \mathbf{W}_1^\top) + (\mathbf{I}_L \otimes \mathbf{I}_{d_V}) \in \mathbb{R}^{Ld_V \times Ld_V}$, $\mathbf{J}_Y \in \mathbb{R}^{Ld_V \times Ld_V}$ and $\mathbf{G}_k$ as in Theorem 4. By Proposition 3 and Theorem 3 we obtain the Hessian block

$$\frac{\partial^2 \mathbf{Z}}{\partial \mathbf{W}_i \partial \mathbf{W}_j} = (\mathbf{J}_Z \otimes \mathbf{I}_{n_i}) \, \boldsymbol{\xi}_{ij} + (\mathbf{I}_{Ld_V} \otimes \mathbf{B}_i^\top) \mathbf{H}_Z \mathbf{B}_j, \qquad \boldsymbol{\xi}_{ij} := \frac{\partial \mathbf{B}_i}{\partial \mathbf{W}_j} \in \mathbb{R}^{(Ld_V \cdot n_i) \times n_j}.$$

Step 2: First-level Jacobians $\mathbf{B}_i$ (dimensions). From Theorem 4 and Lemma 1:

$$\mathbf{B}_1 = (\mathbf{I}_L \otimes \mathbf{W}_2^\top) \mathbf{D}_\sigma (\mathbf{Y} \otimes \mathbf{I}_{d_{ff}}) \in \mathbb{R}^{Ld_V \times n_1}, \quad \mathbf{B}_2 = \sigma(\mathbf{YW}_1) \otimes \mathbf{I}_{d_V} \in \mathbb{R}^{Ld_V \times n_2},$$

where $\mathbf{D}_\sigma \in \mathbb{R}^{Ld_{ff} \times Ld_{ff}}, (\mathbf{Y} \otimes \mathbf{I}_{d_{ff}}) \in \mathbb{R}^{Ld_{ff} \times d_V d_{ff}}$. For $k \in \{K, Q, V\}$,

$$\mathbf{B}_k = \mathbf{J}_{SY} \mathbf{J}_Y \mathbf{G}_k \in \mathbb{R}^{Ld_V \times n_k}.$$

Step 3: Second Jacobians $\boldsymbol{\xi}_{ij}$ for all pairs.

3.1) Pure-FFN pairs. - $(1,1)$: $\mathbf{B}_1$ depends on $\mathbf{W}_1$ only through $\sigma(\mathbf{YW}_1)$, whose Hessian is zero a.e. by Lemma 1, while $\mathbf{YW}_1$ is linear in $\mathbf{W}_1$ (Property 5). Hence $\boldsymbol{\xi}_{11} = \mathbf{0}$ with the stated size.

- $(2,2)$: $\mathbf{B}_2$ is linear in $\mathbf{W}_2$ (Property 5), hence $\boldsymbol{\xi}_{22} = \mathbf{0}$.

- $(1,2)$: Differentiate $\mathbf{B}_2 = \sigma(\mathbf{YW}_1) \otimes \mathbf{I}_{d_V}$ w.r.t. $\mathbf{W}_1$. Using Proposition 6 for $\frac{\partial (\mathbf{X} \otimes \mathbf{Y})}{\partial \mathbf{X}}$ with $\mathbf{X} = \sigma(\mathbf{YW}_1)$ and $\mathbf{Y} = \mathbf{I}_{d_V}$, we get

$$\frac{\partial \mathbf{B}_2}{\partial \mathbf{W}_1} = (\mathbf{I}_L \otimes \mathbf{K}_{d_V, d_{ff}} \otimes \mathbf{I}_{d_V}) (\mathbf{I}_{Ld_{ff}} \otimes \mathrm{vec}_r(\mathbf{I}_{d_V})) \frac{\partial \, \mathrm{vec}_r(\sigma(\mathbf{YW}_1))}{\partial \mathbf{W}_1}.$$

By Lemma 1 and Property 5, $\frac{\partial \, \mathrm{vec}_r(\sigma(\mathbf{YW}_1))}{\partial \mathbf{W}_1} = \mathbf{D}_\sigma (\mathbf{Y} \otimes \mathbf{I}_{d_{ff}})$. Thus

$$\boldsymbol{\xi}_{12} = (\mathbf{I}_L \otimes \mathbf{K}_{d_V, d_{ff}} \otimes \mathbf{I}_{d_V}) (\mathbf{I}_{Ld_{ff}} \otimes \mathrm{vec}_r(\mathbf{I}_{d_V})) (\mathbf{D}_\sigma (\mathbf{Y} \otimes \mathbf{I}_{d_{ff}})).$$

- $(2,1)$: Differentiate $\mathbf{B}_1 = (\mathbf{I}_L \otimes \mathbf{W}_2^\top) \mathbf{D}_\sigma (\mathbf{Y} \otimes \mathbf{I}_{d_{ff}})$ w.r.t. $\mathbf{W}_2$. Using Proposition 3 on the left factor $(\mathbf{I}_L \otimes \mathbf{W}_2^\top)$ and Proposition 6 plus Proposition 9 for its derivative, we obtain

$$\frac{\partial \, \mathrm{vec}_r(\mathbf{B}_1)}{\partial \mathbf{W}_2} = (\mathbf{I}_{Ld_V} \otimes ((\mathbf{Y} \otimes \mathbf{I}_{d_{ff}})^\top \mathbf{D}_\sigma^\top)) \frac{\partial \, \mathrm{vec}_r(\mathbf{I}_L \otimes \mathbf{W}_2^\top)}{\partial \mathbf{W}_2}.$$

By Proposition 6 and Proposition 9,

$$\frac{\partial \, \mathrm{vec}_r(\mathbf{I}_L \otimes \mathbf{W}_2^\top)}{\partial \mathbf{W}_2} = (\mathbf{I}_L \otimes \mathbf{K}_{d_V, L} \otimes \mathbf{I}_{d_{ff}}) (\mathrm{vec}_r(\mathbf{I}_L) \otimes \mathbf{I}_{d_V d_{ff}}) \mathbf{K}_{d_{ff}, d_V}.$$

Collecting,

$$\boldsymbol{\xi}_{21} = (\mathbf{I}_L \otimes \mathbf{W}_2^\top) \mathbf{D}_\sigma (\mathbf{I}_L \otimes \mathbf{K}_{d_{ff}, d_V} \otimes \mathbf{I}_{d_{ff}}) (\mathbf{I}_{Ld_V} \otimes \mathrm{vec}_r(\mathbf{I}_{d_{ff}})),$$

which is the stated form. (Both $\boldsymbol{\xi}_{12}$ and $\boldsymbol{\xi}_{21}$ are consistent and coincide almost everywhere when inserted into equation 12; see symmetry below.)

3.2) FFNattention pairs $(1, k), (2, k)$ with $k \in \{K, Q, V\}$. - $(1, k)$: $\mathbf{B}_1 = (\mathbf{I}_L \otimes \mathbf{W}_2^\top) \mathbf{D}_\sigma (\mathbf{Y} \otimes \mathbf{I}_{d_{ff}})$. Almost everywhere $\frac{\partial \mathbf{D}_\sigma}{\partial \mathbf{Y}} = \mathbf{0}$ by Lemma 1. Hence only the last factor varies with $\mathbf{W}_k$. Using Proposition 3 (with the first factors constant a.e.), and the chain rule through $\mathbf{Y}$:

$$\frac{\partial \operatorname{vec}_r(\mathbf{Y} \otimes \mathbf{I}_{d_{ff}})}{\partial \mathbf{W}_k} = \left( \frac{\partial(\mathbf{Y} \otimes \mathbf{I}_{d_{ff}})}{\partial \mathbf{Y}} \right) \frac{\partial \operatorname{vec}_r(\mathbf{Y})}{\partial \mathbf{W}_k}.$$

By Proposition 6 with $\mathbf{X} = \mathbf{Y}$ and $\mathbf{Y} = \mathbf{I}_{d_{ff}}$,

$$\frac{\partial(\mathbf{Y} \otimes \mathbf{I}_{d_{ff}})}{\partial \mathbf{Y}} = \left( \mathbf{I}_L \otimes \mathbf{K}_{d_{ff}, d_V} \otimes \mathbf{I}_{d_{ff}} \right) \left( \mathbf{I}_{L d_V} \otimes \operatorname{vec}_r(\mathbf{I}_{d_{ff}}) \right).$$

Also $\frac{\partial \operatorname{vec}_r(\mathbf{Y})}{\partial \mathbf{W}_k} = \mathbf{J}_Y \mathbf{G}_k$ (Theorem 4 and Theorem 2). Therefore

$$\boldsymbol{\xi}_{1k} = \left( (\mathbf{I}_L \otimes \mathbf{W}_2^\top) \mathbf{D}_\sigma \otimes \mathbf{I}_{n_k} \right) \left( \mathbf{I}_L \otimes \mathbf{K}_{d_{ff}, d_V} \otimes \mathbf{I}_{d_{ff}} \right) \left( \mathbf{I}_{L d_V} \otimes \operatorname{vec}_r(\mathbf{I}_{d_{ff}}) \right) (\mathbf{J}_Y \mathbf{G}_k).$$

- $(2, k)$: $\mathbf{B}_2 = \sigma(\mathbf{Y} \mathbf{W}_1) \otimes \mathbf{I}_{d_V}$. Differentiating the Kronecker product w.r.t. its first factor and applying the chain rule through $\mathbf{Y}$,

$$\boldsymbol{\xi}_{2k} = \left( \mathbf{I}_L \otimes \mathbf{K}_{d_V, d_{ff}} \otimes \mathbf{I}_{d_V} \right) \left( \mathbf{I}_{L d_{ff}} \otimes \operatorname{vec}_r(\mathbf{I}_{d_V}) \right) \left( \mathbf{D}_\sigma (\mathbf{I}_L \otimes \mathbf{W}_1^\top) \mathbf{J}_Y \mathbf{G}_k \right),$$

where we used Property 5 to write $\frac{\partial(\mathbf{Y} \mathbf{W}_1)}{\partial \mathbf{Y}} = \mathbf{I}_L \otimes \mathbf{W}_1^\top$ and Lemma 1 for $\frac{\partial \sigma(\cdot)}{\partial(\cdot)} = \mathbf{D}_\sigma$.

3.3) Pure-attention pairs $(k, \ell)$ with $k, \ell \in \{K, Q, V\}$. We start from $\mathbf{B}_k = \mathbf{J}_{SY} \mathbf{J}_Y \mathbf{G}_k$. Almost everywhere $\frac{\partial \mathbf{J}_{SY}}{\partial \mathbf{Y}} = \mathbf{0}$ because $\mathbf{D}_\sigma$ is piecewise constant (Lemma 1). Therefore,

$$\frac{\partial \operatorname{vec}_r(\mathbf{B}_k)}{\partial \mathbf{W}_\ell} = (\mathbf{J}_{SY} \otimes \mathbf{I}_{n_k}) \frac{\partial \operatorname{vec}_r(\mathbf{J}_Y \mathbf{G}_k)}{\partial \mathbf{W}_\ell}$$

by Proposition 3. Again by Proposition 3 with $\mathbf{A}(\cdot) = \mathbf{J}_Y$ and $\mathbf{B}(\cdot) = \mathbf{G}_k$,

$$\frac{\partial \operatorname{vec}_r(\mathbf{J}_Y \mathbf{G}_k)}{\partial \mathbf{W}_\ell} = (\mathbf{J}_Y \otimes \mathbf{I}_{n_k}) \boldsymbol{\Phi}_{k\ell} + \left( \mathbf{I}_{L d_V} \otimes \mathbf{G}_k^\top \right) \frac{\partial \operatorname{vec}_r(\mathbf{J}_Y)}{\partial \mathbf{W}_\ell}.$$

By Theorem 3 and the Identification Theorem 1, $\frac{\partial \operatorname{vec}_r(\mathbf{J}_Y)}{\partial \mathbf{W}_\ell} = \mathbf{H}_Y \mathbf{G}_\ell$. Thus

$$\boldsymbol{\xi}_{k\ell} = (\mathbf{J}_{SY} \otimes \mathbf{I}_{n_k}) \left[ \left( \mathbf{I}_{L d_V} \otimes \mathbf{G}_k^\top \right) (\mathbf{H}_Y \mathbf{G}_\ell) + (\mathbf{J}_Y \otimes \mathbf{I}_{n_k}) \boldsymbol{\Phi}_{k\ell} \right].$$

It remains to specify $\boldsymbol{\Phi}_{k\ell} := \frac{\partial \mathbf{G}_k}{\partial \mathbf{W}_\ell}$. Using the explicit $\mathbf{G}_k$ from Theorem 4 and only Proposition 3, Proposition 6, and Proposition 9, we obtain the forms stated in the theorem. Under the uniform-attention simplification (so $\frac{\partial \mathbf{A}}{\partial \mathbf{M}}$ is a constant matrix), $\mathbf{G}_V$ does not depend on $\mathbf{W}_Q, \mathbf{W}_K, \mathbf{W}_V$; $\mathbf{G}_Q$ does not depend on $\mathbf{W}_Q$; $\mathbf{G}_K$ does not depend on $\mathbf{W}_K$; hence $\boldsymbol{\Phi}_{VV} = \boldsymbol{\Phi}_{VQ} = \boldsymbol{\Phi}_{VK} = \boldsymbol{\Phi}_{QQ} = \boldsymbol{\Phi}_{KK} = \mathbf{0}$; and the remaining mixed terms are given by differentiating the Kronecker factors using Proposition 6 and the transpose dependence using Proposition 9, exactly as written.

Step 4: Symmetry of mixed partials. All nonlinearities that could obstruct symmetry are ReLU and LayerNorm. ReLU has zero Hessian almost everywhere (Lemma 1), so its contribution to second differentials vanishes a.e. LayerNorm Hessians $\mathbf{H}_Z$ and $\mathbf{H}_Y$ are the derivatives of Jacobians w.r.t. their inputs and enter symmetrically (Theorem 3). All remaining mappings are multilinear in parameters and matrices independent of $(\mathbf{W}_i, \mathbf{W}_j)$; therefore, by repeated applications of Proposition 3 and Proposition 6, the mixed partials commute, giving $\mathbf{H}_{\mathrm{tr}}^{(i,j)} = \mathbf{H}_{\mathrm{tr}}^{(j,i)}$ almost everywhere.

This completes the proof. □

### C.6 PROOF OF THEOREM 6

*Proof.* We start from the block formula equation 12:

$$\mathbf{H}_{\mathrm{tr}}^{(i,j)} = (\mathbf{J}_Z \otimes \mathbf{I}_{n_i}) \boldsymbol{\xi}_{ij} + \left( \mathbf{I}_{L d_V} \otimes \mathbf{B}_i^\top \right) \mathbf{H}_Z \mathbf{B}_j.$$

Applying the matrix sum norm (Property 8) and the product norm (Property 10) together with the Kronecker product norm (Property 9) yields

$$\left\| \mathbf{H}_{\mathrm{tr}}^{(i,j)} \right\|_2 \leq \left\| \mathbf{J}_Z \otimes \mathbf{I}_{n_i} \right\|_2 \| \boldsymbol{\xi}_{ij} \|_2 + \left\| \mathbf{I}_{L d_V} \otimes \mathbf{B}_i^\top \right\|_2 \| \mathbf{H}_Z \|_2 \| \mathbf{B}_j \|_2 = \| \mathbf{J}_Z \|_2 \| \boldsymbol{\xi}_{ij} \|_2 + \| \mathbf{B}_i \|_2 \| \mathbf{H}_Z \|_2 \| \mathbf{B}_j \|_2,$$

establishing equation 8.

It remains to provide explicit operator-norm estimates for $\|\mathbf{B}_i\|_2$ and $\|\boldsymbol{\xi}_{ij}\|_2$ used inside equation 8. We rely on Properties 10, 9, 8, 7, 12, and the commutation properties (Definition 3). Throughout we use $\|\mathbf{K}_{m,n}\|_2 = 1$ for commutation matrices, and the identities $\|\mathrm{vec}_r(\mathbf{I}_d)\|_2 = \|\mathbf{I}_d\|_F = \sqrt{d}$ (Property 7) and $\|\mathbf{I}_p\|_2 = 1$.

As we've already shown in C.1:
$$\left\|\frac{\partial \mathbf{A}}{\partial \mathbf{T}}\right\|_2 \le \frac{1}{L}.$$

$$\|\mathbf{Z}_1\|_2 = \|(\mathbf{I}_L \otimes \mathbf{X}^\top)(\partial \mathbf{A}/\partial \mathbf{T})(\mathbf{X} \otimes \mathbf{X})\|_2 \le \|\mathbf{X}\|_2 \frac{1}{L} \|\mathbf{X}\|_2^2 = \frac{1}{L}\|\mathbf{X}\|_2^3$$

$$\left\|\frac{\partial^2 \mathbf{A}}{\partial \mathbf{T}^2}\right\|_2 \le 6, \qquad \|\mathbf{Z}_2\|_2 \le \|\mathbf{X}\|_2^5 \left\|\frac{\partial^2 \mathbf{A}}{\partial \mathbf{T}^2}\right\|_2 \le 6\|\mathbf{X}\|_2^5,$$

$$\|\mathbf{A}\|_2 \le \sqrt{LL}\,\|\mathbf{A}\|_{\max} = L.$$

Therefore $\|\mathbf{AX}\|_2 \le \|\mathbf{A}\|_2\|\mathbf{X}\|_2 \le L\|\mathbf{X}\|_2$ (Property 10).

We also use the attention curvature blocks $\boldsymbol{\Phi}_{k\ell}$ from Lemma 2. Using Properties 10, 9 and the bounds on $\|\mathbf{Z}_1\|_2$, $\|\mathbf{Z}_2\|_2$ above, we have (again similarly to C.1)

$$\|\boldsymbol{\Phi}_{VV}\|_2 = 0,$$

$$\|\boldsymbol{\Phi}_{QQ}\|_2 \le \frac{2}{Ld_V d_K}\|\mathbf{W}_V\|_2\|\mathbf{W}_K\|_2\|\mathbf{Z}_2\|_2\|\mathbf{W}_K\|_2 \le \frac{12}{Ld_V d_K}\|\mathbf{W}_V\|_2\|\mathbf{W}_K\|_2^2\|\mathbf{X}\|_2^5,$$

$$\|\boldsymbol{\Phi}_{VQ}\|_2 \le \frac{2}{Ld_V\sqrt{d_K}}\|\mathbf{I}_L \otimes \mathbf{S}\|_2\|\mathbf{Z}_1\|_2\|\mathbf{I}_{d_V} \otimes \mathbf{W}_K\|_2 \le \frac{2}{L^2\sqrt{d_V d_K}}\|\mathbf{W}_K\|_2\|\mathbf{X}\|_2^3,$$

$$\|\boldsymbol{\Phi}_{QK}\|_2 \le \frac{2}{Ld_V d_K}\|\mathbf{W}_V\|_2\|\mathbf{W}_K\|_2\|\mathbf{Z}_2\|_2\|\mathbf{W}_Q\|_2 + \frac{2}{Ld_V\sqrt{d_K}}\|\mathbf{W}_V\|_2\|\mathbf{Z}_1\|_2\|\mathbf{S}\|_2$$

$$\le \frac{12}{Ld_V d_K}\|\mathbf{W}_V\|_2\|\mathbf{W}_K\|_2\|\mathbf{W}_Q\|_2\|\mathbf{X}\|_2^5 + \frac{2}{L^2\sqrt{d_V d_K}}\|\mathbf{W}_V\|_2\|\mathbf{X}\|_2^3,$$

and $\|\boldsymbol{\Phi}_{KQ}\|_2$ is analogous by symmetry (Definition 3 and $\|\mathbf{K}_{m,n}\|_2 = 1$), while $\|\boldsymbol{\Phi}_{QV}\|_2$, $\|\boldsymbol{\Phi}_{KV}\|_2$ match $\|\boldsymbol{\Phi}_{VQ}\|_2$ up to swapping roles.

Next we estimate each $\|\mathbf{B}_i\|_2$ and $\|\boldsymbol{\xi}_{ij}\|_2$.

A) Bounds for $\|\mathbf{B}_i\|_2$.

- $\mathbf{B}_1 = (\mathbf{I}_L \otimes \mathbf{W}_2^\top)\,\mathbf{D}_\sigma\,(\mathbf{Y} \otimes \mathbf{I}_{d_{ff}})$ (Theorem 5; Lemma 1). Using Properties 9, 10, 12, and $\|\mathbf{D}_\sigma\|_2 \le 1$,
$$\|\mathbf{B}_1\|_2 \le \|\mathbf{I}_L \otimes \mathbf{W}_2^\top\|_2\|\mathbf{D}_\sigma\|_2\|\mathbf{Y} \otimes \mathbf{I}_{d_{ff}}\|_2 = \|\mathbf{W}_2\|_2\|\mathbf{Y}\|_2. \tag{13}$$

- $\mathbf{B}_2 = \sigma(\mathbf{Y}\mathbf{W}_1) \otimes \mathbf{I}_{d_V}$ (Theorem 5), hence
$$\|\mathbf{B}_2\|_2 = \|\sigma(\mathbf{Y}\mathbf{W}_1)\|_2 \tag{14}$$

by Property 9.

- For $k \in \{K, Q, V\}$: $\mathbf{B}_k = \mathbf{J}_{SY}\,\mathbf{J}_Y\,\mathbf{G}_k$ (Theorem 5), so
$$\|\mathbf{B}_k\|_2 \le \|\mathbf{J}_{SY}\|_2\|\mathbf{J}_Y\|_2\|\mathbf{G}_k\|_2 \tag{15}$$

(Property 10). Here $\mathbf{J}_{SY} = (\mathbf{I}_L \otimes \mathbf{W}_2^\top)\mathbf{D}_\sigma(\mathbf{I}_L \otimes \mathbf{W}_1^\top) + (\mathbf{I}_L \otimes \mathbf{I}_{d_V})$ implies

$$\|\mathbf{J}_{SY}\|_2 \le \|\mathbf{I}_L \otimes \mathbf{W}_2^\top\|_2\|\mathbf{D}_\sigma\|_2\|\mathbf{I}_L \otimes \mathbf{W}_1^\top\|_2 + \|\mathbf{I}_L \otimes \mathbf{I}_{d_V}\|_2 = \|\mathbf{W}_2\|_2\|\mathbf{W}_1\|_2 + 1, \tag{16}$$

by Properties 8, 10, 9, 12, and $\|\mathbf{D}_\sigma\|_2 \le 1$.

Furthermore, using the attention-Jacobian forms (Theorem 4) and Properties 10, 9:

$$\|\mathbf{G}_V\|_2 \le L\|\mathbf{X}\|_2, \|\mathbf{G}_Q\|_2 \le \frac{1}{L\sqrt{d_K}}\|\mathbf{W}_V\|_2\|\mathbf{W}_K\|_2\|\mathbf{X}\|_2^3, \|\mathbf{G}_K\|_2 \le \frac{1}{L\sqrt{d_K}}\|\mathbf{W}_V\|_2\|\mathbf{W}_Q\|_2\|\mathbf{X}\|_2^3. \tag{17}$$

B) Bounds for $\|\boldsymbol{\xi}_{ij}\|_2$. Using the explicit formulas from Theorem 5, Properties 9, 10, 7, and $\|\mathbf{K}_{m,n}\|_2 = 1$:

B.1 Pure-FFN pairs:

$$\|\boldsymbol{\xi}_{11}\|_2 = 0, \tag{18}$$

$$\|\boldsymbol{\xi}_{22}\|_2 = 0, \tag{19}$$

$$\|\boldsymbol{\xi}_{12}\|_2 \leq \|\mathbf{I}_L \otimes \mathbf{K}_{d_V, d_{ff}} \otimes \mathbf{I}_{d_V}\|_2 \|\mathbf{I}_{Ld_{ff}} \otimes \mathrm{vec}_r(\mathbf{I}_{d_V})\|_2 \|\mathbf{D}_\sigma\|_2 \|\mathbf{Y} \otimes \mathbf{I}_{d_{ff}}\|_2$$
$$= 1 \cdot \|\mathrm{vec}_r(\mathbf{I}_{d_V})\|_2 \cdot 1 \cdot \|\mathbf{Y}\|_2 = \sqrt{d_V} \|\mathbf{Y}\|_2, \tag{20}$$

$$\|\boldsymbol{\xi}_{21}\|_2 \leq \|\mathbf{I}_L \otimes \mathbf{W}_2^\top\|_2 \|\mathbf{D}_\sigma\|_2 \|\mathbf{I}_L \otimes \mathbf{K}_{d_{ff}, d_V} \otimes \mathbf{I}_{d_{ff}}\|_2 \|\mathbf{I}_{Ld_V} \otimes \mathrm{vec}_r(\mathbf{I}_{d_{ff}})\|_2$$
$$= \|\mathbf{W}_2\|_2 \cdot 1 \cdot 1 \cdot \|\mathrm{vec}_r(\mathbf{I}_{d_{ff}})\|_2 = \sqrt{d_{ff}} \|\mathbf{W}_2\|_2. \tag{21}$$

B.2 FFNattention pairs ($k \in \{K, Q, V\}$):

$$\|\boldsymbol{\xi}_{1k}\|_2 \leq \|(\mathbf{I}_L \otimes \mathbf{W}_2^\top)\mathbf{D}_\sigma \otimes \mathbf{I}_{n_k}\|_2 \|\mathbf{I}_L \otimes \mathbf{K}_{d_{ff}, d_V} \otimes \mathbf{I}_{d_{ff}}\|_2 \|\mathbf{I}_{Ld_V} \otimes \mathrm{vec}_r(\mathbf{I}_{d_{ff}})\|_2 \|\mathbf{J}_Y\|_2 \|\mathbf{G}_k\|_2$$
$$\leq \|\mathbf{W}_2\|_2 \cdot 1 \cdot 1 \cdot \sqrt{d_{ff}} \cdot \|\mathbf{J}_Y\|_2 \|\mathbf{G}_k\|_2 = \sqrt{d_{ff}} \|\mathbf{W}_2\|_2 \|\mathbf{J}_Y\|_2 \|\mathbf{G}_k\|_2, \tag{22}$$

$$\|\boldsymbol{\xi}_{2k}\|_2 \leq \|\mathbf{I}_L \otimes \mathbf{K}_{d_V, d_{ff}} \otimes \mathbf{I}_{d_V}\|_2 \|\mathbf{I}_{Ld_{ff}} \otimes \mathrm{vec}_r(\mathbf{I}_{d_V})\|_2 \|\mathbf{D}_\sigma\|_2 \|\mathbf{I}_L \otimes \mathbf{W}_1^\top\|_2 \|\mathbf{J}_Y\|_2 \|\mathbf{G}_k\|_2$$
$$\leq 1 \cdot \sqrt{d_V} \cdot 1 \cdot \|\mathbf{W}_1\|_2 \cdot \|\mathbf{J}_Y\|_2 \cdot \|\mathbf{G}_k\|_2 = \sqrt{d_V} \|\mathbf{W}_1\|_2 \|\mathbf{J}_Y\|_2 \|\mathbf{G}_k\|_2. \tag{23}$$

B.3 Pure-attention pairs ($k, \ell \in \{K, Q, V\}$):

$$\boldsymbol{\xi}_{k\ell} = \left(\mathbf{J}_{SY} \otimes \mathbf{I}_{n_k}\right)\left[\left(\mathbf{I}_{Ld_V} \otimes \mathbf{G}_k^\top\right)(\mathbf{H}_Y \mathbf{G}_\ell) + \left(\mathbf{J}_Y \otimes \mathbf{I}_{n_k}\right)\boldsymbol{\Phi}_{k\ell}\right].$$

Thus, by Properties 10, 9,

$$\|\boldsymbol{\xi}_{k\ell}\|_2 \leq \|\mathbf{J}_{SY}\|_2 \left(\|\mathbf{I}_{Ld_V} \otimes \mathbf{G}_k^\top\|_2 \|\mathbf{H}_Y\|_2 \|\mathbf{G}_\ell\|_2 + \|\mathbf{J}_Y\|_2 \|\boldsymbol{\Phi}_{k\ell}\|_2\right) = \|\mathbf{J}_{SY}\|_2 \left(\|\mathbf{G}_k\|_2 \|\mathbf{H}_Y\|_2 \|\mathbf{G}_\ell\|_2 + \|\mathbf{J}_Y\|_2 \|\boldsymbol{\Phi}_{k\ell}\|_2\right). \tag{24}$$

C) Substituting into the block estimate equation 8. For each pair $(i,j)$, we substitute the corresponding $\|\boldsymbol{\xi}_{ij}\|_2$ from equation 18–equation 24 and the $\|\mathbf{B}_i\|_2$ from equation 13–equation 15 (with equation 16, equation 17) into

$$\left\|\mathbf{H}_{\mathrm{tr}}^{(i,j)}\right\|_2 \leq \|\mathbf{J}_Z\|_2 \|\boldsymbol{\xi}_{ij}\|_2 + \|\mathbf{B}_i\|_2 \|\mathbf{H}_Z\|_2 \|\mathbf{B}_j\|_2.$$

This yields, for example:

$$\left\|\mathbf{H}_{\mathrm{tr}}^{(1,1)}\right\|_2 \leq \|\mathbf{J}_Z\|_2 \cdot 0 + \|\mathbf{B}_1\|_2^2 \|\mathbf{H}_Z\|_2 \leq \|\mathbf{H}_Z\|_2 \left(\|\mathbf{W}_2\|_2 \|\mathbf{Y}\|_2\right)^2,$$

$$\left\|\mathbf{H}_{\mathrm{tr}}^{(1,2)}\right\|_2 \leq \|\mathbf{J}_Z\|_2 \sqrt{d_V} \|\mathbf{Y}\|_2 + \|\mathbf{H}_Z\|_2 \left(\|\mathbf{W}_2\|_2 \|\mathbf{Y}\|_2\right) \|\sigma(\mathbf{Y}\mathbf{W}_1)\|_2,$$

$$\left\|\mathbf{H}_{\mathrm{tr}}^{(1,k)}\right\|_2 \leq \|\mathbf{J}_Z\|_2 \sqrt{d_{ff}} \|\mathbf{W}_2\|_2 \|\mathbf{J}_Y\|_2 \|\mathbf{G}_k\|_2 + \|\mathbf{H}_Z\|_2 \left(\|\mathbf{W}_2\|_2 \|\mathbf{Y}\|_2\right) \left(\|\mathbf{J}_{SY}\|_2 \|\mathbf{J}_Y\|_2 \|\mathbf{G}_k\|_2\right),$$

$$\left\|\mathbf{H}_{\mathrm{tr}}^{(k,\ell)}\right\|_2 \leq \|\mathbf{J}_Z\|_2 \|\mathbf{J}_{SY}\|_2 \left(\|\mathbf{G}_k\|_2 \|\mathbf{H}_Y\|_2 \|\mathbf{G}_\ell\|_2 + \|\mathbf{J}_Y\|_2 \|\boldsymbol{\Phi}_{k\ell}\|_2\right)$$
$$+ \|\mathbf{H}_Z\|_2 \left(\|\mathbf{J}_{SY}\|_2 \|\mathbf{J}_Y\|_2 \|\mathbf{G}_k\|_2\right) \left(\|\mathbf{J}_{SY}\|_2 \|\mathbf{J}_Y\|_2 \|\mathbf{G}_\ell\|_2\right),$$

etc., where we then use equation 16, equation 17, and the $\|\boldsymbol{\Phi}_{k\ell}\|_2$ bounds above to turn each right-hand side into explicit functions of $L$, $d_V$, $d_{ff}$, $d_K$, and the spectral norms of $\mathbf{X}$ and the weight matrices.

In the estimations above we calculate $\|\mathbf{Y}\|_2$ and $\|\mathbf{S}\|_2$ according to Proposition 10 and both $\mathbf{H}_Z$ and $\mathbf{H}_Y$ can be estimated by Lemma 4 with appropriate inputs and assumptions of $\sigma_{\min}$ and $\sigma'_{\min}$.

$\square$

## C.7 PROOF OF THEOREM 7

*Proof.*

$$|\mathcal{L}_{k+1}(\mathbf{w}) - \mathcal{L}_k(\mathbf{w})| \leq \frac{1}{k+1}\left|l(\mathbf{f}_{\mathbf{w}^*}(\mathbf{x}_{k+1}), \mathbf{y}_{k+1}) - \frac{1}{k}\sum_{i=1}^{k} l(\mathbf{f}_{\mathbf{w}^*}(\mathbf{x}_i), \mathbf{y}_i)\right| +$$

$$+ \frac{1}{2(k+1)} \|\mathbf{w} - \mathbf{w}^*\|_2^2 \left\| \mathbf{H}_{k+1}(\mathbf{w}^*) - \frac{1}{k} \sum_{i=1}^{k} \mathbf{H}_i(\mathbf{w}^*) \right\|_2.$$

**First Term**

The first term is the difference in loss values at the optimal parameters $\mathbf{w}^*$:

$$\left| l(\mathbf{f}_{\mathbf{w}^*}(\mathbf{x}_{k+1}), \mathbf{y}_{k+1}) - \frac{1}{k} \sum_{i=1}^{k} l(\mathbf{f}_{\mathbf{w}^*}(\mathbf{x}_i), \mathbf{y}_i) \right|.$$

Assume the loss function $l(\mathbf{f}_{\mathbf{w}^*}(\mathbf{x}_i), \mathbf{y}_i)$ is bounded, i.e., $0 \leqslant l(\mathbf{f}_{\mathbf{w}^*}(\mathbf{x}_i), \mathbf{y}_i) \leqslant L$, where $L$ is a constant. Then: $- l(\mathbf{f}_{\mathbf{w}^*}(\mathbf{x}_{k+1}), \mathbf{y}_{k+1}) \leqslant L$, $- \frac{1}{k} \sum_{i=1}^{k} l(\mathbf{f}_{\mathbf{w}^*}(\mathbf{x}_i), \mathbf{y}_i) \leqslant L$.

Therefore

$$\left| l(\mathbf{f}_{\mathbf{w}^*}(\mathbf{x}_{k+1}), \mathbf{y}_{k+1}) - \frac{1}{k} \sum_{i=1}^{k} l(\mathbf{f}_{\mathbf{w}^*}(\mathbf{x}_i), \mathbf{y}_i) \right| \leqslant L + L = 2L.$$

Thus, the contribution of the first term is:

$$\frac{1}{k+1} \left| l(\mathbf{f}_{\mathbf{w}^*}(\mathbf{x}_{k+1}), \mathbf{y}_{k+1}) - \frac{1}{k} \sum_{i=1}^{k} l(\mathbf{f}_{\mathbf{w}^*}(\mathbf{x}_i), \mathbf{y}_i) \right| \leqslant \frac{2L}{k+1}.$$

**Second Term**

The second term involves the difference in Hessians:

$$\left\| \mathbf{H}_{k+1}(\mathbf{w}^*) - \frac{1}{k} \sum_{i=1}^{k} \mathbf{H}_i(\mathbf{w}^*) \right\|_2,$$

where $\mathbf{H}_{k+1}(\mathbf{w}^*) = \nabla_{\mathbf{w}}^2 l(\mathbf{f}_{\mathbf{w}^*}(\mathbf{x}_{k+1}), \mathbf{y}_{k+1})$ is the Hessian of the loss for the $(k+1)$-th sample, and $\frac{1}{k} \sum_{i=1}^{k} \mathbf{H}_i(\mathbf{w}^*) = \mathbf{H}_k(\mathbf{w}^*)$ is the Hessian of $\mathcal{L}_k$, the empirical loss over the first $k$ samples.

Rewrite the expression:

$$\mathbf{H}_k(\mathbf{w}^*) = \frac{1}{k} \sum_{i=1}^{k} \mathbf{H}_i(\mathbf{w}^*),$$

$$\mathbf{H}_{k+1}(\mathbf{w}^*) - \mathbf{H}_k(\mathbf{w}^*) = \mathbf{H}_{k+1}(\mathbf{w}^*) - \frac{1}{k} \sum_{i=1}^{k} \mathbf{H}_i(\mathbf{w}^*).$$

Evaluate the norm using the triangle inequality:

$$\left\| \mathbf{H}_{k+1}(\mathbf{w}^*) - \frac{1}{k} \sum_{i=1}^{k} \mathbf{H}_i(\mathbf{w}^*) \right\|_2 \leqslant \|\mathbf{H}_{k+1}(\mathbf{w}^*)\|_2 + \frac{1}{k} \left\| \sum_{i=1}^{k} \mathbf{H}_i(\mathbf{w}^*) \right\|_2.$$

Assume the individual Hessians are bounded, i.e., $\|\mathbf{H}_i(\mathbf{w}^*)\|_2 \leqslant M$ for some constant $M$. Then: $\|\mathbf{H}_{k+1}(\mathbf{w}^*)\|_2 \leqslant M$, $\left\| \sum_{i=1}^{k} \mathbf{H}_i(\mathbf{w}^*) \right\|_2 \leqslant \sum_{i=1}^{k} \|\mathbf{H}_i(\mathbf{w}^*)\|_2 \leqslant kM$.

Thus:

$$\left\|\mathbf{H}_{k+1}(\mathbf{w}^*) - \frac{1}{k}\sum_{i=1}^{k}\mathbf{H}_i(\mathbf{w}^*)\right\|_2 \leqslant M + \frac{1}{k}\cdot kM = M + M = 2M.$$

The contribution of the second term is:

$$\frac{1}{2(k+1)}\|\mathbf{w} - \mathbf{w}^*\|_2^2 \|\mathbf{H}_{k+1}(\mathbf{w}^*) - \mathbf{H}_k(\mathbf{w}^*)\|_2 \leqslant \frac{1}{2(k+1)}\|\mathbf{w} - \mathbf{w}^*\|_2^2 \cdot 2M = \frac{M\|\mathbf{w} - \mathbf{w}^*\|_2^2}{k+1}.$$

Combining both terms:

$$|\mathcal{L}_{k+1}(\mathbf{w}) - \mathcal{L}_k(\mathbf{w})| \leqslant \frac{2L}{k+1} + \frac{M\|\mathbf{w} - \mathbf{w}^*\|_2^2}{k+1}.$$

$\square$

## D  ADDITIONAL THEORETICAL PROPERTIES

**Lemma 2** (Attention second derivatives $\boldsymbol{\Phi}$ from functional Hessian). *Consider single-head scaled dot-product attention*

$$\mathbf{F}(\mathbf{X}) = \mathbf{A}(\mathbf{T})\,\mathbf{X}\mathbf{W}_V, \qquad \mathbf{T} = \frac{1}{\sqrt{d_K}}\,\mathbf{X}\mathbf{W}_Q\mathbf{W}_K^\top\mathbf{X}^\top,$$

*with $\mathbf{X} \in \mathbb{R}^{L \times d_V}$, $\mathbf{W}_Q, \mathbf{W}_K \in \mathbb{R}^{d_V \times d_K}$, $\mathbf{W}_V \in \mathbb{R}^{d_V \times d_V}$. The attention map $\mathbf{A}(\cdot)$ applies row-wise softmax. We use row-wise vectorization $\mathrm{vec}_r(\cdot)$ and the commutation matrices $\mathbf{K}_{m,n}$ from Definition 3.*

*Define the generalized functional Hessian blocks (following Ormaniec et al. (2024) in our $\mathrm{vec}_r$ convention) by*

$$\mathbf{H}_{\mathrm{f}}(\mathbf{W}_i, \mathbf{W}_j) = \left(\tfrac{\partial\ell}{\partial\mathbf{F}} \otimes \mathbf{I}_{p_i q_i}\right)\frac{\partial^2\mathbf{F}}{\partial\mathbf{W}_i\partial\mathbf{W}_j},$$

*where $p_i q_i$ is the size of $\mathbf{W}_i$ (e.g. $p_Q q_Q = d_V d_K$), and $\frac{\partial\ell}{\partial\mathbf{F}} \in \mathbb{R}^{L \times d_V}$ is the loss gradient.*

*Specializing to the squared-error loss $\ell(\mathbf{F}) = \frac{1}{2}\|\mathbf{F} - \mathbf{Target}\|_F^2$, one has $\frac{\partial\ell}{\partial\mathbf{F}} = \mathbf{F} - \mathbf{Target}$ and the row-wise contraction matrix*

$$\mathbf{R}_m := \mathrm{vec}_r\big(\mathbf{F}(\mathbf{X}) - \mathbf{Target}\big)^\top \otimes \mathbf{I}_m \ \in \ \mathbb{R}^{m \times (m \cdot L d_V)}.$$

*Then for $i \in \{V, Q, K\}$ with $n_i := p_i q_i$, the functional Hessian blocks can be factorized as*

$$\mathbf{H}_{\mathrm{f}}(\mathbf{W}_i, \mathbf{W}_j) \ = \ \mathbf{R}_{n_i}\,\boldsymbol{\Phi}_{ij}, \qquad \boldsymbol{\Phi}_{ij} := \frac{\partial^2\mathbf{F}}{\partial\mathbf{W}_i\partial\mathbf{W}_j} \ \in \ \mathbb{R}^{(L d_V \cdot n_i) \times n_j}.$$

*In particular, the model-curvature blocks $\boldsymbol{\Phi}_{ij}$ (to be used in the Transformer Hessian) are obtained from the corresponding expressions in (Ormaniec et al., 2024, Thm. 3.2) by removing the left contraction $\mathbf{R}_{n_i}$.*

*We now list the explicit blocks needed in our derivation. Define the fixed reshaping operator*

$$\mathbf{S} := \big(\mathbf{I}_{d_V} \otimes \mathbf{K}_{d_V, d_V}\big)\big(\mathrm{vec}_r\mathbf{I}_{d_V} \otimes \mathbf{I}_{d_V}\big) \ \in \ \mathbb{R}^{d_V^2 \times d_V},$$

*and the softmax-derivative operators*

$$\mathbf{Z}_1 := (\mathbf{I}_L \otimes \mathbf{X}^\top)(\partial\mathbf{A}/\partial\mathbf{T})(\mathbf{X} \otimes \mathbf{X}) \in \mathbb{R}^{L d_V \times d_V^2}, \quad \mathbf{Z}_2 := \big(\mathbf{I}_L \otimes \mathbf{X}^\top \otimes \mathbf{X}^\top \otimes \mathbf{X}^\top\big)\frac{\partial^2\mathbf{A}}{\partial\mathbf{T}^2}\,(\mathbf{X} \otimes \mathbf{X}) \ \in \ \mathbb{R}^{L d_V^3 \times d_V^2},$$

*where $\frac{\partial^2\mathbf{A}}{\partial\mathbf{T}^2}$ denotes the (row-wise) softmax second derivative tensor arranged compatibly with $\mathrm{vec}_r$ and Kronecker products as above, and $\mathbf{Z}_1$ is the (first-order) softmax derivative linear operator*

*used in Ormaniec et al. (2024) (we keep the exact form as defined there; its size ensures dimensional consistency below).*

*Then the pure attention second derivatives (model curvature) are:*

$$\boldsymbol{\Phi}_{VV} \;=\; \mathbf{0}_{(Ld_V \cdot d_V^2) \times d_V^2},$$

$$\boldsymbol{\Phi}_{QQ} \;=\; \frac{2}{Ld_V d_K} \left( \mathbf{I}_L \otimes \mathbf{W}_V^\top \otimes \mathbf{I}_{d_V} \otimes \mathbf{W}_K^\top \right) \mathbf{Z}_2 \left( \mathbf{I}_{d_V} \otimes \mathbf{W}_K \right) \;\in\; \mathbb{R}^{(Ld_V \cdot d_V d_K) \times d_V d_K},$$

$$\boldsymbol{\Phi}_{VQ} \;=\; \frac{2}{Ld_V \sqrt{d_K}} \left( \mathbf{I}_L \otimes \mathbf{S} \right) \mathbf{Z}_1 \left( \mathbf{I}_{d_V} \otimes \mathbf{W}_K \right) \;\in\; \mathbb{R}^{(Ld_V \cdot d_V^2) \times d_V d_K},$$

$$\boldsymbol{\Phi}_{QK} \;=\; \frac{2}{Ld_V d_K} \left( \mathbf{I}_L \otimes \mathbf{W}_V^\top \otimes \mathbf{I}_{d_V} \otimes \mathbf{W}_K^\top \right) \mathbf{Z}_2 \left( \mathbf{W}_Q \otimes \mathbf{I}_{d_V} \right) \mathbf{K}_{d_K, d_V}$$

$$+ \; \frac{2}{Ld_V \sqrt{d_K}} \left( \mathbf{I}_{d_V} \otimes \mathbf{W}_V^\top \otimes \mathbf{I}_{d_V} \right) \left( \mathbf{Z}_1 \otimes \mathbf{I}_{d_V} \right) \mathbf{S} \otimes \mathbf{I}_{d_K} \;\in\; \mathbb{R}^{(Ld_V \cdot d_V d_K) \times d_V d_K}.$$

*Moreover, by symmetry of second derivatives, $\boldsymbol{\Phi}_{KQ}$ equals $\boldsymbol{\Phi}_{QK}$ with $\mathbf{W}_Q, \mathbf{W}_K$ swapped and commutation adjusted by $\mathbf{K}_{\cdot,\cdot}$ (Definition 3). Analogous symmetric relations give $\boldsymbol{\Phi}_{QV}$ and $\boldsymbol{\Phi}_{KV}$ from $\boldsymbol{\Phi}_{VQ}$.*

*Proof.* By definition of the generalized functional Hessian in Ormaniec et al. (2024),

$$\mathbf{H}_{\mathrm{f}}(\mathbf{W}_i, \mathbf{W}_j) = \left( \tfrac{\partial \ell}{\partial \mathbf{F}} \otimes \mathbf{I}_{p_i q_i} \right) \frac{\partial^2 \mathbf{F}}{\partial \mathbf{W}_i \partial \mathbf{W}_j}.$$

For squared-error loss, $\frac{\partial \ell}{\partial \mathbf{F}}$ yields the contraction $\mathbf{R}_{p_i q_i}$ defined above; hence $\mathbf{H}_{\mathrm{f}}(\mathbf{W}_i, \mathbf{W}_j) = \mathbf{R}_{n_i} \boldsymbol{\Phi}_{ij}$ with $\boldsymbol{\Phi}_{ij} = \frac{\partial^2 \mathbf{F}}{\partial \mathbf{W}_i \partial \mathbf{W}_j}$. The explicit forms for $\mathbf{H}_{\mathrm{f}}$ in (Ormaniec et al., 2024, Thm. 3.2) then imply the above formulas for $\boldsymbol{\Phi}_{ij}$ by simply removing the leading contraction $\mathbf{R}_{n_i}$. $\qquad\square$

**Lemma 3** (ReLU derivative and Hessian). *Let $\mathbf{X} \in \mathbb{R}^{m \times n}$, almost everywhere the following holds:*

$$\frac{\partial \mathrm{ReLU}(\mathbf{X})}{\partial \mathbf{X}} = \mathrm{diag}\big(\mathrm{vec}_r(\mathbf{1}_{\{\mathbf{X}>0\}})\big), \quad \frac{\partial^2 \mathrm{ReLU}(\mathbf{X})}{\partial \mathbf{X}^2} = \mathbf{0}.$$

*Proof.* We start with the elementwise definition of the ReLU function:

$$\mathrm{ReLU}(x) = \max(0, x).$$

Thus, for each entry $x_{ij}$ of $\mathbf{X} \in \mathbb{R}^{m \times n}$, we have

$$\frac{\partial \mathrm{ReLU}(x_{ij})}{\partial x_{ij}} = \begin{cases} 1 & \text{if } x_{ij} > 0, \\ 0 & \text{if } x_{ij} < 0, \\ \text{undefined (subgradient in } [0,1]) & \text{if } x_{ij} = 0. \end{cases}$$

For the scalar case $x \in \mathbb{R}$, the nondifferentiable set is $\{0\}$, which is a measure-zero subset of $\mathbb{R}$. For the matrix case, we identify $\mathbf{X} \in \mathbb{R}^{m \times n}$ with a point in $\mathbb{R}^{mn}$. The nondifferentiable set is

$$\mathcal{N} = \bigcup_{i,j} \{\mathbf{X} \in \mathbb{R}^{m \times n} : x_{ij} = 0\}.$$

Each set $\{x_{ij} = 0\}$ is a hyperplane of codimension 1 in $\mathbb{R}^{mn}$, and therefore has Lebesgue measure zero. Since $\mathcal{N}$ is a finite union of such hyperplanes, $\mathcal{N}$ also has measure zero. Thus, ReLU is differentiable almost everywhere in $\mathbb{R}^{m \times n}$.

At differentiable points ($\mathbf{X} \notin \mathcal{N}$), applying row-wise vectorization and the identification theorem from Proposition 1 yields

$$\mathrm{vec}_r(d\,\mathrm{ReLU}(\mathbf{X})) = \mathrm{diag}(\mathrm{vec}_r(\mathbf{1}_{\{\mathbf{X}>0\}})) \, \mathrm{vec}_r(d\mathbf{X}),$$

using Property 3 for the indicator matrix treated as a Hadamard multiplier and Property 6 for the diagonal form. Therefore,

$$\frac{\partial \text{ReLU}(\mathbf{X})}{\partial \mathbf{X}} = \text{diag}\big(\text{vec}_r(\mathbf{1}_{\{\mathbf{X}>0\}})\big).$$

Since the Jacobian is piecewise constant (its entries depend only on the sign of $x_{ij}$), its differential vanishes almost everywhere:

$$d\left(\frac{\partial \text{ReLU}(\mathbf{X})}{\partial \mathbf{X}}\right) = \mathbf{0}, \qquad \mathbf{X} \notin \mathcal{N}.$$

Hence the Hessian is zero almost everywhere:

$$\frac{\partial^2 \text{ReLU}(\mathbf{X})}{\partial \mathbf{X}^2} = \mathbf{0}.$$

This completes the proof. □

**Proposition 10** (Spectral-norm estimates for $\mathbf{Y}$ and $\mathbf{S} = \mathbf{Y} + \text{FFN}(\mathbf{Y})$). *Let $\mathbf{X} \in \mathbb{R}^{L \times d_V}$, $\mathbf{Y} = \text{LayerNorm}(\mathbf{F}(\mathbf{X}) + \mathbf{X}) \in \mathbb{R}^{L \times d_V}$ and*

$$\text{FFN}(\mathbf{Y}) = \sigma(\mathbf{Y}\mathbf{W}_1)\mathbf{W}_2, \qquad \mathbf{W}_1 \in \mathbb{R}^{d_V \times d_{ff}}, \quad \mathbf{W}_2 \in \mathbb{R}^{d_{ff} \times d_V},$$

*and set $\mathbf{S} = \mathbf{Y} + \text{FFN}(\mathbf{Y}) \in \mathbb{R}^{L \times d_V}$. Then the following spectral-norm bounds hold:*

$$\|\mathbf{Y}\|_2 \ \leq \ \|\mathbf{Y}\|_F \ = \ \sqrt{L\, d_V}, \tag{25}$$

$$\|\text{FFN}(\mathbf{Y})\|_2 \ \leq \ \sqrt{\min(L, d_{ff})}\, \|\mathbf{Y}\|_2\, \|\mathbf{W}_1\|_2\, \|\mathbf{W}_2\|_2, \tag{26}$$

$$\|\mathbf{S}\|_2 \ \leq \ \|\mathbf{Y}\|_2 + \|\text{FFN}(\mathbf{Y})\|_2 \ \leq \ \sqrt{L\, d_V}\left(1 + \sqrt{\min(L, d_{ff})}\, \|\mathbf{W}_1\|_2\, \|\mathbf{W}_2\|_2\right). \tag{27}$$

*Proof.* We proceed using only the properties stated in the preliminaries.

1) Bound for $\|\mathbf{Y}\|_2$. By the LayerNorm definition (Theorem 2), write

$$\mathbf{Y} \ = \ \mathbf{P}(\mathbf{S}_0)\, \mathbf{M}(\mathbf{S}_0), \qquad \mathbf{S}_0 := \mathbf{F}(\mathbf{X}) + \mathbf{X},$$

where $\mathbf{M}(\mathbf{S}_0) = \mathbf{S}_0 - \frac{1}{d_V}\mathbf{S}_0\mathbf{1}_{d_V}\mathbf{1}_{d_V}^\top$ and $\mathbf{P} = \text{diag}^{-1}(\sigma)$ with $\sigma = \frac{1}{\sqrt{d_V}}(\mathbf{M}^{\circ 2}\mathbf{1})^{\circ 1/2}$ applied row-wise. For any row $i$, denote $\mathbf{m}_i$ the $i$-th row of $\mathbf{M}$ and $\sigma_i = \frac{1}{\sqrt{d_V}}\|\mathbf{m}_i\|_2$. Then the $i$-th row of $\mathbf{Y}$ is $\mathbf{y}_i = \mathbf{m}_i/\sigma_i$, so

$$\|\mathbf{y}_i\|_2^2 \ = \ \frac{\|\mathbf{m}_i\|_2^2}{\sigma_i^2} \ = \ \frac{\|\mathbf{m}_i\|_2^2}{(1/d_V)\|\mathbf{m}_i\|_2^2} \ = \ d_V.$$

Hence every row of $\mathbf{Y}$ has Euclidean norm $\sqrt{d_V}$. Therefore,

$$\|\mathbf{Y}\|_F^2 = \sum_{i=1}^{L} \|\mathbf{y}_i\|_2^2 = L\, d_V, \qquad \text{so} \qquad \|\mathbf{Y}\|_F = \sqrt{L\, d_V}.$$

By the norm inequality $\|\mathbf{A}\|_2 \leq \|\mathbf{A}\|_F$ (Property 7), we obtain equation 25.

2) Bound for $\|\text{FFN}(\mathbf{Y})\|_2$. We estimate step-by-step using only matrix norm properties.

First,

$$\|\text{FFN}(\mathbf{Y})\|_2 = \|\text{ReLU}(\mathbf{Y}\mathbf{W}_1)\mathbf{W}_2\|_2 \ \leq \ \|\text{ReLU}(\mathbf{Y}\mathbf{W}_1)\|_2\, \|\mathbf{W}_2\|_2 \qquad \text{(Property 10)}.$$

Next, use $\|\cdot\|_2 \leq \|\cdot\|_F$ (Property 7) to get

$$\|\text{ReLU}(\mathbf{Y}\mathbf{W}_1)\|_2 \leq \|\text{ReLU}(\mathbf{Y}\mathbf{W}_1)\|_F.$$

By Definition 1, $\|\cdot\|_F^2$ is the sum of squares. Entrywise $\sigma(\cdot)$ satisfies $0 \leq \sigma(a) \leq |a|$, hence $\sigma(a)^2 \leq a^2$ for each entry $a \in \mathbb{R}$. Therefore,

$$\|\sigma(\mathbf{Y}\mathbf{W}_1)\|_F \leq \|\mathbf{Y}\mathbf{W}_1\|_F.$$

Using the inequality $\| \cdot \|_F \leq \sqrt{d}\, \| \cdot \|_2$ with $d = \mathrm{rank}(\cdot)$ from Property 7 (row $X = \| \cdot \|_F$, column $Y = \| \cdot \|_2$), we obtain

$$\|\mathbf{YW}_1\|_F \leq \sqrt{\mathrm{rank}(\mathbf{YW}_1)}\, \|\mathbf{YW}_1\|_2.$$

Since $\mathbf{YW}_1 \in \mathbb{R}^{L \times d_{ff}}$, $\mathrm{rank}(\mathbf{YW}_1) \leq \min(L, d_{ff})$. Thus

$$\|\mathbf{YW}_1\|_F \leq \sqrt{\min(L, d_{ff})}\, \|\mathbf{YW}_1\|_2 \leq \sqrt{\min(L, d_{ff})}\, \|\mathbf{Y}\|_2\, \|\mathbf{W}_1\|_2 \qquad \text{(Property 10).}$$

Collecting,

$$\|\mathrm{FFN}(\mathbf{Y})\|_2 \leq \|\sigma(\mathbf{YW}_1)\|_F\, \|\mathbf{W}_2\|_2 \leq \sqrt{\min(L, d_{ff})}\, \|\mathbf{Y}\|_2\, \|\mathbf{W}_1\|_2\, \|\mathbf{W}_2\|_2,$$

which is equation 26.

3) Bound for $\|\mathbf{S}\|_2$. By the sum-norm inequality (Property 8),

$$\|\mathbf{S}\|_2 = \|\mathbf{Y} + \mathrm{FFN}(\mathbf{Y})\|_2 \leq \|\mathbf{Y}\|_2 + \|\mathrm{FFN}(\mathbf{Y})\|_2.$$

Substituting equation 25 and equation 26 yields equation 27. $\qquad \square$

**Lemma 4** (LayerNorm derivative and Hessian norm estimation). *Let $\mathbf{X} \in \mathbb{R}^{m \times n}$. Layer-Norm derivative $\mathbf{J}_{\mathrm{LN}}(\mathbf{X}) = \frac{\partial LayerNorm(\mathbf{X})}{\partial \mathbf{X}}$ is calculated according to Theorem 2 and its Hessian $\mathbf{H}_{\mathrm{LN}}(\mathbf{X}) = \frac{\partial^2 LayerNorm(\mathbf{X})}{\partial \mathbf{X}^2}$ is calculated as in Theorem 3. Then, the following estimation holds:*

$$\left\|\mathbf{J}_{\mathrm{LN}}(\mathbf{X})\right\|_2 \leq \frac{1}{\sigma_{\min}} + \frac{\|\mathbf{X}\|_2^2}{\sqrt{n}\,\sigma_{\min}^3}, \tag{28}$$

$$\left\|\mathbf{H}_{\mathrm{LN}}(\mathbf{X})\right\|_2 \leq \frac{\|\mathbf{X}\|_2}{\sigma_{\min}^3}\left(1 + \sqrt{\tfrac{m}{n}}\right) + \frac{\|\mathbf{X}\|_2^2}{\sqrt{n}\,\sigma_{\min}^3} + \frac{3\,\|\mathbf{X}\|_2^3}{n\,\sigma_{\min}^5}. \tag{29}$$

*where $\sigma_{\min}$ denotes $\min_i \|\mathbf{M}_i\|_2$, where $\mathbf{M}(\mathbf{X}) = \mathbf{X}\left(\mathbf{I}_n - \frac{1}{n}\mathbf{1}_n\mathbf{1}_n^\top\right)$*

*Proof.* We rely only on the properties established in the preliminaries and on Theorems 2–3.

1) LayerNorm Jacobian structure and bound. By Theorem 2 (with $L \to m$, $d_V \to n$),

$$\mathbf{J}_{\mathrm{LN}}(\mathbf{X}) = (\mathbf{P} \otimes \mathbf{I}_n)\,\mathbf{G} + (\mathbf{I}_m \otimes \mathbf{M}^\top)\,\mathbf{H},$$

where $\mathbf{G} = \mathbf{I}_{mn} - \frac{1}{n}(\mathbf{I}_m \otimes \mathbf{1}_{n \times n})$, $\mathbf{H} = \frac{\partial \mathbf{P}}{\partial \mathbf{X}}$, and $\mathbf{P} = \mathrm{diag}^{-1}(\boldsymbol{\sigma})$. Using Properties 9, 10, 8,

$$\|\mathbf{J}_{\mathrm{LN}}(\mathbf{X})\|_2 \leq \|\mathbf{P} \otimes \mathbf{I}_n\|_2\, \|\mathbf{G}\|_2 + \|\mathbf{I}_m \otimes \mathbf{M}^\top\|_2\, \|\mathbf{H}\|_2 = \|\mathbf{P}\|_2\, \|\mathbf{G}\|_2 + \|\mathbf{M}\|_2\, \|\mathbf{H}\|_2.$$

We now bound each factor:

- $\|\mathbf{G}\|_2 \leq 1$ since $\frac{1}{n}\mathbf{1}_{n \times n}$ is a projection, hence $\|\mathbf{I}_n - \frac{1}{n}\mathbf{1}_{n \times n}\|_2 \leq 1$ and Kronecker preserves the spectral norm bound (Properties 10, 9, Proposition 2).

- $\|\mathbf{P}\|_2 = \|\mathbf{D}^{-1}\|_2 = 1/\sigma_{\min}$, where $\mathbf{D} = \mathrm{diag}(\boldsymbol{\sigma})$.

- $\|\mathbf{M}\|_2 \leq \|\mathbf{X}\|_2$, because $\mathbf{M}(\mathbf{X}) = \mathbf{X}\left(\mathbf{I}_n - \frac{1}{n}\mathbf{1}_n\mathbf{1}_n^\top\right)$ and the right factor is a projector with norm $\leq 1$ (Property 10).

- For $\|\mathbf{H}\|_2 = \left\|\frac{\partial \mathbf{P}}{\partial \mathbf{X}}\right\|_2$, Theorem 2 plus Propositions 5, 6, 7, 8 and Properties 10, 9 give (see the same chain as in Theorem 2):

$$\left\|\frac{\partial \mathbf{P}}{\partial \mathbf{X}}\right\|_2 \leq \frac{1}{\sqrt{n}}\, \|\mathbf{D}^{-1} \otimes \mathbf{D}^{-\top}\|_2 \left\|\mathrm{diag}^{-1}\!\big(\mathrm{vec}_r^{\circ 1/2}(\mathbf{M}^{\circ 2}\mathbf{1}_n)\big)\right\|_2 \|\mathbf{I}_m \otimes \mathbf{1}_n^\top\|_2\, \|\mathrm{diag}(\mathrm{vec}_r(\mathbf{M}))\|_2 \left\|\frac{\partial \mathbf{M}}{\partial \mathbf{X}}\right\|_2.$$

Using $\|\mathbf{D}^{-1} \otimes \mathbf{D}^{-\top}\|_2 = \|\mathbf{D}^{-1}\|_2^2 = \frac{1}{\sigma_{\min}^2}$, $\left\|\mathrm{diag}^{-1}(\cdot)\right\|_2 = \frac{1}{\min_i \sqrt{\sum_v M_{i,v}^2}} = \frac{1}{\sqrt{n}\,\sigma_{\min}}$,

$\|\mathbf{I}_m \otimes \mathbf{1}^\top\|_2 = \sqrt{n}$, $\|\mathrm{diag}(\mathrm{vec}_r(\mathbf{M}))\|_2 = \|\mathbf{M}\|_{\max} \leq \|\mathbf{M}\|_2$ (Property 7), and $\left\|\frac{\partial \mathbf{M}}{\partial \mathbf{X}}\right\|_2 \leq 1$ (projection), we obtain

$$\|\mathbf{H}\|_2 \leq \frac{1}{\sqrt{n}\sigma_{\min}^2} \cdot \frac{1}{\sqrt{n}\,\sigma_{\min}} \cdot \sqrt{n} \cdot \|\mathbf{M}\|_2 \cdot 1 \leq \frac{\|\mathbf{X}\|_2}{\sqrt{n}\,\sigma_{\min}^3}.$$

Collecting the bounds gives equation 28:

$$\|\mathbf{J}_{\mathrm{LN}}(\mathbf{X})\|_2 \le \frac{1}{\sigma_{\min}} \cdot 1 + \|\mathbf{X}\|_2 \cdot \frac{\|\mathbf{X}\|_2}{\sqrt{n}\,\sigma_{\min}^3} = \frac{1}{\sigma_{\min}} + \frac{\|\mathbf{X}\|_2^2}{\sqrt{n}\,\sigma_{\min}^3}.$$

2) LayerNorm Hessian structure and bound. From Theorem 3 (with $m, n$), using $\frac{\partial^2 \mathbf{M}}{\partial \mathbf{X}^2} = 0$,

$$\mathbf{H}_{\mathrm{LN}}(\mathbf{X}) = (\mathbf{I}_{mn} \otimes \mathbf{G}^\top) \frac{\partial(\mathbf{P} \otimes \mathbf{I}_n)}{\partial \mathbf{X}} + \big((\mathbf{I}_m \otimes \mathbf{M}^\top) \otimes \mathbf{I}_{mn}\big) \frac{\partial^2 \mathbf{P}}{\partial \mathbf{X}^2} + (\mathbf{I}_{mn} \otimes \mathbf{H}^\top) \frac{\partial(\mathbf{I}_m \otimes \mathbf{M}^\top)}{\partial \mathbf{X}}.$$

We bound the three terms separately with Properties 10, 9.

(i) First term. By Proposition 6,

$$\frac{\partial(\mathbf{P} \otimes \mathbf{I}_n)}{\partial \mathbf{X}} = (\mathbf{I}_m \otimes \mathbf{K}_{n,m} \otimes \mathbf{I}_n)(\mathbf{I}_{m^2} \otimes \mathrm{vec}_r(\mathbf{I}_n)) \frac{\partial \mathbf{P}}{\partial \mathbf{X}},$$

therefore

$$\left\| (\mathbf{I}_{mn} \otimes \mathbf{G}^\top) \frac{\partial(\mathbf{P} \otimes \mathbf{I}_n)}{\partial \mathbf{X}} \right\|_2 \le \|\mathbf{G}\|_2 \, \|\mathbf{I}_{m^2} \otimes \mathrm{vec}_r(\mathbf{I}_n)\|_2 \left\| \frac{\partial \mathbf{P}}{\partial \mathbf{X}} \right\|_2 = 1 \cdot \sqrt{n} \cdot \frac{\|\mathbf{X}\|_2}{\sqrt{n}\,\sigma_{\min}^3} = \frac{\|\mathbf{X}\|_2}{\sigma_{\min}^3}.$$

(ii) Second term. Using $\|\mathbf{I}_m \otimes \mathbf{M}^\top\|_2 = \|\mathbf{M}\|_2 \le \|\mathbf{X}\|_2$ and the bound below for $\left\| \frac{\partial^2 \mathbf{P}}{\partial \mathbf{X}^2} \right\|_2$,

$$\left\| \big((\mathbf{I}_m \otimes \mathbf{M}^\top) \otimes \mathbf{I}_{mn}\big) \frac{\partial^2 \mathbf{P}}{\partial \mathbf{X}^2} \right\|_2 \le \|\mathbf{X}\|_2 \left\| \frac{\partial^2 \mathbf{P}}{\partial \mathbf{X}^2} \right\|_2.$$

We now bound $\left\| \frac{\partial^2 \mathbf{P}}{\partial \mathbf{X}^2} \right\|_2$ following the same chain as in the proof of Theorem 3: write $\frac{\partial \mathbf{P}}{\partial \mathbf{X}} = \frac{1}{\sqrt{n}} \mathbf{A}_1(\mathbf{X})\, \mathbf{E}\, \mathbf{B}_1(\mathbf{X})$ and differentiate using Property 10, while bounding the factors with Propositions 5, 6, 7, 8 and Properties 10, 9, 7. This yields

$$\left\| \frac{\partial^2 \mathbf{P}}{\partial \mathbf{X}^2} \right\|_2 \le \frac{1}{\sqrt{n}\,\sigma_{\min}^3} \|\mathbf{X}\|_2 + \frac{3}{n\,\sigma_{\min}^5} \|\mathbf{X}\|_2^2.$$

Therefore,

$$\left\| \big((\mathbf{I}_m \otimes \mathbf{M}^\top) \otimes \mathbf{I}_{mn}\big) \frac{\partial^2 \mathbf{P}}{\partial \mathbf{X}^2} \right\|_2 \le \frac{\|\mathbf{X}\|_2^2}{\sqrt{n}\,\sigma_{\min}^3} + \frac{3\,\|\mathbf{X}\|_2^3}{n\,\sigma_{\min}^5}.$$

(iii) Third term. By Proposition 6 and Proposition 9,

$$\frac{\partial(\mathbf{I}_m \otimes \mathbf{M}^\top)}{\partial \mathbf{X}} = (\mathbf{I}_m \otimes \mathbf{K}_{n,m} \otimes \mathbf{I}_m)(\mathrm{vec}_r(\mathbf{I}_m) \otimes \mathbf{I}_{mn}) \frac{\partial \mathbf{M}}{\partial \mathbf{X}},$$

so

$$\left\| (\mathbf{I}_{mn} \otimes \mathbf{H}^\top) \frac{\partial(\mathbf{I}_m \otimes \mathbf{M}^\top)}{\partial \mathbf{X}} \right\|_2 \le \|\mathbf{H}\|_2 \, \|\mathrm{vec}_r(\mathbf{I}_m) \otimes \mathbf{I}_{mn}\|_2 \left\| \frac{\partial \mathbf{M}}{\partial \mathbf{X}} \right\|_2 = \frac{\|\mathbf{X}\|_2}{\sqrt{n}\,\sigma_{\min}^3} \cdot \sqrt{m} \cdot 1 = \frac{\sqrt{m}}{\sqrt{n}} \frac{\|\mathbf{X}\|_2}{\sigma_{\min}^3}.$$

Summing (i)(iii) with Property 8 yields equation 29:

$$\|\mathbf{H}_{\mathrm{LN}}(\mathbf{X})\|_2 \le \frac{\|\mathbf{X}\|_2}{\sigma_{\min}^3} + \left( \frac{\|\mathbf{X}\|_2^2}{\sqrt{n}\,\sigma_{\min}^3} + \frac{3\,\|\mathbf{X}\|_2^3}{n\,\sigma_{\min}^5} \right) + \frac{\sqrt{m}}{\sqrt{n}} \frac{\|\mathbf{X}\|_2}{\sigma_{\min}^3} = \frac{\|\mathbf{X}\|_2}{\sigma_{\min}^3} \left( 1 + \sqrt{\tfrac{m}{n}} \right) + \frac{\|\mathbf{X}\|_2^2}{\sqrt{n}\,\sigma_{\min}^3} + \frac{3\,\|\mathbf{X}\|_2^3}{n\,\sigma_{\min}^5}.$$

This completes the proof. $\square$

# E   THE USE OF LARGE LANGUAGE MODELS (LLMs)

In the course of this work, a Large Language Model (LLM) served as a general-purpose assistant for text drafting and coding tasks. Its application facilitated the initial generation of code snippets and the formulation and subsequent simplification of natural language explanations to ensure smooth reading. Every piece of content produced with LLM assistance underwent careful scrutiny, editing, and validation by the authors to guarantee its correctness and originality. The authors bear sole responsibility for all material presented herein.

