# OpenReview forum: "Closing the Curvature Gap: Full Transformer Hessians and Their Implications for Scaling Laws"
_ICLR.cc/2026/Conference — ICLR 2026 Conference Desk Rejected Submission_

### Official Review · Reviewer_4hPT · 2025-10-30

**Soundness:** 3
**Presentation:** 2
**Contribution:** 1
**Rating:** 2
**Confidence:** 4

**Summary:**

The paper derives Hessian expressions for full Transformer blocks, extending prior analyses beyond self-attention to include residual connections, LayerNorm, and feed-forward components. It also include a bound on the loss evolution as a function of the number of samples. The authors support their theory empirically using Vision Transformers (ViT).

**Strengths:**

- Addresses a gap in theoretical understanding of Transformer curvature by incorporating LayerNorm and MLP components.
- Provides mathematical derivations of the Hessian for the full block.
- Given the growing interests on curvature-aware optimization and why Adam (or other adaptive optimizers) works in Transformers, the topic is timely could provide comprehensive insights by studying the full block.

**Weaknesses:**

### Weaknesses
- **Lack of intuition**: While the derivations are detailed, the paper does not sufficiently discuss intuition or implications. For example, how does adding LayerNorm change the Hessian compared to Ormaniec et al.? What are the implications for trainability or curvature?
- **Positioning and impact**: Without a stronger conceptual narrative, the work risks feeling incremental. Clearer articulation of why these results matter in practice (e.g., implications for training dynamics, scaling, or optimizer design) would strengthen the contribution. It is also unclear what the implications are for compute-optimal scaling, batch size scaling, etc, which are claimed in the introduction. In particular, it is unclear how theorem 7, which bounds the difference of the losses from $k$ to $k+1$. On Theorem 7, it also unclear what the bound measures, how it ties with the literature, what is the effect of the architectural components, etc..
- **Empirical validation**: Figures visualizing Hessian structure do not include theoretical predictions, making it hard to assess agreement. The experiments mostly confirm block-wise heterogeneity, which is known in the cited relevant literature.
- **Exposition clarity**: Some notation (e.g., `k`) is not defined upon first use. Section 4 and Theorem 5 are difficult to follow — improved structure and guiding commentary would help substantially.


### Overall Assessment
This work tackles an important problem and offers some theoretical contributions. However, the exposition and discussion currently fall short of conveying intuition, significance, and practical implications. I would suggest significantly strengthening the conceptual narrative and more directly validating theoretical predictions.

**Questions:**

- Can you provide intuition for how LayerNorm changes curvature propagation relative to prior work? How does it influence the formula in Theorem 5? Is there any interesting interaction with attention / MLP block?

---

### Official Review · Reviewer_Tc4P · 2025-10-31

**Soundness:** 2
**Presentation:** 1
**Contribution:** 2
**Rating:** 4
**Confidence:** 4

**Summary:**

The paper theoretically studies the loss Hessian of a full Transformer block, by deriving closed-form expressions for the loss Hessian of a model consisting of a single Transformer block. It also derives bounds on the largest eigenvalues/singular values of the loss Hessian blocks as well as the whole loss Hessian. It then uses the derived bounds to derive a convergence bound of the loss with an increasing dataset size.

**Strengths:**

* The paper significantly extends previous results on the closed-form neural network loss Hessian, deriving the exact structure of the loss Hessian for a whole Transformer block.
* The theoretical analysis is non-trivial and the Hessian structure itself can be of interest to researcher in the deep learning community, including those interested in optimization, pruning, loss landscape analysis, etc.

**Weaknesses:**

1. In the contributions section, the paper claims to provide bounds on the loss landscape evolution with dataset size (theorem 7). It is not clear what the authors mean by dataset size. Theorem 7 seems to be referring to the sequence length and the Hessian bounds are derived for a single sequence of fixed length. At the same time the experiment from Figure 6 concerns growing the dataset by adding a batch of sequences.
2. Using the term "scaling laws" in the title suggests that the paper draws a tight connection between the Hessian and the scaling laws. Yet the paper touches on the connection to scaling laws only in the conclusion.
3. The proofs partially rely on the uniform attention assumption which is not discussed in the text and I believe it’s wrongly used in the reasoning. I also think that by performing a few additional calculations this assumption could be removed from the analysis completely. Specifically:
    1. In the proof of the bound (which theorem) the uniform attention assumption is invoked, yet it is not mentioned anywhere in the text. Ultimately, the result is used to provide a bound at convergence when uniform attention is quite unlikely. This should be discussed in the text.
    2. In the proof of the second derivative matrix of the full Transformer block in lines 1598-1602 the authors claim that under uniform attention assumption the mixed second derivatives of the attention wrt to the queries and keys disappear. I would argue this is not true. The uniform attention assumption from [1] means that we analyse the system at a point where the attention scores attain uniform values, not that the attention or its derivative do not depend on the query and key weight matrices. So instead first assuming the uniform attention in the formulas for the first derivative and then computing the second derivative (which indeed gives the derivative 0) the authors should first compute the second derivative (as already done in previous work [2]) and only then substitute the uniform attention scores.
4. Experiments :
    1. There is no experimental verification of how tight the obtained upper bounds for the Hessian norm are neither in real nor synthetic systems. Here I mean a comparison of what the bound would predict and what are the actual values observed in the Hessian (e.g. at init and at convergence).
    2. It's not clear what is the purpose of presenting the loss Hessian in Figures 3 and 4. What insights should I gain other than that the magnitudes and the structure are heterogenous (which has already been established in [2])? How does this experiment tie back to the derived expressions and bounds exactly? Also the authors say “We train the model for a number of epochs, obtaining pretty high accuracy on a validation dataset (>50%)”. What does “pretty high” mean here? 50% on MNIST is not a lot.
5. The presentation of the paper should be improved. Specifically:
    1. The paper states many (potentially useful) expressions and bounds, yet there is hardly any interpretation of these bounds. What insights do we gain from these expressions and bounds? If the ultimate goal is to bind the convergence of the loss in terms of the dataset size using the Hessian (Theorem 7), I would expect to see some discussion on how these bounds depend on the sequence length and batch size.
   2. The writing could be improved: there are spaces and interpunction signs missing, grammatical errors, citations do not use \citep, etc.
   3. Figure 5 does not have labels on the axes. When the Hessian is plotted it is not clear from the figures which parts of it correspond to which parameters.

Minor issues:

* Some references seem to be broken. They are either placeholder references that say “Placeholder - replace with actual entry” or they have some additional information that aren’t usually part of the reference, like “Empirical analysis of large-batch training dynamics.” in line 556 or “ Self-Attention Block decomposition” in line 523.
* The paper derives a few second derivatives of the network components wrt to their inputs and calls them “Hessians”. Technically a Hessian is a second derivative of a scalar function (see [3] for example). Here the authors compute second derivatives of vector or matrix-values functions, so I would think of a different name, or just referred to it as a matrix of second derivatives (as in previous work).
* In a few places there is a space missing, e.g. lines 20, 60, 61
* Please use \citep where necessary, otherwise the sentence flow is broken frequently and it’s hard to read
Inconsistently capitalising transformer/Transformer
* Reusing the letter $H$ for the derivative of $P$ makes the statement a bit confusing, as $H$ is used before to denote the Hessian. Same for the loss bound $L$ in theorem 7, when it was used before to denote sequence length.
* Double “where” in line 250
* I assume in Theorem 7 the authors assume that the loss is bounded **at the minimum** and not in general. Otherwise the upper bound could easily be made tighter. The inequality reflects this assumption but the theorem says “the loss function is bounded” instead of “the loss function at the minimum is bounded”.

[1] Noci et al., Signal Propagation in Transformers: Theoretical Perspectives and the Role of Rank Collapse

[2] Ormaniec et al., What Does It Mean to Be a Transformer? Insights from a Theoretical Hessian Analysis

[3] Callahan, James J. (2010). Advanced Calculus: A Geometric View. Springer Science & Business Media.

**Questions:**

Other than the questions from the “Weaknesses” sections I would like to learn/clarify the following:
1. There has been some evidence that the loss landscape along the layer norm parameters is special in a sense that it really requires adaptivity [4] in the optimizer. Can the authors provide some explanation of that phenomenon through the derived loss Hessian expressions?
2. Can the authors comment on and demonstrate experimentally how tight the upper bounds on spectral norms are?
3. In lines 988-995 the authors are assuming that the attention scores are uniform without stating that in the theorem statement. Is this assumption necessary? Can the authors state the same bound for this expression for arbitrary attention scores instead (see Remark 3.1 in [2])? I would expect it is possible, but if not, the assumption should be explicitly stated in the theorem.
4. Can the authors elaborate more or provide some experimental results on the connection between their Hessian results (expressions and theorem 7) and the scaling laws?


[2] Ormaniec et al., What Does It Mean to Be a Transformer? Insights from a Theoretical Hessian Analysis

[4] Zhao et al., Deconstructing What Makes a Good Optimizer for Autoregressive Language Models

---

### Official Review · Reviewer_3Jav · 2025-10-31

**Soundness:** 2
**Presentation:** 2
**Contribution:** 2
**Rating:** 2
**Confidence:** 3

**Summary:**

This paper presents a theoretical study of Transformer optimization landscapes, aiming to fill the gap left by the lack of second-order (Hessian) analyses for Layer Normalization and feed-forward networks. The authors derive explicit Hessian expressions for these components to complete the curvature characterization of full Transformer blocks and claim that their framework provides insights into convergence dynamics, scaling laws, and loss landscape evolution.  Empirically, the paper evaluates its theoretical predictions using Vision Transformers trained on standard datasets. The experiments analyze block-wise Hessian magnitudes and show that the observed curvature patterns align with the theoretical expectations. The results also indicate that the loss landscape becomes more stable as dataset size increases, consistent with the derived convergence inequality based on a second-order Taylor expansion.

**Strengths:**

1. The paper tackles an important theoretical gap by deriving explicit second-order (Hessian) formulations for Layer Normalization and feed-forward components, extending prior analyses limited to self-attention.
2. The mathematical derivations are well-structured, providing a framework for understanding curvature propagation and its connection to optimization dynamics and scaling laws.
3. The empirical results on Vision Transformers, though limited in scope, are consistent with the theoretical predictions.

**Weaknesses:**

* The experiments are limited to Vision Transformers, with no validation on large language models or multimodal architectures, which are far more relevant for understanding Transformer optimization in practice. While the topic is theoretically relevant, the contribution is incremental and the experimental validation is minimal, offering limited support for the broader theoretical claims.

* The analysis focuses only on the mean-squared error loss, whereas most Transformer models use causal or cross-entropy losses, limiting the applicability of the theoretical results.
* The practical value of the work is unclear, as the paper does not provide actionable insights or implications for improving training or optimization strategies.
* The writing quality needs improvement, with numerous typos, missing punctuation, and formatting inconsistencies that hinder readability.

**Questions:**

1. What is the purpose of Figure 1? It does not seem to bring any value.
2. what accuracy >50% is considered as " pretty high accuracy"?

typo:
- line 60 blockincluding;
- line 61 (FFNs)lacks;
- line155 constant
- line 265, punctuation
- line 349 outputs is
- line 377 and line 395, what do you mean: the Values corresponding blocks have larger value
missing punctuation in a lot of places.

---

### Official Review · Reviewer_AP8y · 2025-11-01

**Soundness:** 3
**Presentation:** 3
**Contribution:** 3
**Rating:** 4
**Confidence:** 3

**Summary:**

This paper presents a complete analytical characterization of the full Transformer Hessian, including the previously missing LayerNorm and Feed-Forward (FFN) second-order derivatives.  By extending prior attention-only analyses, the authors derive explicit Jacobians and Hessians for every sublayer, provide spectral-norm bounds (Theorems 1 & 6), and formulate a second-order Taylor inequality that yields a data-dependent convergence bound  $|L_{k+1}(w)-L_k(w)|\!\le\!2L/(k{+}1)+M\|w-w^\*\|^2/(k{+}1)$.  Empirical visualizations on ViT blocks confirm heterogeneous curvature (Value–Value terms dominating), and a log–log loss-difference plot aligns with a predicted $1/(k{+}1)$ decay, which the paper interprets as a theoretical underpinning of neural scaling laws.

**Strengths:**

A substantive assessment of the strengths of the paper, touching on each of the following dimensions: originality, quality, clarity, and significance. We encourage reviewers to be broad in their definitions of originality and significance. For example, originality may arise from a new definition or problem formulation, creative combinations of existing ideas, application to a new domain, or removing limitations from prior results.
- First complete derivation of LayerNorm and FFN Hessians within Transformer blocks.
- Clear theoretical formulation linking curvature magnitude to data size and convergence stability.
- Empirical curvature visualizations match analytical predictions.
- Provides a geometric lens that could motivate curvature-aware optimization or adaptive scaling heuristics.

**Weaknesses:**

- Lacks quantitative comparison to established empirical scaling laws [1].
- No experimental or theoretical reconciliation with data-efficient scaling or model-growth strategies [2, 3].
- Assumes shared minima and MSE loss, limiting generality.
- Experiments confined to small-scale ViTs; no language-model validation.
- The connection between Hessian bounds $M$ and power-law exponents $\alpha_D,\alpha_N$ is asserted but not derived.
- The connection to **data-efficient scaling** [2] is not acknowledged. That work already explores data-scarcity scaling breakdowns and recovery via model initialization, which share conceptual ground with curvature-based explanations for optimization difficulty. The present paper could greatly benefit from either comparing to or reinterpreting those findings through its curvature framework.
- The main limitations concern scope and empirical breadth. The theoretical results are derived under simplifying assumptions: single-head attention, post-norm structure, squared-error loss, and a shared minimizer across dataset sizes. While the authors argue that their framework generalizes, explicit proofs or discussions for pre-norm, multi-head, and masked-attention settings are missing. Empirically, the evaluation is confined to Vision Transformers, leaving language or multi-head cases unexplored. Finally, the bound’s tightness depends on a curvature constant $M$ that is not empirically quantified, making the scaling advice somewhat abstract.
- At last, the reference section contains “Placeholder” entries (e.g., Pennington et al., 2017), suggesting incomplete preparation.

[1] Kaplan et al., "Scaling Laws for Neural Language Models," arXiv:2001.08361, 2020.

[2] Wang et al., "Data Efficient Neural Scaling Law via Model Reusing," ICML 2023.

[3] Wang et al., "Learning to Grow Pretrained Models for Efficient Transformer Training (LiGO)," ICLR 2023.

**Questions:**

1. Can the curvature constant $M$ or its dependence on $\|X\|_2$ be empirically fit to reproduce the exponents $\alpha_D,\alpha_N$ reported in [1]?
2. Could the proposed bound explain the phase-transition behavior observed in [2] under data scarcity?
3. How does the curvature-based stabilization compare with model-reusing or progressive-growth mechanisms from [3]?
4. Does the dominance of the Value–Value Hessian persist in multi-head or cross-entropy settings?
5. Could curvature metrics serve as diagnostics for switching from data scaling to model scaling as suggested in compute-optimal policies [3]?

[1] Kaplan et al., "Scaling Laws for Neural Language Models," arXiv:2001.08361, 2020.

[2] Wang et al., "Data Efficient Neural Scaling Law via Model Reusing," ICML 2023.

[3] Wang et al., "Learning to Grow Pretrained Models for Efficient Transformer Training (LiGO)," ICLR 2023.

---

### Note · Program_Chairs · 2026-01-17
**Submission Desk Rejected by Program Chairs**

The following references in this submission do not refer to real documents and/or have major errors in bibliographic information:

 Yuhuai Wu et al. Towards understanding generalization of deep learning. Placeholder - replace with actual entry, 2017. Please provide full citation details.